# Expected Pinball Loss For Quantile Regression And Inverse CDF Estimation

## Abstract

We analyze and improve a recent strategy to train a quantile regression model by minimizing an expected pinball loss over all quantiles. We give an asymptotic convergence rate that shows that minimizing the expected pinball loss can be more efficient at estimating single quantiles than training with the standard pinball loss for that quantile, an insight that generalizes the known deficiencies of the sample quantile in the unconditioned setting. Then, to guarantee a legitimate inverse CDF, we propose using flexible deep lattice networks with a monotonicity constraint on the quantile input to guarantee non-crossing quantiles, and show lattice models can be regularized to the same location-scale family. Our analysis and experiments on simulated and real datasets show that the proposed method produces state-of-the-art legitimate inverse CDF estimates that are likely to be as good or better for specific target quantiles.

## 1 Introduction

Real world applications often seek estimates of the quantiles of a random variable. For example, an airline would like to tell passengers that 90% of the time their flight will arrive within 4 hours, so they know they can make their connecting flights. In addition, one might condition that estimate on features. For example, if the forecast is for 8 inches of snow at the destination airport that day, the conditional estimate might worsen to 90% of the time their flight arriving within 6 hours and 30 minutes.

Quantile regression learns a model from training examples to produce such estimates for quantiles. If the model can estimate all quantiles, it is an inverse CDF estimator.

### 1.1 Formal Definitions

Formally, let random variable $Y \in \mathbb{R}$ have cumulative distribution function (CDF) $F$, and for $\tau \in (0,1)$, the $\tau$-quantile of $Y$ is defined as $q_\tau = F^{-1}(\tau) = \inf\{q : P(Y \leq q) \geq \tau\}$. In the *conditional* setting where one also has random feature vector $X \in \mathbb{R}^D$, the *conditional* $\tau$-quantile of $Y$ for feature vector $X$ is defined as $q_\tau(x) = \inf\{q : P(Y \leq q | X = x) \geq \tau\}$.

*Quantile regression* takes as training data a set of $n$ pairs $(x, y) \in \mathbb{R}^D \times \mathbb{R}$ from a joint distribution over $(X, Y)$, and forms an estimator for one or more of the conditional quantiles of $Y$ for any value of $X$. A standard objective to train a model to estimate the quantile for $\tau$ is to minimize the *pinball loss* (Koenker & Bassett, 1978), $L_\tau(y, \hat{y}) = \max(\tau(y - \hat{y}), (\tau - 1)(y - \hat{y}))$ for $y, \hat{y} \in \mathbb{R}$. In the unconditioned case with no features $X$, the training data is simply a set of $n$ scalar values $\{y_i\}_{i=1}^n$, and minimizing the pinball loss has the satisfying property that it selects the empirical quantile of the training set (Chernozhukov, 2005).

### 1.2 Expected Pinball Loss

Recent work by Tagasovska & Lopez-Paz (2019) proposed training one deep neural network (DNN) model $f(x, \tau)$ that takes $\tau$ as an input, and is trained to minimize an *expected* pinball loss over a random $\mathcal{T}$, drawn from a uniform distribution. That work is important because it provides *simultaneous quantile regression* of

the entire inverse CDF. However, that work left open a couple of important theoretical and practical issues that we address in this paper.

### 1.3  Is Expected Pinball Loss Worse At Specific Quantiles?

The first issue we address is whether one pays a price in estimation accuracy for a model $f(x, \tau)$ that can predict any quantile, compared to a model trained specifically for one target quantile. Surprisingly, we show theoretically with a convergence rate analysis that learning the full inverse CDF model $f(x, \tau)$ can produce a more efficient estimator for a single quantile than minimizing the pinball loss for just a single quantile, depending on the true distribution, quantile, and function class. Our simulations and real-world experiments confirm that one does often win on single quantile estimates by training with the *expected* pinball loss, though the full inverse CDF model $f(\tau; x)$ may take a bit longer to train and a bit more memory to store.

We also demonstrate a novel use case of the expected pinball loss for single quantile regression - training with a Beta distribution over $\tau$ - and show how that can also outperform single-quantile regression (without the added bonus of accurately estimating the full inverse CDF).

### 1.4  The Problem Of Non-crossing Quantiles

The second issue we address is that training a DNN model with the expected pinball loss as proposed by Tagasovska & Lopez-Paz (2019) may fail to produce a legitimate inverse CDF because it does not guarantee *non-crossing quantiles*: that any two quantile estimates satisfy the common sense requirement that $q_\tau(x) \geq q_{\tau'}(x)$ for $\tau \geq \tau'$ at every $x$. For example, a model that does not guarantee non-crossing quantiles might tell a passenger there is a 90% chance their flight will arrive in 4 hours, but that there is a 80% chance their flight will arrive in 4 hours and 12 minutes. Such non-crossing quantile estimates have long been thought objectionable (Koenker & Bassett, 1978).

To test whether non-experts people would actually notice or mind non-crossing estimates, we emailed 100 frequent customers of *Company Name Removed for Blind Review*, who are estimated to be 70% college-educated and 20% with post-graduate education, and told them we were considering giving them estimates of how long it would take for their package to arrive, and gave them example arrival time estimates for the $50\%, 80\%, 90\%$ quantiles, and asked for their feedback on how useful they would find such estimates. However, the example estimates were all made-up: we gave 50 of the customers estimates with crossing 80% and 90% quantiles (3 days, 8 days and 7 days), and the other 50 customers got estimates with non-crossing quantile predictions (3 days, 7 days, 8 days). From the 50 emails with non-crossing estimates, we received 9 emails back to say they would appreciate such estimates, and no concerns. From the 50 emails with crossing estimates, we received 16 emails back all expressing enthusiasm for having such predictions, but 11 of the 16 emails pointed out the crossing quantiles with negative adjectives including "wrong," "broken," "buggy," "didn't make sense to me."

Embarrassing quantile-crossing mistakes are known to happen often in quantile regression (He, 1997; Koenker & Bassett, 1978); see also Table 1. Tagasovska & Lopez-Paz (2019) hypothesized that training with an expected pinball loss induces smoothing across $\tau$ that would reduce non-crossing. However, we show experimentally (Table 3) that a flexible model like a DNN optimized to minimize the expected pinball loss easily suffers a problematic amount of quantile crossing. To address non-crossing without losing model flexibility, we propose using deep lattice networks (DLNs) (You et al., 2017) with a monotonicity shape constraint on $\tau$ to guarantee non-crossing quantiles, thus providing legitimate inverse CDF estimates. Additionally, we show that the DLN function class is amenable to two additional kinds of useful regularization for quantile regression: restricting the learned inverse CDF to a location-scale family, and restricting other features to have only monotonic effect on the predictions.

### 1.5  Other Uncertainty Estimation Problems

In this paper, we focus on estimating multiple quantiles and the extreme case of all quantiles: inverse CDF estimation. Such problems are only part of the landscape of estimating uncertainty. A closely related problem is estimating *prediction intervals* such that an $\alpha$-interval contains the true label alpha is and sharp around $\alpha$.

Prediction intervals can be generated by optimizing for the calibration and sharpness of the interval (Pearce et al., 2018; Chung et al., 2021). Tagasovska & Lopez-Paz (2019) instead formed them by choosing a pair of quantiles from a multiple quantile regression model, which works well because the standard pinball lose used for training quantile regression models balances calibration and sharpness. A different type of uncertainty estimation problem is estimating the uncertainty of a specific statistic, such as the mean in a regression model; such problems are often handled with Bayesian methods. All of these estimation problems become more challenging in the out-of-domain (OOD) setting.

### 1.6   Paper Organization

Next, in Section 2 we formalize the problem of producing legitimate inverse CDF estimates with quantile regression. Then in Section 3 we dig into what is known about beating the standard *pinball loss* by smoothing over multiple quantile estimates, and we give a new asymptotic convergence rate for minimizing the expected pinball loss. In Section 4 we propose using DLNs as the model architecture. Experiments in Section 5 on simulations and real-world data show that the proposed non-crossing DLNs provide competitive, trust-worthy estimates.

## 2   Estimating A Legitimate Inverse CDF

We give a constrained optimization problem to train a quantile regression estimator without crossing quantiles, which can produce a legitimate inverse CDF. Let $\{(x_i, y_i)\}_{i=1}^n$ be a training set where each $x_i \in \mathbb{R}^D$ is a feature vector and $y_i \in \mathbb{R}$ is a corresponding label. Denote the estimator $f(x, \tau; \theta)$, where $\tau \in (0, 1)$ specifies the quantile of interest, and the model $f : \mathbb{R}^{D+1} \to \mathbb{R}$ is parameterized by $\theta \in \mathbb{R}^m$. Recall the standard pinball loss given $\tau$ is $L_\tau(y, \hat{y}) = \max(\tau(y - \hat{y}), (\tau - 1)(y - \hat{y}))$. Let $\mathcal{T} \sim P_\mathcal{T}$ denote a random $\tau$ so we can minimize the *expected* pinball loss over $\mathcal{T}$ to train $f(x, \tau; \theta)$ as in Tagasovska & Lopez-Paz (2019), though here we generalize to $P_\mathcal{T}$, rather than assuming a uniform distribution for $\mathcal{T}$. The standard pinball loss can be written as the special case where $P_\mathcal{T}$ is the Dirac delta distribution on the target quantile.

Also, we constrain the empirical loss minimization with non-crossing constraints, producing the following constrained training objective:

$$\min_\theta \mathbb{E}_\mathcal{T} \sum_{i=1}^n L_\mathcal{T}(y_i, f(x_i, \mathcal{T}; \theta)) \tag{1}$$

$$\text{s.t. } f(x, \tau^+; \theta) \geq f(x, \tau^-; \theta) \, \forall x \in \mathbb{R}^D, \tau^+ \geq \tau^-. \tag{2}$$

### 2.0.1   Related Work In Minimizing Sum Of Pinball Losses

The idea of training one model that can predict multiple quantiles by minimizing a sum of pinball losses with non-crossing constraints dates back to at least the 2006 work of Takeuchi et al. (2006) for a *pre-specified, discrete set* of quantiles. Other prior work used a similar mechanism for a *pre-specified discrete set* of quantiles, and for *more-restrictive* function classes that are monotonic on $\tau$, such as linear models (Bondell et al., 2010), and two-layer *monotonic* neural networks (Cannon, 2018), which are known to have limited flexibility (Daniels & Velikova, 2010). Monotonic DNN models with more layers (Minin et al., 2010; Lang, 2005) or min-max layers (Sill, 1998) can theoretically provide universal approximations, but have not performed as well experimentally as DLNs (You et al., 2017). Monotonic neural nets have also been proposed for estimating a *CDF* (Chilinski & Silva, 2018)).

Training only for (1) without (2) (Tagasovska & Lopez-Paz, 2019) does not guarantee a legitimate inverse CDF because the estimated quantiles can cross. An in-between strategy is to instead add to (1) a penalty on violations of the monotonicity constraint on training samples (and one can add virtual examples as well for more coverage) (Sill & Abu-Mostafa, 1997), but applying that strategy to quantile regression cannot guarantee non-crossing quantiles everywhere.

### 2.0.2   Related Work In Estimating The Inverse CDF

There are two main prior approaches to estimating a *complete* legitimate inverse CDF. The first is nonparametric. For example, k-NN can be extended to predict quantiles by taking the quantiles rather than the mean from within a neighborhood (Bhattacharya & Gangopadhyay, 1990). Similarly, quantile regression forests (Meinshausen, 2006) use random forests leaf nodes to generate local empirical quantiles.

The second strategy is to predict a parametric inverse CDF for any $x$. Traditionally these methods have been fairly rigid. He (1997) developed a method to fit a shared but learned location-scale family across $x$. Yan et al. (2018) proposed a modified 4-parameter Gaussian whose skew and variance was dependent on $x$. Recently, Gasthaus et al. (2019) proposed the *spline quantile function DNN* (SQF-DNN), which is a DNN model that takes an input $x$, and outputs the parameters for a monotonic piecewise-linear quantile function on $\tau$. SQF-DNN can fit any continuous bounded distribution, given sufficient output parameters, and because for any $x$ the output is a monotonic function on $\tau$, guarantees no quantile crossing. Though Gasthaus et al. (2019) focused on RNNs, their framework is easy to apply to the generic quantile regression setting as well, which we do in our experiments.

## 3   Comparison To The Single Quantile Pinball Loss

We show through simulations and convergence rate analysis that minimizing the expected pinball loss can be better at estimating the $\tau$-quantile than minimizing the pinball loss just for $\tau$, and that these positive results are foreshadowed by classic results for unconditional quantile estimates. This section focuses on the unconditioned case (no features $x$), we will return to the problem given features $x$ in Section 4.

### 3.1   Deficiency Of The Sample Quantile

Given only samples from $Y$, minimizing a pinball loss produces the corresponding sample quantile (Chernozhukov, 2005). However, it has long been known that simply taking the sample quantile is not necessarily the best quantile estimator, and in particular that it can be beaten by strategies that smooth multiple sample quantiles (Harrell & Davis, 1982; Kaigh & Lachenbruch, 1982; David & Steinberg, 1986; Reiss, 1980; Falk, 1984).

One important classic result, which follows from the Lehmann-Schaffe theorem, says that if the data are known to be from a uniform $[a, b]$ distribution, then the minimum variance unbiased estimator for the median is *not* the sample median, rather it is the average of the sample min and sample max.

Even if we do not know anything about the true distribution, general-purpose methods that smooth multiple quantiles of the data when estimating a particular quantile have proven effective. In fact, the widely used Harrell-Davis estimator is a weighted average of all the sample order statistics where the weights depend on $\tau$, and the estimator for the median is asymptotically equivalent to computing the mean of bootstrapped medians (Harrell & Davis, 1982).

The prior work above suggests that there can be a better training loss than $\tau$-specific pinball loss for the general problem of quantile regression given features - one that increases efficiency by better smoothing across the quantiles. Indeed, by minimizing the expected pinball loss as in (1), the estimate of any quantile is affected by the other quantiles (given some smoothness in $f$). We provide some further intuition about how it smooths in Appendix A.7.

### 3.2   Comparison to Harrell-Davis, A Classic Quantile Smoother

To emphasize the similarity of minimizing the expected pinball loss to the Harrell-Davis quantile smoothing estimator, which only works in the unconditioned setting of estimating $f(\tau; \theta)$, we compare both to the sample median. We simulate $Y$ drawn from an exponential distribution.

The expected pinball loss approach lets us tune how much to smooth over quantiles in two different ways, both of which effectively recover the sample quantiles at one extreme. The first approach is most directly analogous to the Harrell-Davis estimator: minimizing not just the target pinball loss, but a weighted average

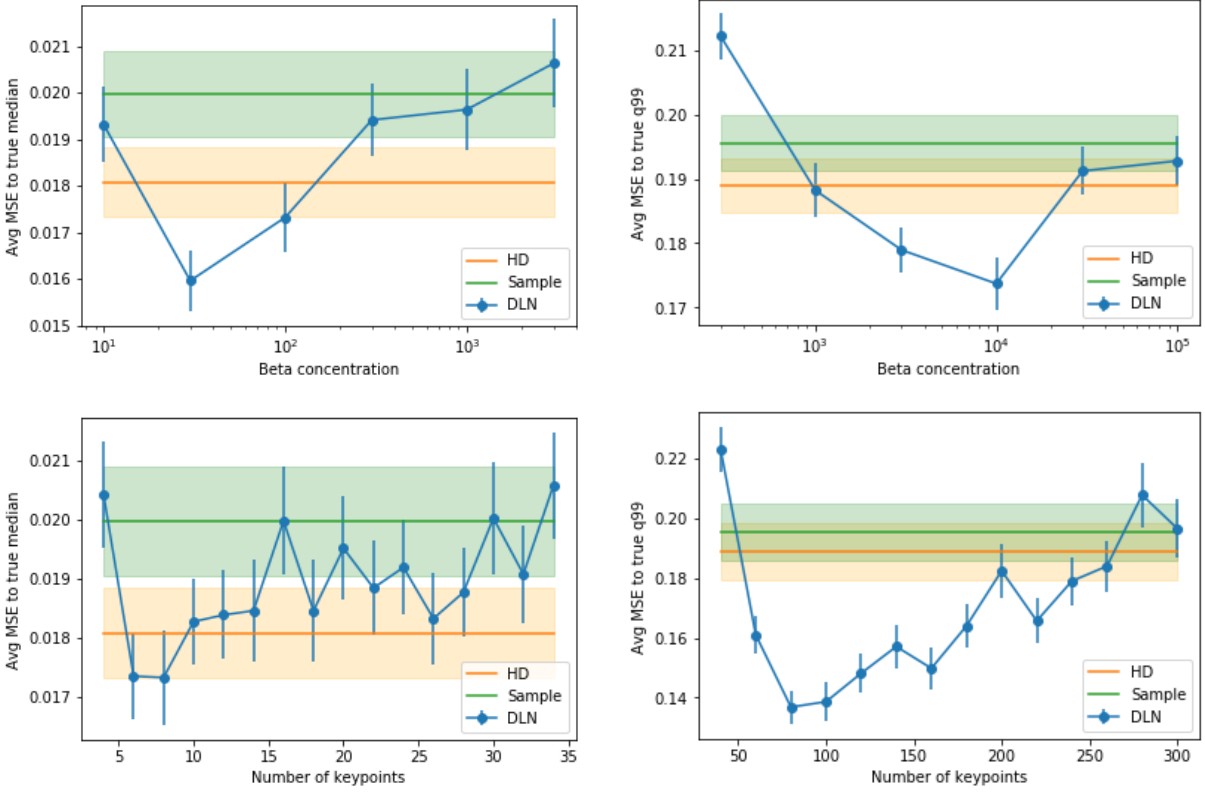

Figure 1: Estimating the median (left plots) and 99th percentile (right plots) of an exponential distribution (no features) by the sample median, by Harrell-Davis, and by solving (1) and (2) with a DLN function class. The median estimates were from $N = 51$ training samples. The $\tau = .99$ estimates were from $N = 505$ training samples. Error bars are computed over 1,000 repeats. **Top:** The DLN is just a linear model on $\tau \in [0, 1]$ with $\theta \in \mathbf{R}^2$ and $P_{\mathcal{T}}$ is a beta distribution. Results are shown for six choices of the beta concentration across the x-axis, where a bigger concentration makes $P_{\mathcal{T}}$ more peaked around the target quantile, imposing less smoothing across quantiles. **Bottom:** The $P_{\mathcal{T}}$ is the uniform distribution, and the DLN is a piece-wise linear function (PLF) with $K$ keypoints. Larger $K$ makes the model more flexible, and imposes less smoothing across quantiles. Results for different choices of $K$ are shown across the x-axis.

of pinball losses around the target quantile. The second, more ambitiously, trains over all quantiles equally, but controls the complexity of the function class that models the inverse CDF.

For the first set of experiments, we use a beta distribution for $P(\mathcal{T})$ with its mode set to the quantile of interest ($\tau = 0.50$, or $\tau = 0.99$), leaving its concentration as a hyperparameter. We restrict $f(\tau)$ to be linear (which is a special case of the proposed DLN function class, see Sec.4).

Results are shown in the top row of Figure 1. As expected, the sample median is not as good as the Harrell-Davis estimate, which regularizes across the quantiles similar to the expected pinball loss. The more concentrated the beta distribution is (further right on the x-axis), the less regularized the DLN estimator is; in the limit all the mass of $P_{\mathcal{T}}$ is concentrated on the target quantile and (1) approaches the standard pinball loss. For both the median (left) and 99th percentile (right) estimate, most choices of the beta concentrations perform better than the sample median, and a broad swath of beta concentrations perform better than Harrell-Davis. In practice, one would need to cross-validate the concentration hyperparameter.

An alternative is to use a uniform distribution for $P_{\mathcal{T}}$, and instead control the amount of regularization across $\tau$ by using a more flexible function $f(\tau)$. We test that strategy in our second set of experiments. We use a piecewise-linear function (PLF) $f(\tau; \theta)$, where the parameters $\theta$ are the values of $f(\tau; \theta)$ at $K$ keypoints

Table 1: Estimating the median and 99th percentile of an exponential distribution via cross-validated DLN training. This is a companion to Figure 1 in which results are additionally provided for a DLN in which the number of keypoints is tuned via 10-fold cross-validation over the $N = 51$ or $N = 505$ training points, respectively, and the model is then retrained on the full data with the selected number of keypoints.

| Model | $\tau = 0.5$ | $\tau = 0.99$ |
|---|---|---|
| CV-DLN, Expected Pinball | $0.019263 \pm 0.00086$ | $\mathbf{0.17566 \pm 0.0080}$ |
| Sample Quantile | $0.019968 \pm 0.00046$ | $0.19552 \pm 0.0043$ |
| Harrell-Davis | $\mathbf{0.018081 \pm 0.00038}$ | $0.18887 \pm 0.0042$ |

pinned at the $K$ sample quantiles of the data (which is a special case of the proposed DLN function class, see Section. 4). Bigger $K$ makes $f(\tau; \theta)$ more flexible, which reduces the dependence of the target quantile on the impact other quantiles have on the expected loss. The results are shown in the bottom row of Figure 1. On the left for the median estimation, over a wide range of model flexibility $K$, the DLN performs better than the sample median, and for a small range of $K$ better than the Harrell-Davis estimator.

On the right-side of the bottom row of Figure 1, the results are stronger for estimating the tail $\tau = 0.99$ from $N = 505$ training samples. The DLN solving (1) and (2) statistically significantly beats the Harrell-Davis method for a broad range of model flexibility as expressed by the $K$ keypoints in the DLN's PLF.

Table 1 shows that when we choose the value of $K$ based on 10-fold cross-validation, the DLN outperforms the sample quantile but not the Harrell-Davis estimator when predicting the median, and outperforms both when predicting $\tau = 0.99$.

The Harrell-Davis and related estimators can only be applied to the unconditioned case, whereas training with the expected pinball loss (1) and monotonicity constraints (2) extends seamlessly to *conditional* quantile estimation with feature vector $x \in \mathbf{R}^D$, bringing the gains of regularizing across $\tau$ to a much larger class of problems.

### 3.3 Convergence Rate

We give a new asymptotic convergence rate for estimating the full inverse CDF by minimizing the expected pinball loss (1), optionally with a monotonicity constraint (2), and compare that to the convergence rate of single-quantile estimation learned by minimizing an individual pinball loss.

We show that depending on the underlying data distribution and the function class chosen to estimate the inverse CDF, the simultaneous estimation can produce a more efficient estimator for a single target quantile than minimizing the pinball loss for only that quantile.

For tractability, we work in the setting where there are no features $x$ to condition on, only a set of training samples $\{y_i\}$, and we are interested in estimating the quantile $F^{-1}(\tau)$ for a random variable $Y \in \mathbb{R}$ given IID samples $\{y_i\}_{i=1}^n$. Such analysis is relevant to the conditional case because for function classes that are flexible over $X$, they intuitively will have similar dynamics at each value of $X$ as the unconditioned case we work with here. We formalize this in a basic corollary which extends our unconditional theory to conditional estimation for finite categorical $X$ and an extremely flexible function class, but note that general theory for full conditional inverse CDF estimation remains an open problem. Section 5.5 provides experimental evidence for the conditional case.

#### 3.3.1 Review of Single Pinball Loss Convergence Rate

We start with the well-understood case of single-quantile estimation. Let $\theta_\tau^* = \arg \min_\theta E_Y[L_\tau(Y, \theta)]$ be the true minimizer for the corresponding pinball loss and $\hat{\theta}_\tau^{(n)} = \arg \min_\theta \frac{1}{n} \sum_{i=1}^n [L_\tau(y_i, \theta)]$ be the empirical minimizer over our data. (In our context, $\theta^*(\tau)$ is the true $\tau$ quantile of $Y$ and $\hat{\theta}_\tau^{(n)}$ is the $\tau$ sample quantile of the data (Chernozhukov, 2005).) The asymptotic convergence rate for this single quantile estimate is

given by,

$$\sqrt{n}(\hat{\theta}_\tau^{(n)} - \theta_\tau^*) \xrightarrow{d} \mathcal{N}\left(0, \frac{\tau(1-\tau)}{p(\theta_\tau^*)^2}\right), \tag{3}$$

where $p(y)$ is the density for $Y$ (Koenker, 2005).

### 3.3.2 Expected Pinball Loss Convergence Rate

Next, consider learning the full inverse CDF $f(\tau; \theta)$ parameterized by $\theta \in \Theta \subseteq \mathbb{R}^m$ by minimizing the expected pinball loss:

$$\theta^* = \underset{\theta \in \Theta}{\arg\min}\, E_{\mathcal{T}}[E_Y[L_{\mathcal{T}}(Y, f(\mathcal{T}; \theta))]],$$

where $\mathcal{T} \in (0, 1)$ is a random variable drawn from some distribution independently of $Y$. Note that the monotonicity constraints in Equation (2) can be wrapped in the function class $f$ and feasible parameter set $\Theta$. Let $\hat{\theta}^{(n)}$ be the corresponding minimizer under an empirical distribution over $Y$ with samples $\{y_i\}_{i=1}^n$. In Theorem 1 below, we present a general asymptotic convergence rate for $\hat{\theta}^{(n)}$ that depends on the properties of the parametric function $f$, feasible parameter space $\Theta$, and the distribution of $\mathcal{T}$. A natural choice for the distribution of $\mathcal{T}$ is Unif$(0, 1)$, but in fact, depending on the distribution of $Y$ and properties of the function class $f(\tau; \theta)$, other distributions could lead to lower asymptotic variance. In choosing the distribution of $\mathcal{T}$, note that choosing distributions with wider support will make it easier to satisfy the requirements for asymptotic normality in Theorem 1 below (see note in Appendix A.4).

**Theorem 1.** *Suppose $f(\tau; \theta)$ is well specified with a unique $\theta^* \in \Theta \subseteq \mathbb{R}^m$ such that $f(\tau; \theta^*) = F^{-1}(\tau)$ for all $\tau \in (0, 1)$, and that $\theta^*$ is also the unique minimizer for the risk $R(\theta) = E_{Y,\mathcal{T}}[L_{\mathcal{T}}(Y, f(\mathcal{T}; \theta))]$. Suppose the estimator $\hat{\theta}^{(n)}$ is weakly consistent, $\hat{\theta}^{(n)} \xrightarrow{P} \theta^*$. Suppose further that the function $\theta \mapsto f(\tau; \theta)$ is continuous, locally Lipschitz, and twice differentiable at $\theta = \theta^*$ for all $\tau \in (0, 1)$. Then,*

$$\sqrt{n}(f(\tau; \hat{\theta}^{(n)}) - f(\tau; \theta^*)) \xrightarrow{d} \mathcal{N}\left(0, \nabla_\theta f(\tau; \theta^*)^\top Q^{-1} V (Q^{-1})^\top \nabla_\theta f(\tau; \theta^*)\right) \tag{4}$$

*where $Q = E_{\mathcal{T}}[p(f(\mathcal{T}; \theta^*))\Gamma(\mathcal{T})]$, $V = E_{\mathcal{T}}[\mathcal{T}(1 - \mathcal{T})\Gamma(\mathcal{T})]$, with $\Gamma(\tau) = \nabla_\theta f(\tau; \theta^*)\nabla_\theta f(\tau; \theta^*)^\top$.*

Lemma 1 provides a simple example of sufficent conditions for the consistency condition in Theorem 1, which in general depends on properties of the function $f$ and parameter space $\Theta$.

**Lemma 1.** *If $f(\tau, \theta)$ is convex in $\theta$ for all $\tau \in (0, 1)$, $\Theta$ is convex, and $\theta^*$ is in the interior of $\Theta$, then $\hat{\theta}^{(n)}$ is weakly consistent, $\hat{\theta}^{(n)} \xrightarrow{P} \theta^*$.*

Lemma 1 applies when the optimization problem in (1) is convex. Aside from convexity, other example conditions could include compactness of $\Theta$, or further Lipschitz continuity or bounded moment conditions on $f$ (Powell, 2010).

Given a specific distribution of $\mathcal{T}$, a function class of the inverse CDF $f$, and a data distribution $Y$, we can directly compare the asymptotic convergence rate of the direct quantile estimates (Equation (3)) to that of quantile estimates from the learned inverse CDF (Equation (4)). We illustrate this below with several examples.

### 3.3.3 Example 1: Single Parameter Uniform

To build intuition for the implications of Theorem 1, we consider a centered uniform distribution $Y \sim$ Unif$(-\frac{\alpha}{2}, \frac{\alpha}{2})$. Let $f(\tau; \theta) = \theta(\tau - \frac{1}{2})$. Expanding the asymptotic variance term in Theorem 1,

$$Q^{-1} V (Q^{-1})^\top = \frac{\alpha^2 E_{\mathcal{T}}[-\mathcal{T}^4 + 2\mathcal{T}^3 - \frac{5}{4}\mathcal{T}^2 + \frac{1}{4}\mathcal{T}]}{E_{\mathcal{T}}[\mathcal{T}^2 - \mathcal{T} + \frac{1}{4}]^2}.$$

This depends on the first four moments of the distribution of $\mathcal{T}$, the choice of which is up to the algorithm designer. For the natural choice of $\mathcal{T} \sim$ Unif$(0, 1)$, the asymptotic variance for estimating a specific $\tau_0$-quantile using the function $f(\tau_0; \hat{\theta}^{(n)}) = \hat{\theta}^{(n)}(\tau_0 - \frac{1}{2})$ is $1.2\alpha^2(\tau_0 - \frac{1}{2})^2$. This variance is lowest when $\tau_0 = 0.5$, and increases to $0.3\alpha^2$ at the most extreme quantiles when $\tau = 0$ or $\tau = 1$.

We next compare this to estimating the $\tau_0$-quantile using the single pinball loss $L_{\tau_0}$. Equation (3) shows that the estimate $\hat{\theta}_{\tau_0}^{(n)}$ has asymptotic variance $\alpha^2 \tau_0(1 - \tau_0)$. This shows that depending on the exact value of $\tau_0$, the single quantile estimate could be less or more efficient than the estimate resulting from evaluating the function $f(\tau_0; \hat{\theta}^{(n)})$. The single quantile estimate is more efficient for the extreme quantiles, whereas the inverse CDF estimate is more efficient for estimating the median. Intuitively, the inverse CDF median estimate benefits from fitting the inverse CDF function to the surrounding data points, which is reminiscent of classic quantile smoothing.

### 3.3.4 Example 2: Two-Parameter Uniform

We next consider a two-parameter uniform distribution example: for $Y \sim \mathrm{Unif}(\alpha, \beta)$, $p(y) = \frac{1}{\beta - \alpha}$. Let $\theta \in \mathbb{R}^2$ with $f(\tau; \theta) = \theta_0 + \theta_1 \tau$. That is, we learn a two-parameter line as our inverse CDF. We then have $\nabla_\theta f(\tau; \theta) = \begin{bmatrix} 1 \\ \tau \end{bmatrix}$ and can therefore write the $Q$ and $V$ matrices from Theorem 1 as

$$Q = \frac{1}{\beta - \alpha} E_{\mathcal{T}} \begin{bmatrix} 1 & \mathcal{T} \\ \mathcal{T} & \mathcal{T}^2 \end{bmatrix}; \qquad V = E_{\mathcal{T}} \begin{bmatrix} \mathcal{T} - \mathcal{T}^2 & \mathcal{T}^2 - \mathcal{T}^3 \\ \mathcal{T}^2 - \mathcal{T}^3 & \mathcal{T}^3 - \mathcal{T}^4 \end{bmatrix}.$$

Evaluating each of these matrices for $\mathcal{T} \sim \mathrm{Unif}(0, 1)$ we arrive at

$$Q^{-1} V (Q^{-1})^\top = (\beta - \alpha)^2 \begin{bmatrix} \frac{7}{15} & -\frac{3}{5} \\ -\frac{3}{5} & \frac{6}{5} \end{bmatrix},$$

and therefore an asymptotic variance for a particular $\tau_0$-quantile estimate $f(\tau_0; \hat{\theta}^{(n)}) = \hat{\theta}_0 + \hat{\theta}_1 \tau_0$ of $(\beta - \alpha)^2(\frac{7}{15} + \frac{6}{5}\tau_0^2 - \frac{6}{5}\tau_0)$. This is a quadratic function that finds its minimum at $\tau = 0.5$ and is highest (on the unit interval) at $\tau = 0$ and $\tau = 1$.

We can compare it to the convergence for single-pinball regression of $(\beta - \alpha)^2 \tau(1 - \tau)$, which is a quadratic function that attains its *maximum* at $\tau = 0.5$.

As we see in Figure 2, the variance is smaller for the expected pinball regression for quantiles between roughly $\tau = 0.3$ and $\tau = 0.7$. The opposite is true for the extreme quantiles.

We validate these results empirically through simulation in Table 2. In each simulation, we draw 1,000 data points from Uniform(0,1) and seek to best estimate the true 10th, 50th, and 90th quantiles (i.e. 0.1, 0.5, and 0.9). We train a 2-keypoint DLN, which effectively learns a linear model $\theta_0 + \theta_1 \tau$, using the expected pinball loss, and compare it to taking the sample quantile or using the Harrell-Davis estimator. We find the DLN performs best at estimating the median but worse at the extreme quantiles. Note that this pattern differs from the results we see in Figure 1, where on an exponential distribution, a flexible DLN without knowledge of the true distribution is relatively better at an extreme quantile. This is because the specific dynamics depend on the data distribution and the functional form of the model trained with the expected pinball loss.

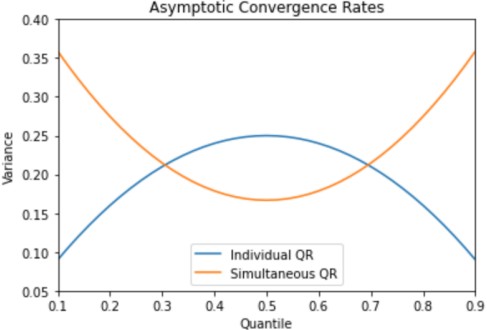

Figure 2: Plots of the asymptotic convergence variances for the expected pinball regression (orange) and individual single-pinball regressions (blue) for different quantiles, when we assume that the true distribution is Uniform($\alpha, \beta$), the expected pinball loss regression learns a linear model, and we assume $\beta - \alpha = 1$. Relaxing the last assumption would scale the y-axis but not change the intersection points.

Overall, these examples illustrate the sometimes counterintuitive fact that it can be more asymptotically efficient to estimate a quantile through learning a full inverse CDF than it would be to just estimate the

Table 2: Simulation results for unconditional quantile estimation given 1,000 samples of a Uniform(0,1) random variable. Results are shown for monotonic DLNs trained with the expected pinball loss over uniformly sampled $\tau$, sample quantiles, and the Harrell-Davis estimator. DLNs are learned with 2 keypoints in order to embed the correct linear assumption on the inverse CDF.

| Model | $\tau = 0.1$ | $\tau = 0.5$ | $\tau = 0.9$ |
|---|---|---|---|
| DLN, Expected Pinball | $0.000109 \pm 0.000005$ | $\mathbf{0.000077 \pm 0.000004}$ | $0.000109 \pm 0.000005$ |
| Sample Quantile | $\mathbf{0.000097 \pm 0.000004}$ | $0.000245 \pm 0.000010$ | $\mathbf{0.000086 \pm 0.000004}$ |
| Harrell-Davis | $\mathbf{0.000089 \pm 0.000004}$ | $0.000247 \pm 0.000011$ | $\mathbf{0.000083 \pm 0.000004}$ |

single quantile alone. In learning the full inverse CDF, the algorithm designer also gets to make a key choice in the distribution of $\mathcal{T}$ used in the learning objective. The examples above showed how the asymptotic convergence rates depend on the moments of $\mathcal{T}$, and we empirically revisit the choice of the distribution of $\mathcal{T}$ in experiments Section 5.

### 3.4 Extension to Conditional Quantile Estimation

The asymptotic theory so far shows that expected pinball loss estimators can sometimes outperform single quantile estimators in the absence of conditioning on any additional features $X$. Conditional quantile estimation in general adds a considerable amount of theoretical complexity which we do not fully address here, and to the best of our knowledge, characterization of convergence rates for full conditional inverse CDF estimation remains an open problem. However, the provided unconditional asymptotic theory can yield intuition and extreme examples for cases when estimating the full conditional inverse CDF would outperform single conditional quantile estimation. For instance, when $X$ is a finite categorical feature, and the function $f$ and parameter space $\Theta$ are expressive enough in $X$, Theorem 1 can be directly applied to the individual quantile estimation problems conditioned on each finite value of $X$. We formalize this in the following Corollary.

**Corollary 1.** *Let $X$ be distributed uniformly over a finite set of categorical values $\mathcal{X}$, letting $m = |\mathcal{X}|$. Suppose that a dataset $\{(x_i, y_i)\}_{i=1}^n$ is created by sampling $\{y_j\}_{j=1}^{\frac{n}{m}}$ values IID from the conditional distribution $Y|X = x$ for each value of $x$ (assuming $m$ divides $n$), and taking the union of these sets.*

*Let $f : \mathcal{X}, (0,1), \Theta \to \mathbb{R}$ be fully separable over $x$; that is, let $\Theta = \Theta_1 \times ... \times \Theta_m$, where $\Theta_x$ represents a copy of a given parameter space $\Theta_0$ for each value of $x \in \mathcal{X}$. Let $f$ take the form $f(x, \tau; \theta) = \sum_{x_0 \in \mathcal{X}} \mathbb{1}(x = x_0)g(\tau; \theta_{x_0})$ for some function $g : (0,1), \Theta_0 \to \mathbb{R}$.*

*Suppose $f(x, \tau; \theta)$ is well specified with a unique $\theta^* \in \Theta \subseteq \mathbb{R}^m$ such that $f(x, \tau; \theta^*) = q_\tau(x)$ for all $\tau \in (0,1)$ and all $x \in \mathcal{X}$, and that $\theta^*$ is also a unique minimizer for the risk $R(\theta) = \frac{1}{m} \sum_{x \in \mathcal{X}} E_{Y,\mathcal{T}}[L_{\mathcal{T}}(Y, f(x, \mathcal{T}; \theta))|X = x]$. Suppose the estimator $\hat{\theta}^{(n)}$ is weakly consistent, $\hat{\theta}^{(n)} \xrightarrow{P} \theta^*$. Suppose further that the function $\theta \mapsto f(x, \tau; \theta)$ is continuous, locally Lipschitz, and twice differentiable at $\theta = \theta^*$ for all $\tau \in (0,1)$ and all $x \in \mathcal{X}$. Then for each $x \in \mathcal{X}$,*

$$\sqrt{n/m}(f(x, \tau; \hat{\theta}^{(n)}) - f(x, \tau; \theta^*)) \xrightarrow{d} \mathcal{N}\left(0, \nabla_\theta f(x, \tau; \theta^*)^\top Q^{-1} V (Q^{-1})^\top \nabla_\theta f(x, \tau; \theta^*)\right)$$

*where $Q = E_{\mathcal{T}}[p_x(f(x, \mathcal{T}; \theta^*))\Gamma(\mathcal{T})], V = E_{\mathcal{T}}[\mathcal{T}(1 - \mathcal{T})\Gamma(\mathcal{T})]$, with $p_x(y)$ being the density of $Y|X = x$ and $\Gamma(\tau) = \nabla_\theta f(x, \tau; \theta^*) \nabla_\theta f(x, \tau; \theta^*)^\top$.*

More generally, function classes with a high degree of flexibility such as neural networks may approach this extreme categorical separability described above. We describe one such flexible function class below which, given enough parameters, can be specified to meet this separability condition for finite categorical features. Of course, computational constraints and generalizability must also be considered.

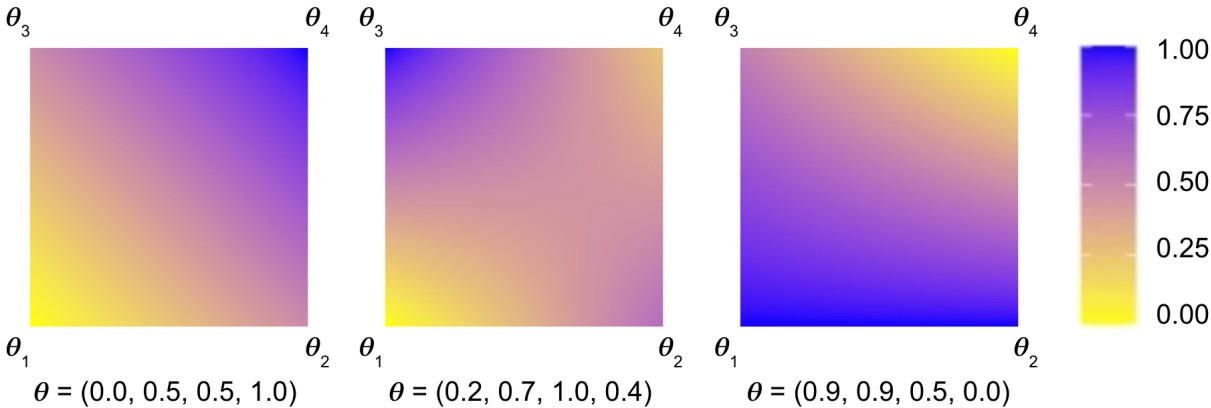

Figure 3: Example 2D lattices sized $2 \times 2$ have four parameters. Each parameter represents the value the lattice function takes at one of the four corners of the input domain $[0, 1]^2$. The lattice function values in-between the vertices are computed by bilinear interpolation of the four parameter values, so these functions can be re-parameterized and expressed as bilinear polynomial functions over $[0, 1]^2$ of the form $f(x) = a + bx_1 + cx_2 + dx_1x_2$. A 2D lattice with more knots, e.g. $3 \times 3$, would be locally bilinear across each cell of the lattice.

## 4  DLNs for Legitimate but Flexible Inverse CDFs

We propose using DLNs (You et al., 2017) as the function class in (1) because of their flexibility and because they efficiently enable three key regularization strategies: non-crossing constraints, restricting the learned distribution to location-scale families, and monotonic features, as explained below.

### 4.1  Review of Deep Lattice Networks

DLNs are multi-layer models where some layers are lattices or ensembles of lattices. Lattices build upon a truly ancient strategy for approximating functions over a bounded domain (Campbell-Kelly et al., 2003): record the function's values on a regular grid of inputs, and then one can linearly interpolate between the recorded values. Generalizing to multiple dimensions, a lattice function is a multi-dimensional look-up table whose parameters are the look-up table values, and the function is produced by linearly interpolating the look-up table values surrounding an input point. See Fig. 3 for example 2D lattices.

In the special case of a one-dimensional domain, a lattice is just a piecewise-linear function. We recommend reading Gupta et al. (2016) for the basics of lattices with monotonicity constraints. See also Garcia et al. (2012) for basics of lattice models and their relationship to other splines.

Mathematically, a lattice function $f(x) : \mathbf{R}^D \to \mathbf{R}$ can be expressed $f(x) = \theta^T \psi(x)$, where the parameters $\theta \in \mathbf{R}^p$ are the look-up table values, and the nonlinear transformation $\psi(x) \in \mathbf{R}^p$ maps an input $x \in \mathbf{R}^D$ to the ultra-sparse vector of linear interpolation weights over the look-up table values.

Garcia & Gupta (2009) showed that lattice models can be trained using a standard empirical risk minimization framework. Lattices can approximate any continuous bounded function if they are sampled on a regular grid with enough knots.

A key reason to use lattice functions in an application like quantile regression is that the regular structure of their parameters makes them amenable to shape constraints. Here, the constraint (2) requires $f(x, \tau; \theta)$ to be monotonically increasing in $\tau$, which is imposed efficiently with sparse linear inequality constraints on the lattice parameters corresponding to forcing any neighboring gridpoints in the direction of $\tau$ to be increasing (Gupta et al., 2016). To more efficiently represent functions, lattice models are usually architected with a first layer that separately calibrates each input with a learned one-dimensional piecewise linear function

(PLF) whose knots need-not be regularly-spaced referred to as a calibrator, then fuses the calibrated inputs together with a coarser multi-dimensional lattice (Garcia et al., 2012; Gupta et al., 2016).

The main drawback to lattices is that the number of lattice parameters needed to approximate a $D$-dimensional functions blows up exponentially as $O(2^D)$. This problem is handled by making an ensemble of calibrated lattices (Canini et al., 2016) much as one makes a random forest, and these calibration and lattice layers can be mixed with other layers like linear layers, and cascaded ad infinitum, forming arbitrarily *deep lattice networks* (DLNs) (You et al., 2017; Cotter et al., 2019). Importantly, as long as each layer of a DLN is trained to respect a monotonic feature like $\tau$, the entire DLN will be monotonic in $\tau$.

More generally, one could use unrestricted ReLU or embedding layers for the first few model layers on $x$, then fuse in $\tau$ later with $\tau$-monotonic DLN layers. Figure 4 illustrates how an arbitrary neural net architecture can be used to reduce the dimensionality of the input features of $x$ before fusing with $\tau$, allowing our approach to be used for essentially any type and size of input.

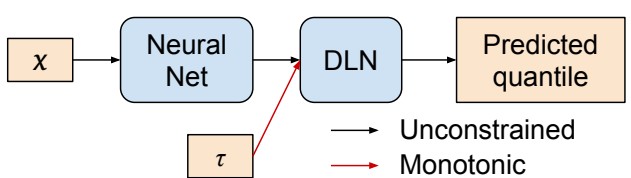

Figure 4: Block diagram showing how an arbitrary neural net architecture can be used to reduce the dimensionality of $x$ to a more manageable size, before combining in a DLN architecture with $\tau$, which is constrained to be monotonic.

In our experiments, our DLNs are built with the standard two-layer calibrated lattice (Gupta et al., 2016) and calibrated-ensemble-of-lattices (Canini et al., 2016) architectures offered by the open-source TensorFlow Lattice library (TensorFlow Blog Post, 2020). The TensorFlow Lattice library enables solving the constrained optimization problem (1) and (2) with Dykstra's projection algorithm. In our experiments, we found our DLN models trained in a similar amount of time as the DNN and SQF-DNN models.

### 4.2 Location-Scale Family Models

We show that by different architecture choices for the DLN model used can regularize the estimated inverse CDF in semantically-meaningful ways, similar to prior more rigid distributional approaches. As a simple example, if one constrains the DLN architecture to not have any interactions between $\tau$ and any of the $x$ features, then one learns a regression model with homoskedastic errors. As another example, the basic two-layer DLN called a *calibrated lattice* model (Gupta et al., 2016) described in Section 4.1 can be constrained to learn distributions across $x$ that come from a shared, learned, location-scale family:

**Lemma 2.** Let $f(x, \tau)$ be a calibrated lattice model with piece-wise linear calibrator $c(\tau) : [0, 1] \to [0, 1]$ for $\tau$, and let there be two knots in the regular grid in the direction of $\tau$, and the lattice parameters are interpolated with standard multilinear interpolation. Then $f(x, \tau)$ represents an inverse CDF function $F^{-1}(y|x)$ where the estimated distribution for every $x$ is from the same location-scale family as the calibrator $c(\tau)$.

The proof is in the Appendix. Note the number of keypoints in $c(\tau)$ controls the complexity of the learned base distribution, allowing the model to approximate location-scale families like the Gaussian, gamma, or Pareto distributions. Then in the second layer of $f$, the number of knots in the lattice over the $c(\tau)$ input controls how much the distribution should be allowed to vary across $x$. Two knots in the lattice limits us to a shared location-scale family, as noted above, while three knots in the lattice gives an extra degree of freedom to shrink or stretch one side of the distribution differently across $x$. More lattice knots across $\tau$, or ensembling of lattices, or more layers can all be used to move the model towards full generality.

### 4.3 Other Monotonic Features

Using DLNs also enables imposing monotonicity on the features $x$ if domain knowledge says they should have a monotonic impact on the estimates (Gupta et al., 2016). For example, a pizza delivery time estimator might treat constrain an input quantifying how bad traffic is to only increase estimated quantiles (that is, worse traffic never makes the estimated delivery time shorter).

Our experience with real-world applications of quantile estimation is that real-world quantile estimates do often use features that are past measurements (or otherwise strong correlates) to predict the future distribution of measurements. For example, in the real-world Puzzles experiment in Section 5, we predict the quantiles of how long a club member will keep a rented puzzle, and one of the features is how long they kept their last puzzle. We constrain that input to have a monotonically positive effect on the estimate. (However, earlier hold-times are not necessarily monotonically indicative of the next future hold-time, because they may indicate a strongly decreasing trend over time). Domain knowledge about the relationship between the model inputs-and-ouput that can be captured with monotonicity constraints is a tuning-free, semantically-meaningful regularizer that can improve test performance and increase model explainability (Gupta et al., 2016), and where applicable, even help capture ethical rules (Wang & Gupta, 2020).

DLNs also enable multi-dimensional shape constraints to capture domain knowledge like complementary features (Gupta et al., 2020), and functions defined on sets (Cotter et al., 2019).

## 5 Experiments

Using simulations and real data, we first show that training a DLN that is monotonic in $\tau$ to satisfy both (1) and (2) works well compared to training a DNN for (1) as done in Tagasovska & Lopez-Paz (2019) or the SQF-DNN parametric approach of Gasthaus et al. (2019); the results are in Tables 3 and 5. Then we use the proposed $\tau$-monotonic DLN models to compare training to minimize expected pinball loss versus training to minimize the pinball loss for a specific $\tau$; the results are in Tables 7 and 8.

Bolded results in tables indicate that the metric is not statistically significantly different from the best metric among the models being compared, using an unpaired t-test.

### 5.1 Model Training Details

All hyperparameters were optimized on validation sets. We used Keras models in TensorFlow 2.2 for the unrestricted DNN comparisons that optimize (1) (Tagasovska & Lopez-Paz, 2019), and the SQF-DNN (Gasthaus et al., 2019), which optimizes the same objective in a different manner while also guaranteeing non-crossing quantile estimates. For DLNs, we used the TensorFlow Lattice library (TensorFlow Blog Post, 2020). For all DNN and DLN experiments, we use the Adam optimizer (Kingma & Ba, 2015) with its default learning rate of 0.001, except where noted. For DNN models, we validated the number of hidden layers and the hidden dimension. For the SQF-DNN, we also validated the number of distribution keypoints. For the smaller DLN models, we used the common two-layer calibrated lattice architecture (Gupta et al., 2016) and validated over its number of calibration keypoints and lattice vertices. For the larger datasets, we used an ensemble of calibrated lattices for the DLN model, and then also validated over the the number and dimensionality of the base models in the ensemble (Canini et al., 2016). For both DLNs and DNNs, we additionally validated over the number of training epochs.

### 5.2 Benchmark and Real Datasets Used

We tested on three publicly available datasets and one proprietary dataset for real-world problems. A co-lab will be made available.

**Air Quality:** The Beijing Multi-Site Air-Quality dataset from UCI (Zhang et al., 2017) contains hourly air quality data from 12 monitoring regions around Beijing. We predict the quantiles of the PM2.5 concentration from $D = 7$ features: temperature, pressure, dew point, rain, wind speed, region, and wind direction. The DLN model is an ensemble of two-layer calibrated lattice models. We split the data by time (not IID) with earlier examples forming a training set of size 252,481, later examples a validation set of size 84,145, and most recent examples a test set of size 84,145.

**Puzzles:** This is a dataset from *(name withheld for blind review)*, a private library for wooden jigsaw puzzles. We predict how long a library member will hold a borrowed puzzle given their $D = 5$ previous hold-times. The DLN model is an ensemble of two-layer calibrated lattice models. Based on their domain knowledge, we also constrain the DLN output to be monotonically increasing in the most-recent-past hold-time feature.

The 936 train and 235 validation examples are IID from past data, while the 210 test samples are the most recent samples (not IID). The anonymized dataset is publicly available on their website *(website withheld for blind review)*.

**Traffic:** This is a proprietary dataset from a large internet services company *(name withheld for blind review)* for estimating the time to drive a route. The DLN is a two-layer calibrated lattice whose $D = 4$ inputs are 1 categorical and 3 continuous features. We used 1,000 examples each for training, validation, and testing, with the training examples occurring earlier in time than the validation and test examples (not IID).

**Wine:** We used the Wine Reviews dataset from Kaggle (Bahri, 2018). We predict the quantiles of wine quality on a 100-point scale. We used $D = 42$ features, including price, a 1-d learned embedding for the country of origin (aka categorical calibration (Gupta et al., 2016)), and 40 Boolean features describing each wine. The DLN model is an ensemble of two-layer calibrated lattice models. The DLN output is constrained to be monotonically increasing in the *price* feature, as suggested in prior work (Gupta et al., 2018). The data was split IID with 84,641 examples for training, 12,091 for validation, and 24,184 for testing.

### 5.3 Model Architecture Experiments

We start by demonstrating the efficacy of using monotonic DLNs to predict the inverse CDF on simulations, and then on the real data.

#### 5.3.1 Simulations

We used simulations from a recent quantile regression survey paper (Torossian et al., 2020) based on the sine, Griewank, Michalewicz, and Ackley functions with carefully designed noise distributions to represent a range of variances and skews across the respective input domains. We used 250 training examples for the 1-D sine and Michalewicz functions, 1,000 examples for the 2-D Griewank function, and 10,000 examples for the 9-D Ackley function.

Our metric is the average squared difference between the estimated and true quantile curves sampled at 99 uniform quantiles $\tau \in 0.01, \ldots, 0.99$, averaged over 100 repeats, and averaged over values of $x$ across the domain (we use fine grids of 1,000 and 10,000 $x$ values for the 1-D and 2-D experiments, respectively, and a random sample of 10,000 $x$-values for the 9-D experiment). We also report the fraction of test points for which at least two of their 99 quantiles crossed.

Results in Table 3 demonstrate that the proposed monotonic-in-$\tau$ DLNs are the best or statistically tied for the best across all four disparate simulations. The DNNs were trained with the same sampled expected pinball loss, and are sometimes close in performance, but suffer substantially from crossing quantiles. For example, on the sine-skew distribution, quantile-crossing was observed for the DNN model between at least two quantiles on 38% of test $x$ values!

Spline quantile functions (Gasthaus et al., 2019) avoid crossing by construction, but their accuracy was much worse than the DLN accuracy on three of the four datasets, and not better on the fourth. In particular, the DLNs do much better at the Griewank and Ackley simulations, which have the lowest signal-to-noise ratios (Torossian et al., 2020).

Table 3: Simulations: Quantile MSE and percent crossing violations for $\tau \in \{0.01, 0.02, \ldots, 0.99\}$.

| | Sine-skew (1,7) | Griewank | Michalewicz | Ackley |
|---|---|---|---|---|
| Model | MSE, *Crossing* | MSE, *Crossing* | MSE, *Crossing* | MSE, *Crossing* |
| DNN | **3.53 ± 0.09**, *38%* | 1.29 ± 0.02, *5%* | 0.311 ± 0.011, *12%* | 237 ± 6, *0.3%* |
| SQF-DNN | 5.68 ± 0.11, *0%* | 1.05 ± 0.01, *0%* | **0.232 ± 0.006**, *0%* | 667 ± 81, *0%* |
| DLN | **3.51 ± 0.13**, *0%* | **0.55 ± 0.01**, *0%* | **0.219 ± 0.006**, *0%* | **206 ± 2**, *0%* |

## 5.4 Training Time Comparison

We also provide timing results in Table 4 for the Ackley simulation, which as a complex 9-dimensional problem required non-trivial training times. We compared timings of the different models with their validated hyperparameters. The validated DNN chose half the number of training batches as the DLN, and this caused it to be twice as fast. Each training batch took roughly the same amount of time though, and more granular hyperparameter tuning of the number of training batches could have led to different results. The SQF-DNN, meanwhile, is noticeably slower to train. However, while these implementations were comparable in their use of software and processors, all of these approaches could certainly be better optimized for speed. For example, compiler-optimized DLN models can execute ten times faster than an optimized C++ implementation, which was 100 times faster than a TensorFlow implementation (Zhang et al., 2021).

Table 4: Simulations: Training time, in seconds, for the validated model of each type on the Ackley simulation from Table 3. We also report the number of training steps used by each model (i.e. the number of batches seen by the model) and the time taken per training step.

| Model | Train Time (s) | Training Steps | Time Per Batch (ms) |
|---|---|---|---|
| DNN | 3446 | 500000 | 6.9 |
| SQF-DNN | 28042 | 100000 | 280 |
| DLN | 8444 | 1000000 | 8.4 |

### 5.4.1 Real Data

Table 5 compares these models on the four real datasets. We report the pinball loss averaged over $\tau \in \{0.01, 0.02, \ldots, 0.99\}$, which is a standard metric for judging conditional quantile estimates, and one which approximates the *continuous ranked probability score* metric used for measuring the quality of probabilistic forecasts (Gasthaus et al., 2019; Yan et al., 2018). The $\tau$-monotonic DLNs performed the best or statistically similar to the best on three of the four problems, and was statistically significantly better than the DNNs trained with the same expected pinball loss for all four problems. Unlike the simulation results in Table 3, the SQF-DNN (monotonic by construction) performs notably better on these real datasets than the DNN in Table 5, but is never better than the DLN.

We also report the *absolute deviation calibration error* (Chung et al., 2021) in Table 6 (some authors use a squared variant (Kuleshov et al., 2018)), the formula is given in A.8. By that metric the DLN is stat. sig. better in 3 of the 4 cases than SQF-DNN, with substantial wins over SQF-DNN on Traffic and Wine.

## 5.5 Expected Pinball Loss Experiments

We next empirically show that training a full inverse CDF by minimizing the expected pinball loss is at least as good at estimating specific quantiles as training with the pinball loss just for those specific quantiles.

### 5.5.1 Simulations

We ran simulations on the 1D sine-skew function as per Torossian et al. (2020), with three noise scenarios illustrated in Figure 5: noise parameters $(a, b) = (1, 1)$ for symmetric low-noise, $(7, 7)$ for symmetric high-noise, and $(1, 7)$ for asymmetric high-noise. These simulations are the simplest extension of the unconditioned context studied empirically and theoretically in Section 3 to the conditioned case, in that there is a single $x$ feature, and enough data for a flexible quantile regression model to approximate the corresponding sample quantile of $y$ within a small neighborhood of any value of $x$.

The results shown in Table 7 are the error in predicting the median given either $N = 100$ or $N = 1,000$ training samples. Overall, the single pinball loss model struggles relative to the expected pinball loss models, even as we increase the number of data points from 100 to 1000, at which point there are about 50 data points within each 0.1 region of $x \in [-1, 1]$.

Table 5: Real data experiments: Pinball loss on the test set, averaged over $\tau \in \{0.01, 0.02, \ldots, 0.99\}$.

| Model | Air Quality | Puzzles | Traffic | Wine |
|---|---|---|---|---|
| DNN | $18.041 \pm 0.088$ | $3.253 \pm 0.024$ | $0.0494 \pm 0.00037$ | $0.6603 \pm 0.0088$ |
| SQF-DNN | $\mathbf{17.356 \pm 0.106}$ | $\mathbf{3.108 \pm 0.044}$ | $0.0491 \pm 0.00022$ | $0.6552 \pm 0.0037$ |
| DLN | $\mathbf{17.280 \pm 0.157}$ | $\mathbf{3.092 \pm 0.020}$ | $\mathbf{0.0479 \pm 0.00003}$ | $\mathbf{0.6381 \pm 0.0005}$ |

Table 6: Real data experiments: Calibration on the test set, averaged over $\tau \in \{0.01, 0.02, \ldots, 0.99\}$.

| Model | Air Quality | Puzzles | Traffic | Wine |
|---|---|---|---|---|
| DNN | $0.0761 \pm 0.0038$ | $0.0826 \pm 0.0031$ | $0.0562 \pm 0.0014$ | $0.0253 \pm 0.0010$ |
| SQF-DNN | $0.0798 \pm 0.0042$ | $\mathbf{0.0547 \pm 0.0031}$ | $0.0600 \pm 0.0062$ | $0.0671 \pm 0.0083$ |
| DLN | $\mathbf{0.0588 \pm 0.0057}$ | $0.0681 \pm 0.0052$ | $\mathbf{0.0210 \pm 0.0027}$ | $\mathbf{0.0125 \pm 0.0031}$ |

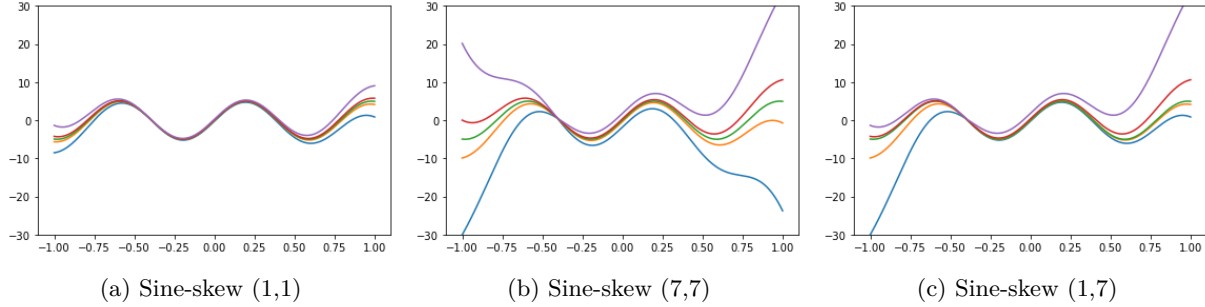

(a) Sine-skew (1,1)  (b) Sine-skew (7,7)  (c) Sine-skew (1,7)

Figure 5: The colored lines show the true quantiles for $\tau = 0.1, 0.4, 0.5, 0.6, 0.9$ for the sine-skew distribution simulations Torossian et al. (2020).

The uniform pinball loss model which trains equally across all quantiles performs quite well in both symmetric-noise cases $(1, 1)$ and $(7, 7)$. However, for the simulation with highly skewed noise $(1, 7)$ it struggles a bit relative to the models trained with a Beta pinball loss, perhaps because the extreme quantiles start becoming very uninformative about the median. Still, even in that case, it performs similarly to the single pinball loss models.

### 5.5.2 Real Data

In Table 8 we give the accuracy at predicting three specific target quantiles on the four real datasets by either training a full inverse CDF by minimizing $E_{\mathcal{T}}$ where $\mathcal{T}$ is drawn uniformly from $(0, 1)$; or training one model to minimize the discrete sum of the three pinball losses for the $\tau$s considered; or training three separate models that minimize $E_{\mathcal{T}}$ where $\mathcal{T}$ is concentrated around the target quantiles; or training three separate models that minimize each specific $\tau$'s pinball loss.

The test metric is computed on finite test sets, so the results are not as clean as the simulations where we can compare to the true quantiles.

The inverse CDF model is statistically significantly tied or better in every case for the median $\tau = 0.5$, which is the quantile we expected the smoothing of the expected pinball loss to be most helpful. We hypothesize that the win in median accuracy for Puzzles is due to the small size of the train set and high randomness of the truth, making that problem most helped by the regularization of $\tau$-smoothing. Even for the more extreme quantiles of $\tau = 0.7$ and $\tau = 0.9$, the full inverse CDF model is statistically significantly better than training for those specific quantiles (though for $\tau = 0.9$, the model trained for three discrete quantiles is even better).

Table 7: Sine-skew experiment with different sine-skew noise choices (1,1), (7,7) and (1,7). We present results for DLNs trained with four different distributions over $\tau$: the uniform distribution, the beta distribution with a mode of 0.5 and concentration of 10, the beta distribution with a mode of 0.5 and concentration of 100, and just the 0.5 pinball loss. The metric is the MSE between the true median and estimated median, averaged over $x \sim \text{Unif}(-1, 1)$.

| $(a, b)$ | $N$ | min $E_{\mathcal{T}}$, Unif | min $E_{\mathcal{T}}$, Beta(10) | $E_{\mathcal{T}}$, Beta(100) | min single-$\tau$ loss |
|---|---|---|---|---|---|
| $(1, 1)$ | 100 | $0.421 \pm 0.025$ | $0.348 \pm 0.015$ | $\mathbf{0.301 \pm 0.016}$ | $0.588 \pm 0.036$ |
| $(1, 1)$ | 1,000 | $\mathbf{0.050 \pm 0.003}$ | $0.057 \pm 0.002$ | $0.056 \pm 0.002$ | $0.067 \pm 0.003$ |
| $(7, 7)$ | 100 | $\mathbf{7.334 \pm 0.367}$ | $6.654 \pm 0.330$ | $7.402 \pm 0.397$ | $9.860 \pm 0.560$ |
| $(7, 7)$ | 1,000 | $\mathbf{0.967 \pm 0.058}$ | $\mathbf{0.861 \pm 0.036}$ | $1.057 \pm 0.037$ | $1.470 \pm 0.086$ |
| $(1, 7)$ | 100 | $4.458 \pm 0.259$ | $\mathbf{3.515 \pm 0.178}$ | $\mathbf{3.031 \pm 0.179}$ | $4.416 \pm 0.386$ |
| $(1, 7)$ | 1,000 | $0.451 \pm 0.037$ | $\mathbf{0.336 \pm 0.022}$ | $\mathbf{0.338 \pm 0.020}$ | $0.542 \pm 0.067$ |

Table 8: Accuracy of single quantile estimates, comaparing one model trained to minimize the expected pinball loss with respect to uniform $P_{\mathcal{T}}$; one model trained to minimize the sum of the three pinball losses for the three quantiles shown (Discrete); three separate models trained with individual beta distributions $P_{\mathcal{T}}$ with the mode set at the target quantile and the beta concentration chosen on the validation set; and three separate models trained with individual $\tau$ pinball losses (Single). All models were DLNs with the same architecture as in Table 5, and monotonic on $\tau$ and any input features noted in the text.

| Air Quality: | Model | Pinball loss ($\tau = 0.5$) | Pinball loss ($\tau = 0.9$) | Pinball loss ($\tau = 0.99$) |
|---|---|---|---|---|
| | $\tau \sim \text{Unif}(0, 1)$ | $\mathbf{23.576 \pm 0.047}$ | $14.850 \pm 0.075$ | $2.947 \pm 0.025$ |
| | $\tau \sim \text{Discrete}$ | $24.083 \pm 0.073$ | $15.439 \pm 0.062$ | $3.042 \pm 0.012$ |
| | $\tau \sim \text{Beta}$ | $\mathbf{23.586 \pm 0.183}$ | $\mathbf{13.942 \pm 0.054}$ | $\mathbf{2.709 \pm 0.029}$ |
| | Single $\tau \in \mathbf{T}$ | $\mathbf{23.634 \pm 0.068}$ | $14.908 \pm 0.092$ | $\mathbf{2.700 \pm 0.013}$ |
| **Traffic:** | Model | Pinball loss ($\tau = 0.5$) | Pinball loss ($\tau = 0.9$) | Pinball loss ($\tau = 0.99$) |
| | $\tau \sim \text{Unif}(0, 1)$ | $\mathbf{0.064386 \pm 0.00014}$ | $0.040159 \pm 0.00018$ | $\mathbf{0.010645 \pm 0.00018}$ |
| | $\tau \sim \text{Discrete}$ | $0.064578 \pm 0.00028$ | $\mathbf{0.039804 \pm 0.00014}$ | $0.011290 \pm 0.00014$ |
| | $\tau \sim \text{Beta}$ | $0.064988 \pm 0.00030$ | $\mathbf{0.039549 \pm 0.00009}$ | $\mathbf{0.01064 \pm 0.00011}$ |
| | Single $\tau \in \mathbf{T}$ | $\mathbf{0.064791 \pm 0.00012}$ | $0.039900 \pm 0.00010$ | $\mathbf{0.01070 \pm 0.00018}$ |
| **Wine:** | Model | Pinball loss ($\tau = 0.1$) | Pinball loss ($\tau = 0.5$) | Pinball loss ($\tau = 0.9$) |
| | $\tau \sim \text{Unif}(0, 1)$ | $0.4099 \pm 0.0006$ | $\mathbf{0.8889 \pm 0.0017}$ | $0.3773 \pm 0.0019$ |
| | $\tau \sim \text{Discrete}$ | $0.4094 \pm 0.0005$ | $0.8891 \pm 0.0010$ | $0.3706 \pm 0.0003$ |
| | $\tau \sim \text{Beta}$ | $\mathbf{0.4047 \pm 0.0005}$ | $\mathbf{0.8871 \pm 0.0008}$ | $\mathbf{0.3665 \pm 0.0004}$ |
| | Single $\tau \in \mathbf{T}$ | $\mathbf{0.4049 \pm 0.0004}$ | $\mathbf{0.8867 \pm 0.0006}$ | $\mathbf{0.3663 \pm 0.0003}$ |
| **Puzzles:** | Model | Pinball loss ($\tau = 0.5$) | Pinball loss ($\tau = 0.7$) | Pinball loss ($\tau = 0.9$) |
| | $\tau \sim \text{Unif}(0, 1)$ | $\mathbf{4.173 \pm 0.013}$ | $\mathbf{4.219 \pm 0.021}$ | $2.705 \pm 0.029$ |
| | $\tau \sim \text{Discrete}$ | $4.359 \pm 0.014$ | $\mathbf{4.204 \pm 0.011}$ | $\mathbf{2.614 \pm 0.010}$ |
| | $\tau \sim \text{Beta}$ | $4.318 \pm 0.013$ | $4.277 \pm 0.018$ | $2.681 \pm 0.011$ |
| | Single $\tau \in \mathbf{T}$ | $4.293 \pm 0.013$ | $4.350 \pm 0.022$ | $2.708 \pm 0.012$ |

We expected the $E_{\mathcal{T}}$ loss to be least useful for extreme quantiles, since the smoothing will be more one-sided. Indeed, the wins for the single-quantile models are on more extreme quantiles: the 99th percentile for Air Quality, and the 10th and 90th percentile for Wine. For Wine, we hypothesize this is because it is a fairly easy regression problem with a large number of training samples and one continuous feature (wine price) that correlates highly with the label (wine quality), so the extra regularization is not helping. But for the large Traffic dataset, even the 99th percentile is statistically tied, and the single quantile model is worse even at the extreme quantiles for Puzzles. Interestingly, between the two single-quantile modeling approaches, training with $\tau$ sampled from a Beta distribution is statistically similar or better than training with a single pinball loss in every case. Overall, the results in Table 8 show that the full inverse CDFs are at least as good as the single-quantile models.

## 6 Conclusions And Some Open Questions

We gave theoretical and empirical evidence that training quantile regression models by minimizing the expected pinball loss can perform as well or better on any individual quantile than models trained for that quantile alone. We showed that minimizing the expected pinball loss is not sufficient to provide legitimate inverse CDF estimates, one must also produce a model whose quantiles are guaranteed to not cross. We showed that DLN models can be effectively constrained to be monotonic in $\tau$ and produce state-of-the-art quantile estimates. The non-crossing issue is important not just for theoretical reasons, but because it is confusing to many non-experts if a model's quantile estimates cross, eroding trust in that model, and by association, all AI.

We provide a new asymptotic convergence rate for estimation of the full inverse CDF, assuming an oracle that minimizes the empirical loss. We leave open the question of a non-asymptotic convergence rate, and whether some of our assumptions can be relaxed, or whether better convergence rates can be had with some more stringent assumptions. Theorem 1 is for unconditioned quantile estimation, with a simple extension to separable conditional quantile estimation in Corollary 1. Important open work would be providing stronger results for conditional quantile estimation where there is smoothing across the features.

A key question is the choice of distribution of $\mathcal{T}$ when minimizing the expected pinball loss. We note one can control the regularization across quantiles both by the shape of $P_{\mathcal{T}}$ and by the model flexibility with respect to $\tau$. Thus we showed a uniform distribution $P_{\mathcal{T}}$ generally provides good results if one is validating the model flexibility over $\tau$, and a uniform distribution has no hyperparameters and gives you a good estimate of the entire inverse CDF for free. However, if one is only interested in a single $\tau$, we also showed that one may do better with a beta distribution for $P_{\mathcal{T}}$, and that a concentration of 10 likely makes a good default choice, but for best results one would want to validate the concentration.

One open question is whether the proposed methodology improves the accuracy of prediction intervals (Pearce et al., 2018), as measured by traditional metrics such as interval width and calibration. Tagasovska & Lopez-Paz (2019) have shown that minimizing the expected pinball loss can be a useful strategy in that regard, and our results in this paper about improved quantile accuracy and calibration suggest there may be improved performance in prediction intervals as well.

Our experiments, like most published quantile regression experiments, were limited to datasets with $D = 42$ features and less. This is partly because one needs a large amount of data to make decent estimates of tail quantiles. In theory the proposed methodology can be applied to any size problem by only bringing $\tau$ in for the last layers as shown in Fig. 4, but an open question is how well this or any quantile regression works in practice with a large number of features, and whether there are novel scaling issues.

In particular, we hypothesize that for larger models there may be an increase in *quantile collapse* (Lopez-Martin et al., 2021). Quantile collapse is the edge case of non-crossing, defined as a flat $f(\tau)$ over some range of $\tau$. For example, the airline tells passengers there is an 80% chance the flight will arrive in 4 hours and 6 minutes, and that there is a 90% chance the flight will arrive in 4 hours and 6 minutes. Our monotonicity constraint (2) allows quantile collapse because it only requires monotonicity in $\tau$, not *strict* monotonicity.

Quantile collapse can happen in regions of sparse data where there are no training samples to fit to such that the model only has to satisfy (2), which it can do with a flat $f(x, \tau)$. We belive this can partly be solved

by initializing with a $f(x, \tau)$ that is strictly monotonically increasing (as we do in our experiments), as then one only gets quantile collapse if there is enough relevant training data to pull the model to the edge of the feasibility of (2).

Quantile collapse is also a risk if the model $f(x, \tau)$ has too much flexibility in $x$ such that the data is sparse with respect to that flexibility. For example, assume $x$ is a categorical feature with 1 million possible values (e.g. a language model), and that $f(x, \tau)$ is flexible enough to learn an independent estimate for each value of $x$. Suppose for one of those categorical values there is only one training sample $(x_i, y_i)$. Then since the pinball loss (expected or standard) will be minimized when $f(x, \tau) = y_i$, there is no $\tau$ dependence, and for that category you will have quantile collapse. More generally, if you have a very flexible model and sparse data in some regions, we expect you may get quantile collapse if no other regularization to avoid it is used.

### Broader Impact Statement

We hope the broader impact of this work to be positive in increasing the deserved trustworthiness of AI by promoting quantile regression models with no crossing-quantiles. The alternative of quantile estimates that cross will undoubtedly erode public confidence in AI. For example, prior work would launch a quantile regression model that predicts there is a 90 percent chance your pizza will be delivered in 30 minutes, and that there is an 80 percent chance it will be delivered in 34 minutes. Such models are unnecessarily confusing and untrustworthy for non-experts, and using them in practice may then reduce the ability of worthy AI to have a positive impact on the world.

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

# A  Appendix

## A.1  Proof of Lemma 2

**Lemma 2.** *Let $f(x, \tau)$ be a calibrated lattice model with piece-wise linear calibrator $c(\tau) : [0, 1] \to [0, 1]$ for $\tau$, and let there be two knots in the regular grid in the direction of $\tau$, and the lattice parameters are interpolated with standard multilinear interpolation. Then $f(x, \tau)$ represents an inverse CDF function $F^{-1}(y|x)$ where the estimated distribution for every $x$ is from the same location-scale family as the calibrator $c(\tau)$.*

*Proof.* If a random variable $Y$ conditioned on $X$ belongs to the location-scale family then for $\tau \in (0, 1)$ and $a \in \mathbb{R}$ and $b > 0$, it must hold that the conditional inverse CDF satisfies $F^{-1}_{Y|X=z}(\tau) = a + bF^{-1}_{Y|X=x}(\tau)$. Note that for $\tau \in (0, 1)$, interpolating a lattice with two lattice vertices in the $\tau$ dimension yields the estimate $\hat{F}^{-1}_{Y|X=z}(\tau) = f(z, \tau) = f(z, 0) + c(\tau)(f(z, 1) - f(z, 0))$. Thus mapping to the location-scale property, $a = f(z, 0)$, $b = f(z, 1) - f(z, 0)$, and $\hat{F}^{-1}_{Y|X=x}(\tau) = c(\tau)$. Thus every estimated conditional inverse CDF $\hat{F}^{-1}_{Y|X=z}(\tau)$ is a translation and scaling of the piecewise linear function $c(\tau)$. $\square$

## A.2  Proof of Theorem 1

**Theorem 1.** *Suppose $f(\tau; \theta)$ is well specified with a unique $\theta^* \in \Theta \subseteq \mathbb{R}^m$ such that $f(\tau; \theta^*) = F^{-1}(\tau)$ for all $\tau \in (0, 1)$, and let $\mathcal{T}$ be distributed such that $\theta^*$ is also a unique minimizer for the risk $R(\theta) = E_{Y, \mathcal{T}}[L_{\mathcal{T}}(Y, f(\mathcal{T}; \theta))]$. Suppose the estimator $\hat{\theta}^{(n)}$ is weakly consistent, $\hat{\theta}^{(n)} \xrightarrow{P} \theta^*$. Suppose further that the function $\theta \mapsto f(\tau; \theta)$ is continuous, locally Lipschitz, and twice differentiable at $\theta = \theta^*$ for all $\tau \in (0, 1)$. Then,*

$$\sqrt{n}(f(\tau; \hat{\theta}^{(n)}) - f(\tau; \theta^*)) \xrightarrow{d} \mathcal{N}\left(0, \nabla_\theta f(\tau; \theta^*)^\top Q^{-1} V (Q^{-1})^\top \nabla_\theta f(\tau; \theta^*)\right)$$

*where,*

$$Q = E_{\mathcal{T}}[p(f(\mathcal{T}; \theta^*))\Gamma(\mathcal{T})], V = E_{\mathcal{T}}[\mathcal{T}(1 - \mathcal{T})\Gamma(\mathcal{T})],$$

*with $\Gamma(\tau) = \nabla_\theta f(\tau; \theta^*)\nabla_\theta f(\tau; \theta^*)^\top$.*

*Proof.* Consider the loss function $\bar{L}(y, \theta) = E_{\mathcal{T}}[L_{\mathcal{T}}(y, f(\mathcal{T}; \theta))]$. Let $\hat{\theta}^{(n)}$ minimize the risk with an empirical distribution over $Y$ with IID samples $\{y_i\}_{i=1}^n$, $\hat{\theta}^{(n)} = \arg\min_\theta \frac{1}{n} \sum_{i=1}^n \bar{L}(y_i, \theta)$. We assume that $Y$ is a continuous random variable with density $p(y)$, and $\mathcal{T}$ is independent of $Y$.

We apply Theorem 5.23 from van der Vaart (1998)[1], which says that under some regularity conditions,

$$\sqrt{n}(\hat{\theta}^{(n)} - \theta^*) = -\nabla^2 R(\theta^*)^{-1} \cdot \frac{1}{\sqrt{n}} \sum_{i=1}^n \nabla_\theta \bar{L}(y_i, \theta^*) + o_P(1),$$

where $o_P(1)$ refers to a sequence of random variables that converges in probability to 0: $X_n = o_P(1)$ if and only if $X_n \xrightarrow{P} 0$. Specifically, Theorem 5.23 from van der Vaart (1998) requires the following conditions:

(i) $\hat{\theta}^{(n)}$ is weakly consistent for $\theta^*$: $\hat{\theta}^{(n)} \xrightarrow{P} \theta^*$. This is assumed, but in general this consistency depends on properties of the function $f$ and the feasible parameter space $\Theta$. See Lemma 1 for an example.

(ii) $\frac{1}{n} \sum_{i=1}^n \bar{L}(y_i, \hat{\theta}^{(n)}) \le \inf_\theta \frac{1}{n} \sum_{i=1}^n \bar{L}(y_i, \theta) + o_P(1/n)$: true by optimality of $\hat{\theta}^{(n)}$.

(iii) The loss $\bar{L}(y, \theta)$ is locally Lipschitz near $\theta = \theta^*$. This holds as long as $f(\tau; \theta)$ is locally Lipschitz near $\theta = \theta^*$.

(iv) For $P$-almost all $y$, the function $\theta \mapsto \bar{L}(y, \theta)$ is differentiable at $\theta = \theta^*$.

(v) $R(\theta)$ is twice differentiable at $\theta^*$ with positive definite Hessian $\nabla^2 R(\theta^*) \succ 0$.

To verify condition (iv), note that

$$\nabla_\theta \bar{L}(y, \theta^*) = \nabla_\theta E_{\mathcal{T}}[L_{\mathcal{T}}(y, f(\mathcal{T}; \theta^*)] = E_{\mathcal{T}}[\nabla_\theta L_{\mathcal{T}}(y, f(\mathcal{T}; \theta^*))].$$

Since $Y$ is a continuous random variable with a density near $f(\tau; \theta^*)$, $\nabla_\theta L_\tau(y, f(\tau, \theta^*))$ exists for $P$-almost every $y$ as long as $f(\tau, \theta)$ is also differentiable at $\theta = \theta^*$. This is because for continuous $Y$, the $\tau$-quantile is unique, and the derivative $\nabla_\gamma L_\tau(y, \gamma)$ exists with probability 1. The full gradient is:

$$\nabla_\theta L_\tau(y, f(\tau; \theta^*)) = ((1 - \tau)\mathbb{1}(f(\tau; \theta^*) \ge y) - \tau\mathbb{1}(f(\tau; \theta^*) \le y)) \nabla_\theta f(\tau; \theta^*).$$

Therefore, $\nabla_\theta \bar{L}(y, \theta^*) = E_{\mathcal{T}}[\nabla_\theta L_{\mathcal{T}}(y, f(\mathcal{T}; \theta^*))]$ is also well defined and exists for $P$-almost every $y$.

To verify condition (v), we explicitly compute the Hessian:

$$\begin{aligned}
\nabla R(\theta^*) &= E_Y[\nabla_\theta \bar{L}(Y, \theta^*)] \\
&= E_Y[E_{\mathcal{T}}[\nabla_\theta L_{\mathcal{T}}(Y, f(\mathcal{T}; \theta^*))]] \\
&= E_{\mathcal{T}}[E_Y[\nabla_\theta L_{\mathcal{T}}(Y, f(\mathcal{T}; \theta^*))]] \quad \text{since } \mathcal{T} \perp\!\!\!\perp Y \\
&= E_{\mathcal{T}}[((1 - \mathcal{T})P(f(\mathcal{T}; \theta^*) \ge Y) - \mathcal{T}P(f(\mathcal{T}; \theta^*) \le Y)) \nabla_\theta f(\mathcal{T}; \theta^*)] \\
&= E_{\mathcal{T}}[(P(Y \le f(\mathcal{T}; \theta^*)) - \mathcal{T})\nabla_\theta f(\mathcal{T}; \theta^*)] \\
&= E_{\mathcal{T}}[(F(f(\mathcal{T}; \theta^*)) - \mathcal{T})\nabla_\theta f(\mathcal{T}; \theta^*)].
\end{aligned}$$

$$\begin{aligned}
\nabla^2 R(\theta^*) &= E_{\mathcal{T}}[\nabla_\theta \{(F(f(\mathcal{T}; \theta^*)) - \mathcal{T})\nabla_\theta f(\mathcal{T}; \theta^*)\}] \\
&= E_{\mathcal{T}}[\nabla_\theta f(\mathcal{T}; \theta^*)F'(f(\mathcal{T}; \theta^*))\nabla_\theta f(\mathcal{T}; \theta^*)^\top + (F(f(\mathcal{T}; \theta^*)) - \mathcal{T})\nabla_\theta^2 f(\mathcal{T}; \theta^*)] \\
&= E_{\mathcal{T}}[p(f(\mathcal{T}; \theta^*))\nabla_\theta f(\mathcal{T}; \theta^*)\nabla_\theta f(\mathcal{T}; \theta^*)^\top] \\
&= Q \succ 0.
\end{aligned}$$

The third equality in the computation of the Hessian above follows by assumption that $f$ is well specified, since $f(\tau; \theta^*) = F^{-1}(\tau)$, so $F(f(\tau; \theta^*)) - \tau = 0$.

[1]A. W. van der Vaart. *Asymptotic Statistics*. Cambridge Series in Statistical and Probabilistic Mathematics. Cambridge University Press, 1998.

With all conditions verified, applying Theorem 5.23 from van der Vaart (1998) we have,

$$\sqrt{n}(\hat{\theta}^{(n)} - \theta^*) = -Q^{-1} \cdot \frac{1}{\sqrt{n}} \sum_{i=1}^{n} E_{\mathcal{T}}[\nabla_\theta L_{\mathcal{T}}(y_i, f(\mathcal{T}; \theta^*))] + o_P(1). \tag{5}$$

Focusing on the sum term,

$$\frac{1}{\sqrt{n}} \sum_{i=1}^{n} E_{\mathcal{T}}[\nabla_\theta L_{\mathcal{T}}(y_i, f(\mathcal{T}; \theta^*))]$$

$$= \frac{1}{\sqrt{n}} \sum_{i=1}^{n} E_{\mathcal{T}}\left[((1 - \mathcal{T})\mathbb{1}(f(\mathcal{T}; \theta^*) \geq y_i) - \mathcal{T}\mathbb{1}(f(\mathcal{T}; \theta^*) \leq y_i)) \nabla_\theta f(\mathcal{T}; \theta^*)\right]$$

$$= \frac{1}{\sqrt{n}} \sum_{i=1}^{n} E_{\mathcal{T}}\left[(\mathbb{1}(y_i \leq f(\mathcal{T}; \theta^*)) - \mathcal{T})\nabla_\theta f(\mathcal{T}; \theta^*)\right]$$

Since $\mathbb{1}(y_i \leq f(\mathcal{T}; \theta^*))$ are IID samples from Bernoulli$(\mathcal{T})$ and $\mathcal{T}$ is independent of $Y$, the random variable $E_{\mathcal{T}}\left[(\mathbb{1}(Y \leq f(\mathcal{T}; \theta^*)) - \mathcal{T})\nabla_\theta f(\mathcal{T}; \theta^*)\right]$ has mean $\mathbf{0}$ and variance

$$V = E_{\mathcal{T}}[\mathcal{T}(1 - \mathcal{T})\nabla_\theta f(\mathcal{T}; \theta^*)\nabla_\theta f(\mathcal{T}; \theta^*)^\top].$$

By the central limit theorem, the sum converges in distribution to $\mathcal{N}(\mathbf{0}, V)$. Combining this with Equation (5) yields,

$$\sqrt{n}(\hat{\theta}^{(n)} - \theta^*) \xrightarrow{d} \mathcal{N}(\mathbf{0}, Q^{-1}V(Q^{-1})^\top).$$

Applying the delta method with $f(\tau; \theta)$ as a function of $\theta$ for a fixed desired quantile $\tau$, we have the final result,

$$\sqrt{n}(f(\tau; \hat{\theta}^{(n)}) - f(\tau; \theta^*)) \xrightarrow{d} \mathcal{N}\left(\mathbf{0}, \nabla_\theta f(\tau; \theta^*)^\top Q^{-1}V(Q^{-1})^\top \nabla_\theta f(\tau; \theta^*)\right).$$

$\square$

### A.3 Sufficient conditions for consistency

**Lemma 1.** *If $f(\tau, \theta)$ is convex in $\theta$ for all $\tau \in (0, 1])$, $\Theta$ is convex, and $\theta^*$ is in the interior of $\Theta$, then $\hat{\theta}^{(n)}$ is weakly consistent, $\hat{\theta}^{(n)} \xrightarrow{P} \theta^*$.*

*Proof.* The result follows from Proposition 7.4 from Hayashi (2011), provided we show that $\bar{L}(y, \theta)$ is also convex.

$$\bar{L}(y, \theta) = E_{\mathcal{T}}[L_{\mathcal{T}}(y, f(\mathcal{T}; \theta))]$$
$$= E_{\mathcal{T}}[\max(\mathcal{T}(y - f(\mathcal{T}; \theta)), (\mathcal{T} - 1)(y - f(\mathcal{T}; \theta)))]$$

For any fixed scalar value $\tau$, the term $\tau(y - f(\tau; \theta))$ is convex in $\theta$ since it is a linear mapping of $f(\tau; \theta)$. The same applies to the term $(\tau - 1)(y - f(\tau; \theta))$. Therefore, $\max(\tau(y - f(\tau; \theta)), (\tau - 1)(y - f(\tau; \theta)))$, as the pointwise maximum of two convex functions, is also convex in $\theta$. Finally, the linearity of the $E_{\mathcal{T}}[\cdot]$ operator gives that $\bar{L}(y, \theta)$ is convex in $\theta$. $\square$

### A.4 A note on uniqueness of $\theta^*$

Theorem 1 and Lemma 1 assume that $f(\tau; \theta)$ is well specified with a unique $\theta^* \in \Theta \subseteq \mathbb{R}^m$ such that $f(\tau; \theta^*) = F^{-1}(\tau)$ for all $\tau \in (0, 1)$, and that $\theta^*$ is also the unique minimizer for the risk $R(\theta) = E_{Y, \mathcal{T}}[L_{\mathcal{T}}(Y, f(\mathcal{T}; \theta))]$. The property that $f(\tau; \theta^*) = F^{-1}(\tau)$ directly implies that $\theta^*$ is a minimizer of $R(\theta)$; however, whether $\theta^*$ is the *unique* minimizer of $R(\theta)$ depends on the distribution of $\mathcal{T}$ and function class defined by $f$ and $\Theta$.

**Lemma 3.** Suppose $f(\tau; \theta)$ is well specified with a unique $\theta^* \in \Theta \subseteq \mathbb{R}^m$ such that $f(\tau; \theta^*) = F^{-1}(\tau)$ for all $\tau \in (0, 1)$. For $\theta \in \Theta$, define the set $T_\theta := \{\tau : f(\tau; \theta) \neq f(\tau; \theta^*)\}$. If for all $\theta \in \Theta$ where $\theta \neq \theta^*$, the set $T_\theta$ has measure greater than 0 under random variable $\mathcal{T}$, then $\theta^*$ is also the unique minimizer for the risk $R(\theta) = E_{Y, \mathcal{T}}[L_\mathcal{T}(Y, f(\mathcal{T}; \theta))]$.

*Proof.* Suppose there exists $\theta'$ where $\theta' \neq \theta^*$ but $\theta'$ also minimizes of the risk $R(\theta)$. Since $\theta' \neq \theta^*$, the set $T_{\theta_0}$ is nonempty, and for all $\tau \in T_{\theta'}$,

$$f(\tau; \theta') \neq F^{-1}(\tau) \implies \theta' \neq \arg\min_\theta E_Y[L_\tau(Y, f(\tau; \theta^*))]$$

$$\implies E_Y[L_\tau(Y, f(\tau; \theta'))] > E_Y[L_\tau(Y, f(\tau; \theta^*))].$$

Then since $T_{\theta'}$ has measure greater than 0 under random variable $\mathcal{T}$, this implies

$$E_\mathcal{T}[E_Y[L_\mathcal{T}(Y, f(\mathcal{T}; \theta'))]] > E_\mathcal{T}[E_Y[L_\mathcal{T}(Y, f(\mathcal{T}; \theta^*))]],$$

which contradicts that $\theta'$ minimizes the risk $R(\theta)$. $\square$

The conditions of uniqueness can be satisfied by a combination of wide enough support of $\mathcal{T}$, and a limited enough function class given by $f, \Theta$. For example, if $\mathcal{T}$ is supported on the full interval $(0, 1)$, and $f(\tau; \theta) = \sum_{j=1}^m \theta_j \tau^j$ is a polynomial function over $\Theta \subseteq \mathbb{R}^m$, then any pair $\theta \neq \theta'$ would yield a set $\{\tau : f(\tau; \theta) \neq f(\tau; \theta')\}$ with measure greater than 0.

Uniqueness breaks down of the function class is too expressive. For example, if $f(\tau; \theta)$ is infinitely expressive over $\Theta$ (e.g. $\Theta = \mathbb{R} \times (0, 1)$, and $f(\tau; \theta) = \theta_0 \mathbb{1}(\theta_1 = \tau)$), then it's possible to find a pair $\theta \neq \theta'$ that yield a set $\{\tau : f(\tau; \theta) \neq f(\tau; \theta')\}$ that contains only a single value of $\tau$ with measure 0.

Uniqueness can also break down if the support of $\mathcal{T}$ is too limited. For example, if $\mathcal{T}$ is only supported on a single value $\tau_0$, and $f(\tau; \theta) = \sum_{j=1}^m \theta_j \tau^j$ is a polynomial function, then it's possible to find a pair $\theta \neq \theta'$ where $f(\tau_0; \theta) = f(\tau_0; \theta')$, and thus $\{\tau : f(\tau; \theta) \neq f(\tau; \theta')\}$ has measure 0 for $\mathcal{T}$.

### A.5 Basic extension to categorical features

**Corollary 1.** *Let $X$ be distributed uniformly over a finite set of categorical values $\mathcal{X}$. For notational convenience, let $\mathcal{X} = \{1, ..., m\} \subset \mathbb{N}$. Suppose that a dataset $\{(x_i, y_i)\}_{i=1}^n$ is created by sampling $\{y_j\}_{j=1}^{\frac{n}{m}}$ values IID from the conditional distribution $Y|X = x$ for each value of $x$ (assuming $m$ divides $n$), and taking the union of these sets.*

*Let $f : \mathcal{X}, (0, 1), \Theta \to \mathbb{R}$ be fully separable over $x$; that is, let $\Theta = \Theta_1 \times ... \times \Theta_m$, where $\Theta_x$ represents a copy of a given parameter space $\Theta_0$ for each value of $x \in \mathcal{X}$. Let $f$ take the form*

$$f(x, \tau; \theta) = \sum_{x_0 \in \mathcal{X}} \mathbb{1}(x = x_0) g(\tau; \theta_{x_0})$$

*for some function $g : (0, 1), \Theta_0 \to \mathbb{R}$.*

*Suppose $f(x, \tau; \theta)$ is well specified with a unique $\theta^* \in \Theta \subseteq \mathbb{R}^m$ such that $f(x, \tau; \theta^*) = q_\tau(x)$ for all $\tau \in (0, 1)$ and all $x \in \mathcal{X}$, and that $\theta^*$ is also a unique minimizer for the risk $R(\theta) = \frac{1}{m} \sum_{x \in \mathcal{X}} E_{Y, \mathcal{T}}[L_\mathcal{T}(Y, f(x, \mathcal{T}; \theta))|X = x]$. Suppose the estimator $\hat{\theta}^{(n)}$ is weakly consistent, $\hat{\theta}^{(n)} \xrightarrow{P} \theta^*$. Suppose further that the function $\theta \mapsto f(x, \tau; \theta)$ is continuous, locally Lipschitz, and twice differentiable at $\theta = \theta^*$ for all $\tau \in (0, 1)$ and all $x \in \mathcal{X}$.*

*Then for each $x \in \mathcal{X}$,*

$$\sqrt{\frac{n}{m}}(f(x, \tau; \hat{\theta}^{(n)}) - f(x, \tau; \theta^*)) \xrightarrow{d} \mathcal{N}\left(0, \nabla_\theta f(x, \tau; \theta^*)^\top Q^{-1} V (Q^{-1})^\top \nabla_\theta f(x, \tau; \theta^*)\right)$$

*where,*

$$Q = E_\mathcal{T}[p_x(f(x, \mathcal{T}; \theta^*))\Gamma(\mathcal{T})], V = E_\mathcal{T}[\mathcal{T}(1 - \mathcal{T})\Gamma(\mathcal{T})],$$

*with $p_x(y)$ being the density of $Y|X = x$ and $\Gamma(\tau) = \nabla_\theta f(x, \tau; \theta^*) \nabla_\theta f(x, \tau; \theta^*)^\top$.*

*Proof.* For notational convenience, let $\mathcal{X} = \{1, ..., m\} \subset \mathbb{N}$. Define loss function $\bar{L}(x, y, \theta) = E_{\mathcal{T}}[L_{\mathcal{T}}(y, f(x, \mathcal{T}; \theta))]$. Let $\hat{\theta}^{(n)}$ minimize the empirical risk over samples $\{x_i, y_i\}_{i=1}^n$, $\hat{\theta}^{(n)} = \arg\min_\theta \frac{1}{n} \sum_{i=1}^n \bar{L}(y_i, x_i, \theta)$. By assumption of the separability of $f$ and $\Theta$ by $x$,

$$\hat{\theta}^{(n)} \in \arg\min_{\theta \in \Theta} \frac{1}{n} \sum_{i=1}^n \bar{L}(y_i, x_i, \theta)$$

$$= \arg\min_{\theta \in \Theta} \sum_{x \in \mathcal{X}} \frac{1}{m} \left( \frac{1}{n/m} \sum_{i=1}^n \bar{L}(y_i, x, \theta) \mathbb{1}(x_i = x) \right)$$

$$= \arg\min_{\theta \in \Theta} \sum_{x \in \mathcal{X}} \frac{1}{m} \left( \frac{1}{n/m} \sum_{i=1}^n E_{\mathcal{T}}[L_{\mathcal{T}}(y_i, f(x, \mathcal{T}; \theta))] \mathbb{1}(x_i = x) \right)$$

$$= \arg\min_{\theta \in \Theta} \sum_{x \in \mathcal{X}} \frac{1}{m} \left( \frac{1}{n/m} \sum_{i=1}^n E_{\mathcal{T}}[L_{\mathcal{T}}(y_i, g(\mathcal{T}; \theta_x))] \mathbb{1}(x_i = x) \right)$$

$$= \sum_{x \in \mathcal{X}} \frac{1}{m} \left( \arg\min_{\theta_x \in \Theta_x} \frac{1}{n/m} \sum_{i=1}^n E_{\mathcal{T}}[L_{\mathcal{T}}(y_i, g(\mathcal{T}; \theta_x))] \mathbb{1}(x_i = x) \right)$$

Therefore, each component $\hat{\theta}_x^{(n)}$ minimizes an empirical loss over points with $x_i = x$:

$$\hat{\theta}_x^{(n)} \in \arg\min_{\theta_x \in \Theta_x} \frac{1}{n/m} \sum_{i=1}^n E_{\mathcal{T}}[L_{\mathcal{T}}(y_i, g(\mathcal{T}; \theta_x))] \mathbb{1}(x_i = x).$$

By assumption, components of the optimum $\theta_x^*$ also uniquely minimize each conditional risk

$$\theta_x^* = \arg\min_{\theta_x \in \Theta_x} E_{Y, \mathcal{T}}[L_{\mathcal{T}}(Y, f(x, \mathcal{T}; \theta)) | X = x].$$

Therefore, we may apply Theorem 1 to describe the convergence of each component $\hat{\theta}_x^{(n)}$ to its respective optimum $\theta_x^*$ for each $x \in \mathcal{X}$, yielding the result. $\qquad\square$

### A.6 Additional Examples Illustrating Theorem 1

**Single parameter uniform anchored at 0.** Suppose the true distribution of interest is $Y \sim \text{Unif}(0, \alpha)$. Let $f(\tau; \theta) = \theta\tau$, which correctly parameterizes the true linear inverse CDF. By Theorem 1, the asymptotic variance of $\sqrt{n}(\hat{\theta}^{(n)} - \theta^*)$ is

$$Q^{-1} V (Q^{-1})^\top = \alpha^2 (E_{\mathcal{T}}[\mathcal{T}^3] - E_{\mathcal{T}}[\mathcal{T}^4]) / E_{\mathcal{T}}[\mathcal{T}^2]^2.$$

Thus, the asymptotic variance depends on the distribution of $\mathcal{T}$ through its third and fourth moments, and depends on the distribution of $Y$ through $\alpha$. For $\mathcal{T} \sim \text{Unif}(0, 1)$, the exact asymptotic variance is $0.45\alpha^2$. To estimate a specific $\tau_0$-quantile, we would evaluate the function $f(\tau_0; \hat{\theta}^{(n)}) = \hat{\theta}^{(n)} \tau_0$, which would have an asymptotic variance of $0.45\alpha^2\tau_0^2$.

### A.7 A Connection To Regression On Sample Quantiles

We provide some more intuition here about how the training procedure we describe in this paper - learning a function $f(\tau)$ with an expected pinball loss - smooths sample quantiles in the case that there are no conditioning $x$-features. In particular, we can show that our proposed method (except with a uniform discrete rather than uniform continuous distribution for $\tau$) is equivalent to a regression on the observed sample quantiles with a particular cost function.

**Proposition 1.** *Given a dataset $\{y_i\}_{i=1}^n$ of $n$ elements, construct the dataset $\{\tau(i), y^{(i)}\}_{i=1}^n$ where $y^{(i)}$ represents the ith order statistic of the data and $\tau(i) = \frac{i-1}{n-1}$ is the sample quantile that order statistic represents.*

*Then training a model with an expected pinball loss ([1]) where $\tau$ is drawn uniformly from the discrete distribution $\{\tau(i)\}_{i=1}^{n}$ is equivalent to learning a single-variable regression over the data points $\{\tau(i), y^{(i)}\}_{i=1}^{n}$. The cost function $c$ for the regression at the specific point $(\tau(i), y^{(i)})$ is 0 when $f(\tau(i)) = y^{(i)}$ and otherwise piecewise-linear with slope*

$$c'(f(\tau(i))) = \begin{cases} -\tau(i) - \sum_{j=1}^{i-1} \mathbb{1}(f(\tau(i)) < y^{(j)}) & \text{for } f(\tau_i) < y^{(i)} \\ (1 - \tau(i)) + \sum_{j=i+1}^{n} \mathbb{1}(f(\tau(i)) > y^{(j)}) & \text{for } f(\tau_i) > y^{(i)} \end{cases}$$

*Proof.* The $\tau$-pinball loss can be expressed as

$$L_\tau(y, f(\tau)) = \begin{cases} \tau(y - f(\tau)) & \text{if } f(\tau) < y \\ (1 - \tau)(f(\tau) - y) & \text{if } f(\tau) \geq y \end{cases}$$

When training with an expected pinball loss over the discrete values of $\tau$ corresponding to the sample quantiles of the data, this means that the expected pinball loss is a function only of $f(\tau(i))$, or the quantile predictions of $f$ at those sample quantiles. This is precisely the context of ordinary regression; the loss we face is a function only of the values our function takes at the $x$-coordinates of our data points, which here are set up to be exactly the $\{\tau(i)\}_{i=1}^{n}$.

Let us consider the pinball loss applied to $\tau(i)$. To optimize ([1]), we average the losses across our data points $\{y_i\}_{i=1}^{n}$; for this proof, we (equivalently) work with the sums instead. Let our regression-equivalent cost $c(f(\tau(i)))$ be equal to the relative increase in the $\tau(i)$ pinball loss we face, summed over the data points, compared to if we had predicted $f(\tau(i)) = y^{(i)}$.

By the definition of the $\tau(i)$ pinball loss, decreasing $f(\tau(i))$ by $\epsilon$ will mean that we face an additional cost of $\tau(i)\epsilon$ for every data point we are below and moving away from and recover $(1 - \tau(i))\epsilon$ for every data point we are above and moving towards. Say we start at $f(\tau(i)) = y^{(i)}$ and choose an $\epsilon$ such that $f(\tau(i)) - \epsilon > y^{(i-1)}$. The point $y^{(i)}$ has $\tau(i)(n-1)$ data points below it and $(1-\tau(i))(n-1)$ data points above it by definition. So our marginal loss from decreasing $f(\tau(i))$ by $\epsilon$ is $\tau(i)(1-\tau(i))(n-1)\epsilon$ due to the points above $y^{(i)}$ while we recover $(1-\tau(i))\tau(i)(n-1)\epsilon$ due to the points below it. These are equal! But we also lose $\tau(i)\epsilon$ because we are now moving away from the point $y^{(i)}$ itself, making the slope of $c(f(\tau(i)))$ equal to $-\tau(i)$ in the region between $y^{(i-1)}$ and $y^{(i)}$. Each time we pass a neighboring data point, we start losing $\tau(i)\epsilon$ from that point instead of gaining $(1 - \tau(i))\epsilon$, increasing the steepness of the slope by 1. This is exactly what we want to show: $c'(f(\tau(i)))$ is equal to $-\tau(i)$ minus the number of data points smaller than $y^{(i)}$ that $f(\tau(i))$ is smaller than.

The opposite case - analyzing what happens when we increase $f(\tau(i))$ by $\epsilon$ is essentially analagous. Increasing $f(\tau(i))$ by $\epsilon$ leads to an additional cost of $(1 - \tau(i))\epsilon$ for every data point we are above and moving away from and a decreased cost of $\tau(i)\epsilon$ for every data point we are below and moving towards. This implies the slope of the cost function between $y^{(i)}$ and $y^{(i+1)}$ is $(1 - \tau(i))$, as the cost accrued and recovered by the points above and below $y^{(i)}$ cancel out and we only have the net cost due to moving away from $y^{(i)}$ itself. And for the same reason described above, passing data points adds to the slope by 1. $\qquad\square$

See Figure [6] for some example cost functions.

This cost function is nonstandard in a couple ways. First, it varies per point. And second, its steepness is a function of how many neighboring sample data points the prediction $f(\tau)$ is off by rather than just how far away it is from the sample quantile in an absolute sense.

Still, it otherwise follows the normal properties of a cost function, such as taking the value 0 when the prediction is exactly correct and (weakly) rising in a convex fashion as the prediction is too high or too low. This makes it clear why a sufficiently flexible function $f(\tau)$ will exactly pass through all of the sample quantiles and also why a more limited $f(\tau)$ will serve as a smoother, passing below some sample quantiles and above others.

**Worked Example:** Consider the dataset $\{1, 2, 5, 6, 8\}$. These correspond to the sample quantiles $\{0, 0.25, 0.5, 0.75, 1\}$. Figure [6] plots the sample quantiles on a graph, the least squares regression fit, and the expected pinball loss fit with linear $f(\tau)$. We can see that both are quite similar. For example, both estimate

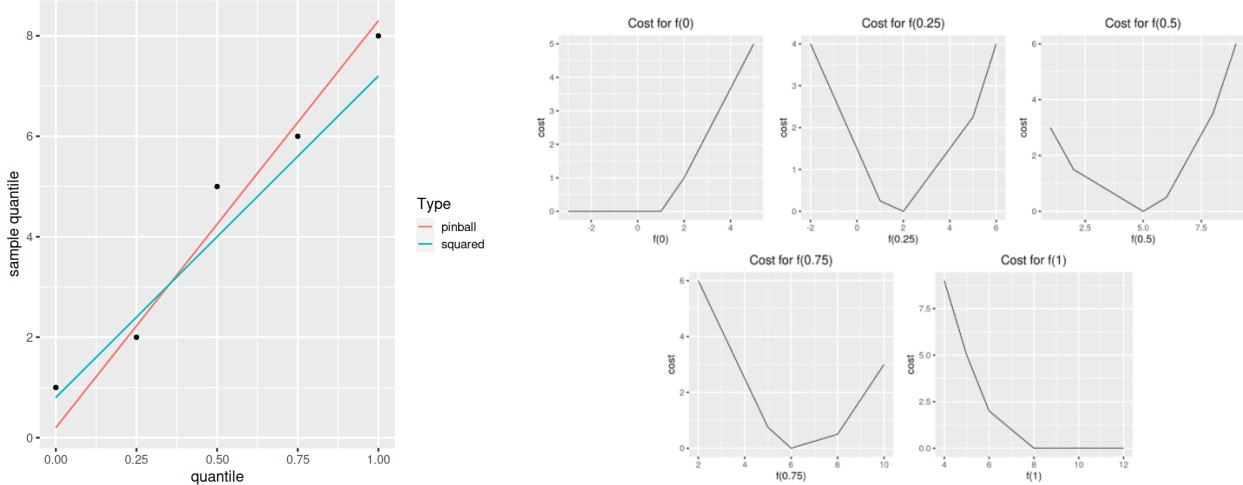

Figure 6: Given the dataset $\{1, 2, 5, 6, 8\}$, we show visually how learning $f(\tau)$ via the expected pinball loss with $\tau$ as a feature can be seen as a type of regression on the sample quantiles. For simplicity, we take $f(\tau)$ to be linear, which would be the correct functional form if the data are drawn from a uniform distribution. On the left, we plot the sample quantiles and compare the lines learned by a direct squared-loss regression on those data points (i.e. $\{\tau(i), y^{(i)}\}_{i=1}^5$) with the line learned by our method (i.e. learning $f(\tau)$ with the expected pinball loss). On the right, we show the equivalent costs imposed on predictions for each sample quantile if we reframed training $f(\tau)$ with an expected pinball loss over $\tau = \{0, 0.25, 0.5, 0.75, 1\}$ as a regression on sample quantiles. The cost is 0 when the prediction is exactly the corresponding sample quantile and weakly rising as it varies.

the median as being lower than the observed sample median. The figure also shows the regression-equivalent cost function for each of the quantile predictions, if we were to think of our method as a regression on sample quantiles.

## A.8 Calibration Error Metric Definition

A predicted conditional inverse CDF $f(x, \tau; \theta)$ is *calibrated* if

$$P(Y \leq f(X, \tau; \theta)) = \tau.$$

A calibration error metric measuring the difference between $P(Y \leq f(X, \tau; \theta))$ and $\tau$ can be approximated over a finite dataset with data points $\{x_i, y_i\}_{i=1}^n$ and quantiles $\tau_1, ..., \tau_m$. We consider the absolute deviation:

$$\frac{1}{m} \sum_{j=1}^m \left| \left( \frac{1}{n} \sum_{i=1^n} \mathbb{1}(y_i \leq f(x_i, \tau_j; \theta)) \right) - \tau_j \right| \tag{6}$$

Others such as Kuleshov et al. (2018) have considered the squared deviation.

## A.9 Hyperparameter Tuning Details

For each real data experiment, we tune hyperparameters over a validation dataset, where the validation dataset is selected according to Section 5.2. The tuning criterion was the validation pinball loss averaged over $\tau \in \{0.01, ..., 0.99\}$. For DLNs, the search spaces for real data experiments were the following:

- The number of calibration keypoints for the piecewise-linear calibration function over $\tau$ were tuned between $\{10, 20, 50, 100\}$. Note that as this number goes up, you get more model flexibility with respect to $\tau$ (only), which makes it effectively like just training separate models for each $\tau$.

- The number of lattice keypoints for $\tau$ was tuned between $\{2, 3, 5, 7, 10\}$. That controls the flexibility of the interactions between $\tau$ and the other features $x$.

- Other feature calibration keypoints were tuned between $\{5, 10, 15, 20\}$, which are common values for DLNs.

- Step sizes were tuned between $\{0.001, 0.005, 0.01, 0.05, 0.1\}$

- minibatch sizes were tuned between $\{1000, 10000\}$.

- Number of steps was tuned between $\{100, 1000, 10000\}$.

To tune the DLN architecture for experiments with real data, we searched over the following hyperparameters:

- Air Quality: number of lattices $\in \{4, 8\}$, number of features per lattice $= 6$,

- Puzzles: number of lattices $\in \{8, 16, 32\}$, number of features per lattice $= 5$

- Traffic: single lattice which includes all features (not ensemble)

- Wine: number of lattices $\in \{100, 200, 400, 800\}$, number of features per lattice $\in \{2, 4, 8\}$.

# Expected Pinball Loss For Quantile Regression And Inverse CDF Estimation

**Anonymous authors**

## Abstract

We analyze and improve a recent strategy to train a quantile regression model by minimizing an expected pinball loss over all quantiles. We give an asymptotic convergence rate that shows that minimizing the expected pinball loss can be more efficient at estimating single quantiles than training with the standard pinball loss for that quantile, an insight that generalizes the known deficiencies of the sample quantile in the unconditioned setting. Then, to guarantee a legitimate inverse CDF, we propose using flexible deep lattice networks with a monotonicity constraint on the quantile input to guarantee non-crossing quantiles, and show lattice models can be regularized to the same location-scale family. Our analysis and experiments on simulated and real datasets show that the proposed method produces state-of-the-art legitimate inverse CDF estimates that are likely to be as good or better for specific target quantiles.

## 1 Introduction

Real world applications often seek estimates of the quantiles of a random variable. For example, ~~a pizza place~~ an airline would like to tell ~~customers~~ passengers that 90% of the time ~~, their order will be delivered within 40 minutes~~their flight will arrive within 4 hours, so they know they can make their connecting flights. In addition, ~~we may condition the~~one might condition that estimate on features. For example, ~~conditioned on the time-of-day being 5pm, and the order being for twenty pizzas~~if the forecast is for 8 inches of snow at the destination airport that day, the conditional estimate ~~is that 90%~~might worsen to 90% of the time their ~~order will arrive in less than 73~~flight arriving within 6 hours and 30 minutes.

~~When there are no features,~~ Quantile regression learns a model from training examples to produce such estimates for quantiles. If the model can estimate all quantiles, it is an inverse CDF estimator.

### 1.1 Formal Definitions

Formally, let random variable $Y \in \mathbb{R}$ have cumulative distribution function (CDF) $F$, and for $\tau \in (0,1)$, the $\tau$-quantile of $Y$ is defined as $q_\tau = F^{-1}(\tau) = \inf\{q : P(Y \leq q) \geq \tau\}$. In the *conditional* setting ~~, the pair of random variables $(X, Y) \in \mathbb{R}^D \times \mathbb{R}$, and~~ where one also has random feature vector $X \in \mathbb{R}^D$, the *conditional* $\tau$-quantile of $Y$ for feature vector $X$ is defined as $q_\tau(x) = \inf\{q : P(Y \leq q | X = x) \geq \tau\}$.

*Quantile regression* takes as training data a set of $n$ pairs $(x, y) \in \mathbb{R}^D \times \mathbb{R}$ from a joint distribution over $(X, Y)$, and forms an estimator for one or more of the conditional quantiles of $Y$ for any value of $X$. A standard objective to train a model to estimate the quantile for $\tau$ is to minimize the *pinball loss* (Koenker & Bassett, 1978), $L_\tau(y, \hat{y}) = \max(\tau(y - \hat{y}), (\tau - 1)(y - \hat{y}))$ for $y, \hat{y} \in \mathbb{R}$. In the unconditioned case with no features $X$, the training data is simply a set of $n$ scalar values $\{y_i\}_{i=1}^n$, and minimizing the pinball loss has the satisfying property that it selects the empirical quantile of the training set (Chernozhukov, 2005).

### 1.2 Expected Pinball Loss

Recent work by Tagasovska & Lopez-Paz (2019) proposed training one deep neural network (DNN) model $f(x, \tau)$ that takes $\tau$ as an input, and is trained to minimize an *expected* pinball loss ~~, where $\tau$ is drawn~~

from the over a random $\mathcal{T}$, drawn from a uniform distribution. That work is important because it provides *simultaneous quantile regression* of the entire inverse CDF. However, that work left open a couple of important theoretical and practical issues that we address in this paper.

### 1.3  Is Expected Pinball Loss Worse At Specific Quantiles?

The first issue we address is whether one pays a price in estimation accuracy for a model $f(x, \tau)$ that can predict any quantile, compared to a model trained specifically for one target quantile. Surprisingly, we show theoretically with a convergence rate analysis that learning the full inverse CDF model $f(x, \tau)$ can produce a more efficient estimator for a single quantile than minimizing the pinball loss for just a single quantile, depending on the true distribution, quantile, and function class. Our simulations and real-world experiments confirm that in practice one does often win on single quantile estimates by training with the *expected* pinball loss, though the full inverse CDF model $f(\tau; x)$ may take a bit longer to train and a bit more memory to store.

We also demonstrate a novel use case of the expected pinball loss for single quantile regression - training with a Beta distribution over $\tau$ - and show how that can also outperform single-quantile regression (without the added bonus of accurately estimating the full inverse CDF).

### 1.4  The Problem Of Non-crossing Quantiles

The second issue we address is that training a DNN model with the expected pinball loss as proposed by Tagasovska & Lopez-Paz (2019) may fail to produce a legitimate inverse CDF because it does not guarantee *non-crossing quantiles*: that any two quantile estimates satisfy the common sense requirement that $q_\tau(x) \geq q_{\tau'}(x)$ for $\tau \geq \tau'$ at every $x$. For example, a model that does not guarantee non-crossing quantiles could tell a customer might tell a passenger there is a 90% chance their delivery will arrive within 38 minutes, but an flight will arrive in 4 hours, but that there is a 80% chance it will arrive within 40 their flight will arrive in 4 hours and 12 minutes. Such a model may make a customer confused and distrust the model, worrying it is brokenor buggy. non-crossing quantile estimates have long been thought objectionable (Koenker & Bassett, 1978).

To test whether non-experts people would actually notice or mind non-crossing estimates, we emailed 100 frequent customers of *Company Name Removed for Blind Review*, who are estimated to be 70% college-educated and 20% with post-graduate education, and told them we were considering giving them estimates of how long it would take for their package to arrive, and gave them example arrival time estimates for the $50\%, 80\%, 90\%$ quantiles, and asked for their feedback on how useful they would find such estimates. However, the example estimates were all made-up: we gave 50 of the customers estimates with crossing 80% and 90% quantiles (3 days, 8 days and 7 days), and the other 50 customers got estimates with non-crossing quantile predictions (3 days, 7 days, 8 days). From the 50 emails with non-crossing estimates, we received 9 emails back to say they would appreciate such estimates, and no concerns. From the 50 emails with crossing estimates, we received 16 emails back all expressing enthusiasm for having such predictions, but 11 of the 16 emails pointed out the crossing quantiles with negative adjectives including "wrong," "broken," "buggy," "didn't make sense to me."

Embarrassing quantile-crossing mistakes are known to happen often in quantile regression in general (He, 1997; Koenker & Bassett, 1978). (He, 1997; Koenker & Bassett, 1978); see also Table 1. Tagasovska & Lopez-Paz (2019) hypothesized that training with an expected pinball loss induces smoothing across $\tau$ that would reduce non-crossing. However, we show experimentally (Table 3) that a flexible model like a DNN optimized to minimize the expected pinball loss easily suffers a problematic amount of quantile crossing. To address non-crossing without losing model flexibility, we propose using deep lattice networks (DLNs) (You et al., 2017) with a monotonicity shape constraint on $\tau$ to guarantee non-crossing quantiles, thus providing legitimate inverse CDF estimates. Additionally, we show that the DLN function class is amenable to two additional kinds of useful regularization for quantile regression: restricting the learned inverse CDF to a location-scale family, and restricting other features to have only monotonic effect on the predictions.

## 1.5    Other Uncertainty Estimation Problems

In this paper, we focus on estimating multiple quantiles and the extreme case of all quantiles: inverse CDF estimation. Such problems are only part of the landscape of estimating uncertainty. A closely related problem is estimating *prediction intervals* such that an $\alpha$-interval contains the true label alpha is and sharp around $\alpha$. Prediction intervals can be generated by optimizing for the calibration and sharpness of the interval (Pearce et al., 2018; Chung et al., 2021). Tagasovska & Lopez-Paz (2019) instead formed them by choosing a pair of quantiles from a multiple quantile regression model, which works well because the standard pinball lose used for training quantile regression models balances calibration and sharpness. A different type of uncertainty estimation problem is estimating the uncertainty of a specific statistic, such as the mean in a regression model; such problems are often handled with Bayesian methods. All of these estimation problems become more challenging in the out-of-domain (OOD) setting.

## 1.6    Paper Organization

Next, in Section 2 we formalize the problem of producing legitimate inverse CDF estimates with quantile regression. Then in Section 3 we dig into what is known about beating the ~~pinball loss~~ standard *pinball loss* by smoothing over multiple quantile estimates, and we give a new asymptotic convergence rate for minimizing the expected pinball loss. In Section 4 we propose using DLNs as the model architecture. Experiments in Section 5 on simulations and real-world data show that the proposed non-crossing DLNs provide competitive, trust-worthy estimates.

## 2    Estimating A Legitimate Inverse CDF

We give a constrained optimization problem to train a quantile regression estimator without crossing quantiles, which can produce a legitimate inverse CDF. Let $\{(x_i, y_i)\}_{i=1}^n$ be a training set where each $x_i \in \mathbb{R}^D$ is a feature vector and $y_i \in \mathbb{R}$ is a corresponding label. Denote the ~~inverse CDF~~ estimator $f(x, \tau; \theta)$, where $\tau \in (0, 1)$ ~~is an auxiliary feature that~~ specifies the quantile of interest, and the model $f : \mathbb{R}^{D+1} \to \mathbb{R}$ is parameterized by $\theta \in \mathbb{R}^m$.

Recall the standard pinball loss given $\tau$ is $L_\tau(y, \hat{y}) = \max(\tau(y - \hat{y}), (\tau - 1)(y - \hat{y}))$. Let ~~$\mathcal{T} \sim \text{unif}(0, 1)$~~ $\mathcal{T} \sim P_{\mathcal{T}}$ denote a random $\tau$ so we can minimize the *expected* pinball loss over $\mathcal{T}$ to train $f(x, \tau; \theta)$ ~~(Tagasovska & Lopez-Paz, 2019). In addition~~ as in Tagasovska & Lopez-Paz (2019), though here we generalize to $P_{\mathcal{T}}$, rather than assuming a uniform distribution for $\mathcal{T}$. The standard pinball loss can be written as the special case where $P_{\mathcal{T}}$ is the Dirac delta distribution on the target quantile.

Also, we constrain the empirical loss minimization with non-crossing constraints, producing the following constrained training objective:

$$\min_\theta \mathbb{E}_{\mathcal{T}} \sum_{i=1}^n L_{\mathcal{T}}(y_i, f(x_i, \mathcal{T}; \theta)) \tag{1}$$

$$\text{s.t. } f(x, \tau^+; \theta) \geq f(x, \tau^-; \theta) \, \forall x \in \mathbb{R}^D, \tau^+ \geq \tau^-. \tag{2}$$

### 2.0.1    Related Work In Minimizing Sum Of Pinball Losses

The idea of training one model that can predict multiple quantiles by minimizing a sum of pinball losses with non-crossing constraints dates back to at least the 2006 work of Takeuchi et al. (2006) for a *pre-specified, discrete set* of quantiles. Other prior work used a similar mechanism for a *pre-specified discrete set* of quantiles, and for *more-restrictive* function classes that are monotonic on $\tau$, such as linear models (Bondell et al., 2010), and two-layer *monotonic* neural networks (Cannon, 2018), which are known to have limited flexibility (Daniels & Velikova, 2010). Monotonic DNN models with more layers (Minin et al., 2010; Lang, 2005) or min-max layers (Sill, 1998) can theoretically provide universal approximations, but have not performed as well experimentally as DLNs (You et al., 2017). Monotonic neural nets have also been proposed for estimating a *CDF* (Chilinski & Silva, 2018)).

Training only for (1) without (2) (Tagasovska & Lopez-Paz, 2019) does not guarantee a legitimate inverse CDF because the estimated quantiles can cross. An in-between strategy is to instead add to (1) a penalty on violations of the monotonicity constraint on training samples (and one can add virtual examples as well for more coverage) (Sill & Abu-Mostafa, 1997), but applying that strategy to quantile regression cannot guarantee non-crossing quantiles everywhere.

### 2.0.2 Related Work In Estimating The Inverse CDF

There are two main prior approaches to estimating a *complete* legitimate inverse CDF. The first is nonparametric. For example, k-NN can be extended to predict quantiles by taking the quantiles rather than the mean from within a neighborhood (Bhattacharya & Gangopadhyay, 1990). Similarly, quantile regression forests (Meinshausen, 2006) use random forests leaf nodes to generate local empirical quantiles.

The second strategy is to predict a parametric inverse CDF for any $x$. Traditionally these methods have been fairly rigid. He (1997) developed a method to fit a shared but learned location-scale family across $x$. Yan et al. (2018) proposed a modified 4-parameter Gaussian whose skew and variance was dependent on $x$. Recently, Gasthaus et al. (2019) proposed the *spline quantile function DNN* (SQF-DNN), which is a DNN model that takes an input $x$, and outputs the parameters for a monotonic piecewise-linear quantile function on $\tau$. SQF-DNN can fit any continuous bounded distribution, given sufficient output parameters, and because for any $x$ the output is a monotonic function on $\tau$, guarantees no quantile crossing. Though Gasthaus et al. (2019) focused on RNNs, their framework is easy to apply to the generic quantile regression setting as well, which we do in our experiments.

## 3 Comparison To The Single Quantile Pinball Loss

We show through simulations and convergence rate analysis that minimizing the expected pinball loss can be better at estimating the $\tau$-quantile than minimizing the pinball loss just for $\tau$, and that these positive results are foreshadowed by classic results for unconditional quantile estimates. This section focuses on the unconditioned case (no features $x$), we will return to the problem given features $x$ in Section 4.

### 3.1 Deficiency Of The Sample Quantile

Given only samples from $Y$, minimizing a pinball loss produces the corresponding sample quantile (Chernozhukov, 2005). However, it has long been known that simply taking the sample quantile is not necessarily the best quantile estimator, and in particular that it can be beaten by strategies that smooth multiple sample quantiles (Harrell & Davis, 1982; Kaigh & Lachenbruch, 1982; David & Steinberg, 1986; Reiss, 1980; Falk, 1984).

One important classic result~~is~~, which follows from the Lehmann-Schaffe theorem, ~~which~~ says that if the data are known to be from a uniform $[a, b]$ distribution, then the ~~average of the sample min and sample max, rather than~~ minimum variance unbiased estimator for the median is *not* the sample median, ~~is the minimum variance unbiased estimator for the median.~~ rather it is the average of the sample min and sample max.

Even if we do not know anything about the true distribution, general-purpose methods that smooth multiple quantiles of the data when estimating a particular quantile have proven effective. In fact, the widely used Harrell-Davis estimator is a weighted average of all the sample order statistics where the weights depend on $\tau$, and the estimator for the median is asymptotically equivalent to computing the mean of bootstrapped medians (Harrell & Davis, 1982).

The prior work above suggests that there can be a better training loss than $\tau$-specific pinball loss for the general problem of quantile regression given features - one that increases efficiency by better smoothing across the quantiles. Indeed, by minimizing the expected pinball loss as in (1), the estimate of any quantile is affected by the other quantiles (given some smoothness in $f$). We provide some further intuition about how it smooths in Appendix A.7.

## 3.2 Comparison to Harrell-Davis, A Classic Quantile Smoother

To emphasize the similarity of minimizing the expected pinball loss to the Harrell-Davis quantile smoothing estimator, which only works in the unconditioned setting of estimating $f(\tau;\theta)$, we compare both to the sample median. We simulate $Y$ drawn from an exponential distribution.

The expected pinball loss approach lets us tune how much to smooth over quantiles in two different ways, both of which effectively recover the sample quantiles at one extreme. The first approach is most directly analogous to the Harrell-Davis estimator: minimizing not just the target pinball loss, but a weighted average of pinball losses around the target quantile. The second, more ambitiously, trains over all quantiles equally, but controls the complexity of the function class that models the inverse CDF.

For the first set of experiments, we use a beta distribution for $P(\mathcal{T})$ with its mode set to the quantile of interest ($\tau = 0.50$, or $\tau = 0.99$), leaving its concentration as a hyperparameter. We restrict $f(\tau)$ to be linear (which is a special case of the proposed DLN function class, see Sec. 4).

Results are shown in the top row of Figure 1. As expected, the sample median is not as good as the Harrell-Davis estimate, which regularizes across the quantiles similar to the expected pinball loss. The more concentrated the beta distribution is (further right on the x-axis), the less regularized the DLN estimator is; in the limit all the mass of $P_\mathcal{T}$ is concentrated on the target quantile and (1) approaches the standard pinball loss. For both the median (left) and 99th percentile (right) estimate, most choices of the beta concentrations perform better than the sample median, and a broad swath of beta concentrations perform better than Harrell-Davis. In practice, one would need to cross-validate the concentration hyperparameter.

Table 1: Estimating the median and 99th percentile of an exponential distribution via cross-validated DLN training. This is a companion to Figure 1 in which results are additionally provided for a DLN in which the number of keypoints is tuned via 10-fold cross-validation over the $N = 51$ or $N = 505$ training points, respectively, and the model is then retrained on the full data with the selected number of keypoints.

| Model | $\tau = 0.5$ | $\tau = 0.99$ |
|---|---|---|
| CV-DLN, Expected Pinball | $0.019263 \pm 0.00086$ | $\mathbf{0.17566 \pm 0.0080}$ |
| Sample Quantile | $0.019968 \pm 0.00046$ | $0.19552 \pm 0.0043$ |
| Harrell-Davis | $\mathbf{0.018081 \pm 0.00038}$ | $0.18887 \pm 0.0042$ |

An alternative is to use a uniform distribution for $P_\mathcal{T}$, and instead control the amount of regularization across $\tau$ by using a more flexible function $f(\tau)$. We test that strategy in our second set of experiments. We use a piecewise-linear function (PLF) $f(\tau;\theta)$, where the parameters $\theta$ are the values of $f(\tau;\theta)$ at $K$ keypoints pinned at the $K$ sample quantiles of the data (which is a special case of the proposed DLN function class, see Section 4). Bigger $K$ makes $f(\tau;\theta)$ more flexible, which reduces the dependence of the target quantile on the impact other quantiles have on the expected loss. The results are shown in the bottom row of Figure 1. On the left for the median estimation, over a wide range of model flexibility $K$, the DLN performs better than the sample median, and for a small range of $K$ better than the Harrell-Davis estimator.

On the right-side of the bottom row of Figure 1, the results are stronger for estimating the tail $\tau = 0.99$ from $N = 505$ training samples. The

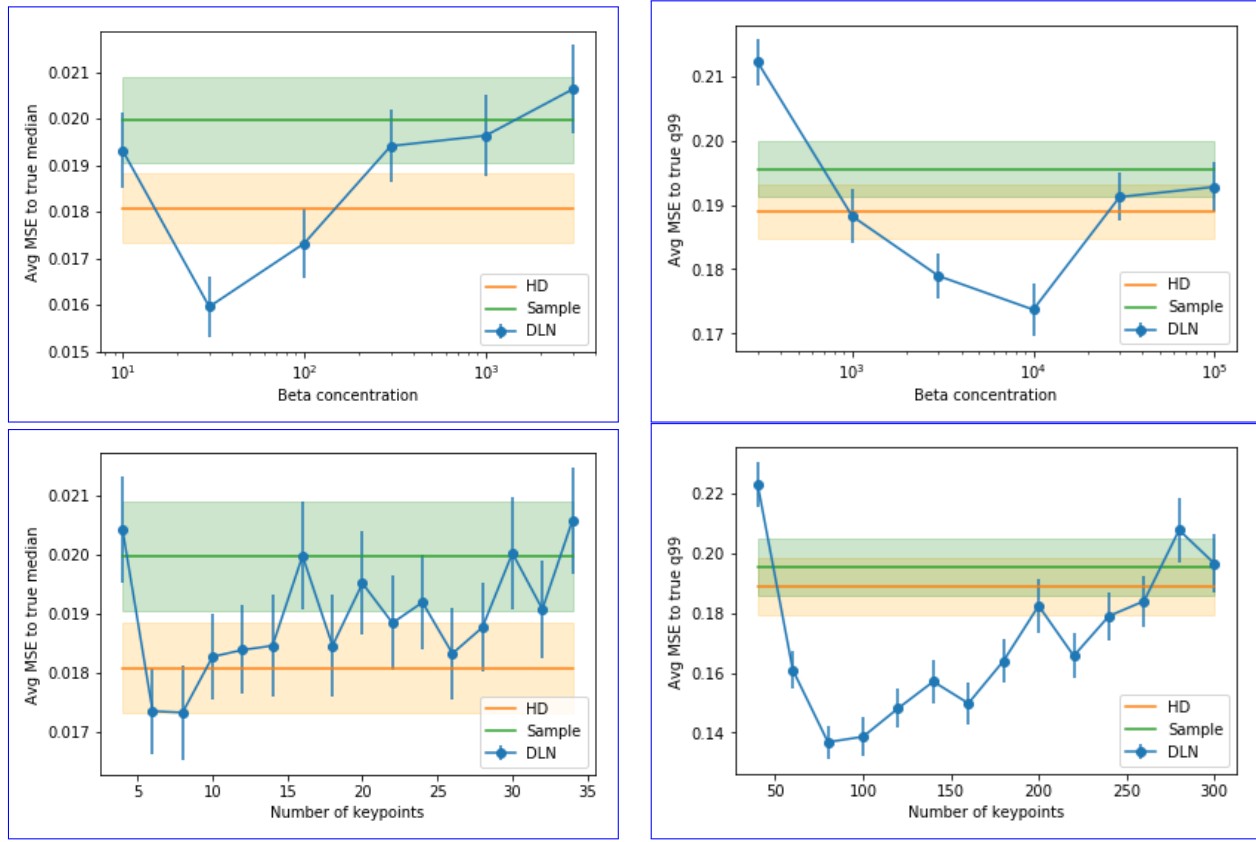

Figure 1: Estimating the median (left plots) and 99th percentile (right plots) of an exponential distribution (no features) by the sample median, by Harrell-Davis, and by solving (1) and (2) with a DLN function class. The median estimates were from $N = 51$ training samples. The $\tau = .99$ estimates were from $N = 505$ training samples. Error bars are computed over 1,000 repeats. **Top:** The DLN is just a linear model on $\tau \in [0, 1]$ with $\theta \in \mathbf{R}^2$ and $P_{\mathcal{T}}$ is a beta distribution. Results are shown for six choices of the beta concentration across the x-axis, where a bigger concentration makes $P_{\mathcal{T}}$ more peaked around the target quantile, imposing less smoothing across quantiles. **Bottom:** The $P_{\mathcal{T}}$ is the uniform distribution, and the DLN is a piece-wise linear function (PLF) with $K$ keypoints. Larger $K$ makes the model more flexible, and imposes less smoothing across quantiles. Results for different choices of $K$ are shown across the x-axis.

DLN solving (1) and (2) statistically significantly beats the Harrell-Davis method for a broad range of model flexibility ~~(that is, number of~~ as expressed by the $K$ keypoints in the ~~fitted PLF)~~DLN's PLF.

Table 1 shows that when we choose the value of $K$ based on 10-fold cross-validation, the DLN outperforms the sample quantile but not the Harrell-Davis estimator when predicting the median, and outperforms both when predicting $\tau = 0.99$.

The Harrell-Davis and related estimators ~~that only apply~~ can only be applied to the unconditioned case, whereas training ~~DLNs as per~~ with the expected pinball loss (1) and monotonicity constraints (2) ~~extend~~ extends seamlessly to *conditional* quantile estimation with ~~any number of $x$ features, which we return to in Sections 4 and 5~~ feature vector $x \in \mathbf{R}^D$, bringing the gains of regularizing across $\tau$ to a much larger class of problems.

~~Estimating the quantiles of an exponential distribution (unconditioned). For estimating the median (left, N=51 data points) and 99th quantile (right, N=505), we compare the efficiency of the sample quantile, Harrell-Davis estimator, and learned DLN( here, a PLF) over $\tau$ trained with an expected pinball loss. The x-axes show the effect of different numbers of keypoints $K$ in the PLF (the DLN), with more keypoints~~

~~corresponding to a more flexible function over $\tau$. The y-axis shows the MSE to the true target quantile. Error bars are computed over 5,000 repeats.~~

### 3.3 Convergence Rate

We give a new asymptotic convergence rate for estimating the full inverse CDF by minimizing the expected pinball loss (1), optionally with a monotonicity constraint (2), and compare that to the convergence rate of single-quantile estimation learned by minimizing an individual pinball loss.

We show that depending on the underlying data distribution and the function class chosen to estimate the inverse CDF, the simultaneous estimation can produce a more efficient estimator for a single target quantile than minimizing the pinball loss for only that quantile.

For tractability, we work in the setting where there are no features $x$ to condition on, only a set of training samples $\{y_i\}$, and we are interested in estimating the quantile $F^{-1}(\tau)$ for a random variable $Y \in \mathbb{R}$ given IID samples $\{y_i\}_{i=1}^n$. Such analysis is relevant to the conditional case because for function classes that are flexible over ~~$x$~~ $X$, they intuitively will have similar dynamics at each value of ~~$x$~~ $X$ as the unconditioned case we work with here~~:~~. We formalize this in a basic corollary which extends our unconditional theory to conditional estimation for finite categorical $X$ and an extremely flexible function class, but note that general theory for full conditional inverse CDF estimation remains an open problem. Section 5.5 provides ~~some experimental evidence~~ experimental evidence for the conditional case.

#### 3.3.1 Review of Single Pinball Loss Convergence Rate

We start with the well-understood case of single-quantile estimation. Let ~~$\theta_\tau^* = \arg\min_\theta E_Y[L_\tau(Y, \theta)]$~~ $\theta_*^* = \arg\min_\theta E_Y[L_\tau(Y, \theta)]$ be the true minimizer for the corresponding pinball loss and ~~$\hat{\theta}_\tau^{(n)} = \arg\min_\theta \sum_{i=1}^n [L_\tau(y_i, \theta)]$~~ $\hat{\theta}_\tau^{(n)} = \arg\min_\theta \frac{1}{n}\sum_{i=1}^n [L_\tau(y_i, \theta)]$ be the empirical minimizer over our data. (In our context, $\theta^*(\tau)$ is the true $\tau$ quantile of $Y$ and $\hat{\theta}_\tau^{(n)}$ is the $\tau$ sample quantile of the data (Chernozhukov, 2005).) The asymptotic convergence rate for this single quantile estimate is given by,

$$\sqrt{n}(\hat{\theta}_\tau^{(n)} - \theta_\tau^*) \xrightarrow{d} \mathcal{N}\left(0, \frac{\tau(1-\tau)}{p(\theta_\tau^*)^2}\right), \tag{3}$$

where $p(y)$ is the density for $Y$ (Koenker, 2005).

#### 3.3.2 Expected Pinball Loss Convergence Rate

Next, consider learning the full inverse CDF $f(\tau; \theta)$ parameterized by $\theta \in \Theta \subseteq \mathbb{R}^m$ by minimizing the expected pinball loss:

$$\theta^* = \underline{\arg\min}\,\underset{\theta \in \Theta}{\arg\min}\, E_\mathcal{T}[E_Y[L_\mathcal{T}(Y, f(\mathcal{T}; \theta))]],$$

where $\mathcal{T} \in (0, 1)$ is a random variable drawn from some distribution independently of $Y$. Note that the monotonicity constraints in Equation (2) can be wrapped in the function class $f$ and feasible parameter set $\Theta$. Let $\hat{\theta}^{(n)}$ be the corresponding minimizer under an empirical distribution over $Y$ with samples $\{y_i\}_{i=1}^n$. In Theorem 1 below, we present a general asymptotic convergence rate for $\hat{\theta}^{(n)}$ that depends on the properties of the parametric function $f$, feasible parameter space $\Theta$, and the distribution of $\mathcal{T}$. A natural choice for the distribution of $\mathcal{T}$ is Unif$(0, 1)$, but in fact, depending on the distribution of $Y$ and properties of the function class $f(\tau; \theta)$, other distributions could lead to lower asymptotic variance. In choosing the distribution of $\mathcal{T}$, note that choosing distributions with wider support will make it easier to satisfy the requirements for asymptotic normality in Theorem 1 below (see note in Appendix A.4).

**Theorem 1.** *Suppose $f(\tau; \theta)$ is well specified ~~: there exists~~ with a unique $\theta^* \in \Theta \subseteq \mathbb{R}^m$ such that $f(\tau; \theta^*) = F^{-1}(\tau)$ for all $\tau \in (0, 1)$~~. Suppose~~, and that $\theta^*$ is also the unique minimizer for the risk $R(\theta) = E_{Y, \mathcal{T}}[L_\mathcal{T}(Y, f(\mathcal{T}; \theta))]$. Suppose the estimator $\hat{\theta}^{(n)}$ is weakly consistent, $\hat{\theta}^{(n)} \xrightarrow{P} \theta^*$. Suppose further that the function $\theta \mapsto f(\tau; \theta)$ is continuous, locally Lipschitz, and twice differentiable at $\theta = \theta^*$ for all*

$\tau \in (0, 1)$. *Then,*

$$\sqrt{n}(f(\tau; \hat{\theta}^{(n)}) - f(\tau; \theta^*)) \xrightarrow{d} \mathcal{N}\left(0, \nabla_\theta f(\tau; \theta^*)^\top Q^{-1} V (Q^{-1})^\top \nabla_\theta f(\tau; \theta^*)\right) \tag{4}$$

*where* $Q = E_\mathcal{T}[p(f(\mathcal{T}; \theta^*))\Gamma(\mathcal{T})]$, $V = E_\mathcal{T}[\mathcal{T}(1 - \mathcal{T})\Gamma(\mathcal{T})]$, *with* $\Gamma(\tau) = \nabla_\theta f(\tau; \theta^*)\nabla_\theta f(\tau; \theta^*)^\top$.

Lemma 1 provides a simple example of sufficent conditions for the consistency condition in Theorem 1, which in general depends on properties of the function $f$ and parameter space $\Theta$.

**Lemma 1.** If $f(\tau, \theta)$ is convex in $\theta$ for all $\tau \in (0, 1)$, $\Theta$ is convex, and $\theta^*$ is in the interior of $\Theta$, then $\hat{\theta}^{(n)}$ is weakly consistent, $\hat{\theta}^{(n)} \xrightarrow{P} \theta^*$.

Lemma 1 applies when the optimization problem in (1) is convex. Aside from convexity, other example conditions could include compactness of $\Theta$, or further Lipschitz continuity or bounded moment conditions on $f$ (Powell, 2010).

Given a specific distribution of $\mathcal{T}$, a function class of the inverse CDF $f$, and a data distribution $Y$, we can directly compare the asymptotic convergence rate of the direct quantile estimates (Equation (3)) to that of quantile estimates from the learned inverse CDF (Equation (4)). We illustrate this below with several examples.

### 3.3.3 Example 1: Single Parameter Uniform

To build intuition for the implications of Theorem 1, we consider a centered uniform distribution $Y \sim \text{Unif}(-\frac{\alpha}{2}, \frac{\alpha}{2})$. Let $f(\tau; \theta) = \theta(\tau - \frac{1}{2})$. Expanding the asymptotic variance term in Theorem 1,

$$Q^{-1} V (Q^{-1})^\top = \frac{\alpha^2 E_\mathcal{T}[-\mathcal{T}^4 + 2\mathcal{T}^3 - \frac{5}{4}\mathcal{T}^2 + \frac{1}{4}\mathcal{T}]}{E_\mathcal{T}[\mathcal{T}^2 - \mathcal{T} + \frac{1}{4}]^2}.$$

This depends on the first four moments of the distribution of $\mathcal{T}$, the choice of which is up to the algorithm designer. For the natural choice of $\mathcal{T} \sim \text{Unif}(0, 1)$, the asymptotic variance for estimating a specific $\tau_0$-quantile using the function $f(\tau_0; \hat{\theta}^{(n)}) = \hat{\theta}^{(n)}(\tau_0 - \frac{1}{2})$ is $1.2\alpha^2(\tau_0 - \frac{1}{2})^2$. This variance is lowest when $\tau_0 = 0.5$, and increases to $0.3\alpha^2$ at the most extreme quantiles when $\tau = 0$ or $\tau = 1$.

We next compare this to estimating the $\tau_0$-quantile using the single pinball loss $L_{\tau_0}$. Equation (3) shows that the estimate $\hat{\theta}^{(n)}_{\tau_0}$ has asymptotic variance $\alpha^2 \tau_0(1 - \tau_0)$. This shows that depending on the exact value of $\tau_0$, the single quantile estimate could be less or more efficient than the estimate resulting from evaluating the function $f(\tau_0; \hat{\theta}^{(n)})$. The single quantile estimate is more efficient for the extreme quantiles, whereas the inverse CDF estimate is more efficient for estimating the median. Intuitively, the inverse CDF median estimate benefits from fitting the inverse CDF function to the surrounding data points, which is reminiscent of classic quantile smoothing.

### 3.3.4 Example 2: Two-Parameter Uniform

We next consider a two-parameter uniform distribution example: for $Y \sim \text{Unif}(\alpha, \beta)$, $p(y) = \frac{1}{\beta - \alpha}$. Let $\theta \in \mathbb{R}^2$ with $f(\tau; \theta) = \theta_0 + \theta_1 \tau$. That is, we learn a two-parameter line as our inverse CDF. We then have $\nabla_\theta f(\tau; \theta) = \begin{bmatrix} 1 \\ \tau \end{bmatrix}$ and can therefore write the $Q$ and $V$ matrices from Theorem 1 as

$$Q = \frac{1}{\beta - \alpha} E_\mathcal{T} \begin{bmatrix} 1 & \mathcal{T} \\ \mathcal{T} & \mathcal{T}^2 \end{bmatrix}; \qquad V = E_\mathcal{T} \begin{bmatrix} \mathcal{T} - \mathcal{T}^2 & \mathcal{T}^2 - \mathcal{T}^3 \\ \mathcal{T}^2 - \mathcal{T}^3 & \mathcal{T}^3 - \mathcal{T}^4 \end{bmatrix}.$$

Evaluating each of these matrices for $\mathcal{T} \sim \text{Unif}(0, 1)$ we arrive at

$$Q^{-1} V (Q^{-1})^\top = (\beta - \alpha)^2 \begin{bmatrix} \frac{7}{15} & -\frac{3}{5} \\ -\frac{3}{5} & \frac{6}{5} \end{bmatrix},$$

and therefore an asymptotic variance for a particular $\tau_0$-quantile estimate $f(\tau_0; \hat{\theta}^{(n)}) = \hat{\theta}_0 + \hat{\theta}_1 \tau_0$ of $(\beta - \alpha)^2(\frac{7}{15} + \frac{6}{5}\tau_0^2 - \frac{6}{5}\tau_0)$. This is a quadratic function that finds its minimum at $\tau = 0.5$ and is highest (on the unit interval) at $\tau = 0$ and $\tau = 1$.

We can compare it to the convergence for single-pinball regression of $(\beta - \alpha)^2 \tau(1 - \tau)$, which is a quadratic function that attains its *maximum* at $\tau = 0.5$.

As we see in Figure 2, the variance is smaller for the expected pinball regression for quantiles between roughly $\tau = 0.3$ and $\tau = 0.7$. The opposite is true for the extreme quantiles.

We validate these results empirically through simulation in Table 2. In each simulation, we draw 1,000 data points from Uniform(0,1) and seek to best estimate the true 10th, 50th, and 90th quantiles (i.e. 0.1, 0.5, and 0.9). We train a 2-keypoint DLN, which effectively learns a linear model $\theta_0 + \theta_1 \tau$, using the expected pinball loss, and compare it to taking the sample quantile or using the Harrell-Davis estimator. We find the DLN performs best at estimating the median but worse at the extreme quantiles. Note that this pattern differs from the results we see in Figure 1, where on an exponential distribution, a flexible DLN without knowledge of the true distribution is relatively better at an extreme quantile. This is because the specific dynamics depend on the data distribution and the functional form of the model trained with the expected pinball loss.

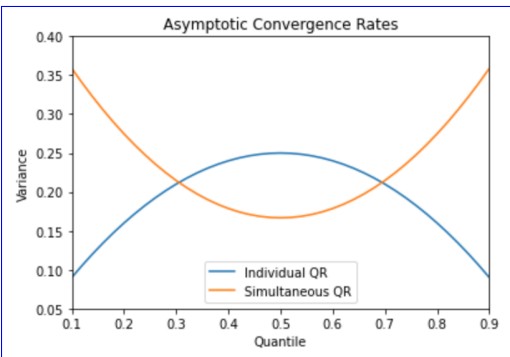

Figure 2: Plots of the asymptotic convergence variances for the expected pinball regression (orange) and individual single-pinball regressions (blue) for different quantiles, when we assume that the true distribution is Uniform$(\alpha, \beta)$, the expected pinball loss regression learns a linear model, and we assume $\beta - \alpha = 1$. Relaxing the last assumption would scale the y-axis but not change the intersection points.

Table 2: Simulation results for unconditional quantile estimation given 1,000 samples of a Uniform(0,1) random variable. Results are shown for monotonic DLNs trained with the expected pinball loss over uniformly sampled $\tau$, sample quantiles, and the Harrell-Davis estimator. DLNs are learned with 2 keypoints in order to embed the correct linear assumption on the inverse CDF.

| Model | $\tau = 0.1$ | $\tau = 0.5$ | $\tau = 0.9$ |
|---|---|---|---|
| DLN, Expected Pinball | $0.000109 \pm 0.000005$ | $\mathbf{0.000077 \pm 0.000004}$ | $0.000109 \pm 0.000005$ |
| Sample Quantile | $\mathbf{0.000097 \pm 0.000004}$ | $0.000245 \pm 0.000010$ | $\mathbf{0.000086 \pm 0.000004}$ |
| Harrell-Davis | $\mathbf{0.000089 \pm 0.000004}$ | $0.000247 \pm 0.000011$ | $\mathbf{0.000083 \pm 0.000004}$ |

Overall, these examples illustrate the sometimes counterintuitive fact that it can be more asymptotically efficient to estimate a quantile through learning a full inverse CDF than it would be to just estimate the single quantile alone. In learning the full inverse CDF, the algorithm designer also gets to make a key choice in the distribution of $\mathcal{T}$ used in the learning objective. The examples above showed how the asymptotic convergence rates depend on the moments of $\mathcal{T}$, and we empirically revisit the choice of the distribution of $\mathcal{T}$ in experiments Section 5.

### 3.4 Extension to Conditional Quantile Estimation

The asymptotic theory so far shows that expected pinball loss estimators can sometimes outperform single quantile estimators in the absence of conditioning on any additional features $X$. Conditional quantile estimation in general adds a considerable amount of theoretical complexity which we do not fully address here, and to the best of our knowledge, characterization of convergence rates for full conditional inverse

CDF estimation remains an open problem. However, the provided unconditional asymptotic theory can yield intuition and extreme examples for cases when estimating the full conditional inverse CDF would outperform single conditional quantile estimation. For instance, when $X$ is a finite categorical feature, and the function $f$ and parameter space $\Theta$ are expressive enough in $X$, Theorem 1 can be directly applied to the individual quantile estimation problems conditioned on each finite value of $X$. We formalize this in the following Corollary.

**Corollary 1.** *Let $X$ be distributed uniformly over a finite set of categorical values $\mathcal{X}$, letting $m = |\mathcal{X}|$. Suppose that a dataset $\{(x_i, y_i)\}_{i=1}^n$ is created by sampling $\{y_j\}_{j=1}^{\frac{n}{m}}$ values IID from the conditional distribution $Y|X = x$ for each value of $x$ (assuming $m$ divides $n$), and taking the union of these sets.*

*Let $f : \mathcal{X}, (0,1), \Theta \to \mathbb{R}$ be fully separable over $x$: that is, let $\Theta = \Theta_1 \times ... \times \Theta_m$, where $\Theta_x$ represents a copy of a given parameter space $\Theta_0$ for each value of $x \in \mathcal{X}$. Let $f$ take the form $f(x, \tau; \theta) = \sum_{x_0 \in \mathcal{X}} \mathbb{1}(x = x_0) g(\tau; \theta_{x_0})$ for some function $g : (0,1), \Theta_0 \to \mathbb{R}$.*

*Suppose $f(x, \tau; \theta)$ is well specified with a unique $\theta^* \in \Theta \subseteq \mathbb{R}^m$ such that $f(x, \tau; \theta^*) = q_\tau(x)$ for all $\tau \in (0,1)$ and all $x \in \mathcal{X}$, and that $\theta^*$ is also a unique minimizer for the risk $R(\theta) = \frac{1}{m} \sum_{x \in \mathcal{X}} E_{Y, \mathcal{T}}[L_{\mathcal{T}}(Y, f(x, \mathcal{T}; \theta)) | X = x]$. Suppose the estimator $\hat{\theta}^{(n)}$ is weakly consistent, $\hat{\theta}^{(n)} \xrightarrow{P} \theta^*$. Suppose further that the function $\theta \mapsto f(x, \tau; \theta)$ is continuous, locally Lipschitz, and twice differentiable at $\theta = \theta^*$ for all $\tau \in (0,1)$ and all $x \in \mathcal{X}$. Then for each $x \in \mathcal{X}$,*

$$\sqrt{n/m}(f(x, \tau; \hat{\theta}^{(n)}) - f(x, \tau; \theta^*)) \xrightarrow{d} \mathcal{N}\left(0, \nabla_\theta f(x, \tau; \theta^*)^\top Q^{-1} V (Q^{-1})^\top \nabla_\theta f(x, \tau; \theta^*)\right)$$

*where $Q = E_{\mathcal{T}}[p_x(f(x, \mathcal{T}; \theta^*))\Gamma(\mathcal{T})], V = E_{\mathcal{T}}[\mathcal{T}(1 - \mathcal{T})\Gamma(\mathcal{T})]$, with $p_x(y)$ being the density of $Y|X = x$ and $\Gamma(\tau) = \nabla_\theta f(x, \tau; \theta^*) \nabla_\theta f(x, \tau; \theta^*)^\top$.*

More generally, function classes with a high degree of flexibility such as neural networks may approach this extreme categorical separability described above. We describe one such flexible function class below which, given enough parameters, can be specified to meet this separability condition for finite categorical features. Of course, computational constraints and generalizability must also be considered.

## 4 DLNs for Legitimate but Flexible Inverse CDFs

We propose using DLNs (You et al., 2017) as the function class in (1) because of their flexibility and because they efficiently enable three key regularization strategies: non-crossing constraints, restricting the learned distribution to location-scale families, and monotonic features, as explained below.

### 4.1 Review of Deep Lattice Networks

~~Deep lattice networks~~ DLNs are multi-layer models where some layers are lattices or ensembles of lattices. Lattices build upon a truly ancient strategy for approximating functions over a bounded domain (Campbell-Kelly et al., 2003): record the function's values on a regular grid of inputs, and then one can linearly interpolate between the recorded values. Generalizing to multiple dimensions, a lattice function is a multi-dimensional look-up table whose parameters are the look-up table values, and the function is produced by linearly interpolating the look-up table values surrounding an input point. See Fig. 3 for example 2D lattices.

In the special case of a one-dimensional domain, a lattice is just a ~~PLF. This strategy is very old, but good papers to understand~~ piecewise-linear function. We recommend reading Gupta et al. (2016) for the basics of ~~lattice functions are Garcia et al. (2012) for the spline perspective, and Gupta et al. (2016) for the machine learning perspective (which~~ lattices with monotonicity constraints. See also Garcia et al. (2012) for basics of lattice models and their relationship to other splines.

Mathematically, a lattice function $f(x) : \mathbf{R}^D \to \mathbf{R}$ can be expressed $f(x) = \theta^T \psi(x)$, where the parameters $\theta \in \mathbf{R}^p$ are the look-up table values, and the nonlinear transformation $\psi(x) \in \mathbf{R}^p$ maps an input $x \in \mathbf{R}^D$ to the ultra-sparse vector of linear interpolation weights over the look-up table values.

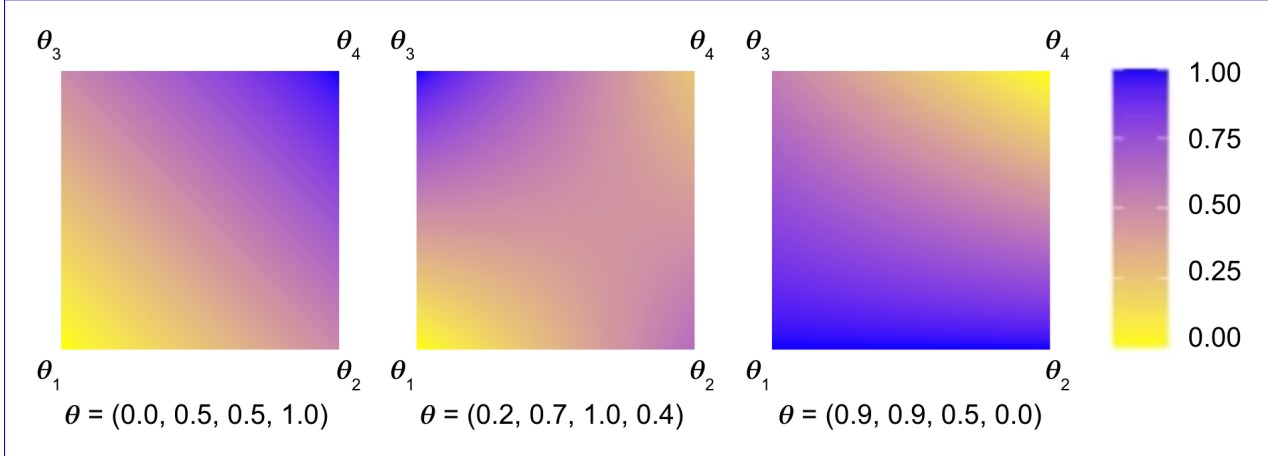

Figure 3: Example 2D lattices sized $2 \times 2$ have four parameters. Each parameter represents the value the lattice function takes at one of the four corners of the input domain $[0,1]^2$. The lattice function values in-between the vertices are computed by bilinear interpolation of the four parameter values, so these functions can be re-parameterized and expressed as bilinear polynomial functions over $[0,1]^2$ of the form $f(x) = a + bx_1 + cx_2 + dx_1x_2$. A 2D lattice with more knots, e.g. $3 \times 3$, would be locally bilinear across each cell of the lattice.

Garcia & Gupta (2009) showed that lattice models can be trained using a standard empirical risk minimization framework. Lattices can approximate any continuous bounded function if they are sampled on a regular grid with enough knots.

A key reason to use lattice functions in an application like quantile regression is that the regular structure of their parameters makes them amenable to shape constraints. Here, the constraint (2) requires $f(x, \tau; \theta)$ to be monotonically increasing in $\tau$, which is imposed efficiently with sparse linear inequality constraints on the lattice parameters corresponding to forcing any neighboring gridpoints in the direction of $\tau$ to be increasing (Gupta et al., 2016). To more efficiently represent functions, lattice models are usually architected with a first layer that separately calibrates each input with a learned one-dimensional piecewise linear function (PLF) whose knots need-not be regularly-spaced referred to as a calibrator, then fuses the calibrated inputs together with a coarser multi-dimensional lattice (Garcia et al., 2012; Gupta et al., 2016).

The main drawback to lattices is that the number of lattice parameters needed to approximate a $D$-dimensional functions blows up exponentially as $O(2^D)$. This problem is handled by making an ensemble of calibrated lattices (~~Canini et al., 2016; Gupta et al., 2018; 2020~~ (Canini et al., 2016) much as one makes a random forest, and these calibration and lattice layers can be mixed with other layers like linear layers, and cascaded ad infinitum, forming arbitrarily *deep lattice networks* (DLNs) (You et al., 2017; Cotter et al., 2019). Importantly, as long as each layer of a DLN is trained to respect a monotonic feature like $\tau$, the entire DLN will be monotonic in $\tau$.

More generally, one could use unrestricted ReLU or embedding layers for the first few model layers on $x$, then fuse in $\tau$ later with $\tau$-monotonic DLN layers. Figure 4 illustrates how an arbitrary neural net architecture can be used to reduce the dimensionality of the input features of $x$ before fusing with $\tau$, allowing our approach to be used for essentially any type and size of input.

~~Our DLN's~~

In our experiments, our DLNs are built with the standard two-layer calibrated lattice (Gupta et al., 2016) and calibrated-ensemble-of-lattices (Canini et al., 2016) architectures offered by the open-source TensorFlow Lattice library (TensorFlow Blog Post, 2020). The TensorFlow Lattice library enables solving the constrained optimization problem (1) and (2) with Dykstra's projection algorithm. In our experiments, we found our DLN models trained in a similar amount of time as the DNN and SQF-DNN models.

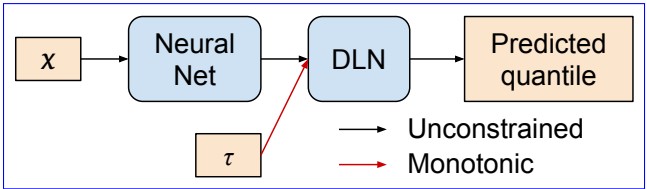

Figure 4: Block diagram showing how an arbitrary neural net architecture can be used to reduce the dimensionality of $x$ to a more manageable size, before combining in a DLN architecture with $\tau$, which is constrained to be monotonic.

## 4.2 Location-Scale Family Models

We show that by different architecture choices for the DLN model used can regularize the estimated inverse CDF in semantically-meaningful ways, similar to prior more rigid distributional approaches. As a simple example, if one constrains the DLN architecture to not have any interactions between $\tau$ and any of the $x$ features, then one learns a regression model with homoskedastic errors. As another example, the basic two-layer DLN called a *calibrated lattice* model (Gupta et al., 2016) described in Section 4.1 can be constrained to learn distributions across $x$ that come from a shared, learned, location-scale family:

~~**Lemma:**~~

**Lemma 2.** Let $f(x, \tau)$ be a calibrated lattice model with piece-wise linear calibrator $c(\tau) : [0, 1] \to [0, 1]$ for $\tau$, and let there be two knots in the regular grid in the direction of $\tau$, and the lattice parameters are interpolated with standard multilinear interpolation. Then $f(x, \tau)$ represents an inverse CDF function $F^{-1}(y|x)$ where the estimated distribution for every $x$ is from the same location-scale family as the calibrator $c(\tau)$.

The proof is in the Appendix. Note the number of keypoints in $c(\tau)$ controls the complexity of the learned base distribution, allowing the model to approximate location-scale families like the Gaussian, gamma, or Pareto distributions. Then in the second layer of $f$, the number of knots in the lattice over the $c(\tau)$ input controls how much the distribution should be allowed to vary across $x$. Two knots in the lattice limits us to a shared location-scale family, as noted above, while three knots in the lattice gives an extra degree of freedom to shrink or stretch one side of the distribution differently across $x$. More lattice knots across $\tau$, or ensembling of lattices, or more layers can all be used to move the model towards full generality.

## 4.3 Other Monotonic Features

Using DLNs also enables imposing monotonicity on the features $x$ if domain knowledge says they should have a monotonic impact on the estimates (Gupta et al., 2016). For example, a pizza delivery time estimator might treat constrain an input quantifying how bad traffic is to only increase estimated quantiles (that is, worse traffic never makes the estimated delivery time shorter).

Our experience with real-world applications of quantile estimation is that real-world quantile estimates do often use features that are past measurements (or otherwise strong correlates) to predict the future distribution of measurements. For example, in the real-world Puzzles experiment in Section 5, we predict the quantiles of how long a club member will keep a rented puzzle, and one of the features is how long they kept their last puzzle. We constrain that input to have a monotonically positive effect on the estimate. (However, earlier hold-times are not necessarily monotonically indicative of the next future hold-time, because they may indicate a strongly decreasing trend over time). Domain knowledge about the relationship between the model inputs-and-ouput that can be captured with monotonicity constraints is a tuning-free, semantically-meaningful regularizer that can improve test performance and increase model explainability (Gupta et al., 2016), and where applicable, even help capture ethical rules (Wang & Gupta, 2020).

DLNs also enable multi-dimensional shape constraints to capture domain knowledge like complementary features (Gupta et al., 2020), and functions defined on sets (Cotter et al., 2019).

## 5 Experiments

Using simulations and real data, we first show that training a DLN that is monotonic in $\tau$ to satisfy both (1) and (2) works well compared to training a DNN for (1) as done in Tagasovska & Lopez-Paz (2019) or the SQF-DNN parametric approach of Gasthaus et al. (2019); the results are in Tables 3 and 5. Then we use the proposed $\tau$-monotonic DLN models to compare training to minimize expected pinball loss versus training to minimize the pinball loss for a specific $\tau$; the results are in Tables 7 and 8.

Bolded results in tables indicate that the metric is not statistically significantly different from the best metric among the models being compared, using an unpaired t-test.

### 5.1 Model Training Details

All hyperparameters were optimized on validation sets. We used Keras models in TensorFlow 2.2 for the unrestricted DNN comparisons that optimize (1) (Tagasovska & Lopez-Paz, 2019), and the SQF-DNN (Gasthaus et al., 2019), which optimizes the same objective in a different manner while also guaranteeing non-crossing quantile estimates. For DLNs, we used the TensorFlow Lattice library (TensorFlow Blog Post, 2020). For all DNN and DLN experiments, we use the Adam optimizer (Kingma & Ba, 2015) with its default learning rate of 0.001, except where noted. For DNN models, we validated the number of hidden layers and the hidden dimension. For the SQF-DNN, we also validated the number of distribution keypoints. For the smaller DLN models, we used the common two-layer calibrated lattice architecture (Gupta et al., 2016) and validated over its number of calibration keypoints and lattice vertices. For the larger datasets, we used an ensemble of calibrated lattices for the DLN model, and then also validated over the the number and dimensionality of the base models in the ensemble (Canini et al., 2016). For both DLNs and DNNs, we additionally validated over the number of training epochs. ~~Training the different methods considered took roughly equally long.~~

### 5.2 Benchmark and Real Datasets Used

We tested on three publicly available datasets and one proprietary dataset for real-world problems. A co-lab will be made available.

**Air Quality:** The Beijing Multi-Site Air-Quality dataset from UCI (Zhang et al., 2017) contains hourly air quality data from 12 monitoring regions around Beijing. We predict the quantiles of the PM2.5 concentration from $D = 7$ features: temperature, pressure, dew point, rain, wind speed, region, and wind direction. The DLN model is an ensemble of two-layer calibrated lattice models. We split the data by time (not IID) with earlier examples forming a training set of size 252,481, later examples a validation set of size 84,145, and most recent examples a test set of size 84,145.

**Puzzles:** This is a dataset from *(name withheld for blind review)*, a private library for wooden jigsaw puzzles. We predict how long a library member will hold a borrowed puzzle given their $D = 5$ previous hold-times. The DLN model is an ensemble of two-layer calibrated lattice models. Based on their domain knowledge, we also constrain the DLN output to be monotonically increasing in the most-recent-past hold-time feature. The 936 train and 235 validation examples are IID from past data, while the 210 test samples are the most recent samples (not IID). The anonymized dataset is publicly available on their website *(website withheld for blind review)*.

**Traffic:** This is a proprietary dataset from a large internet services company *(name withheld for blind review)* for estimating the time to drive a route. The DLN is a two-layer calibrated lattice whose $D = 4$ inputs are 1 categorical and 3 continuous features. We used 1,000 examples each for training, validation, and testing, with the training examples occurring earlier in time than the validation and test examples (not IID).

**Wine:** We used the Wine Reviews dataset from Kaggle (Bahri, 2018). We predict the quantiles of wine quality on a 100-point scale. We used $D = 42$ features, including price, a 1-d learned embedding for the country of origin (aka categorical calibration (Gupta et al., 2016)), and 40 Boolean features describing each wine. The DLN model is an ensemble of two-layer calibrated lattice models. The DLN output is constrained to be monotonically increasing in the *price* feature, as suggested in prior work (Gupta et al., 2018). The data was split IID with 84,641 examples for training, 12,091 for validation, and 24,184 for testing.

### 5.3 Model Architecture Experiments

We start by demonstrating the efficacy of using monotonic DLNs to predict the inverse CDF on simulations, and then on the real data.

#### 5.3.1 Simulations

We used simulations from a recent quantile regression survey paper (Torossian et al., 2020) based on the sine, Griewank, Michalewicz, and Ackley functions with carefully designed noise distributions to represent a range of variances and skews across the respective input domains. We used 250 training examples for the 1-D sine and Michalewicz functions, 1,000 examples for the 2-D Griewank function, and 10,000 examples for the 9-D Ackley function.

Our metric is the average squared difference between the estimated and true quantile curves sampled at 99 uniform quantiles $\tau \in 0.01, \ldots, 0.99$, averaged over 100 repeats, and averaged over values of $x$ across the domain (we use fine grids of 1,000 and 10,000 $x$ values for the 1-D and 2-D experiments, respectively, and a random sample of 10,000 $x$-values for the 9-D experiment). We also report the fraction of test points for which at least two of their 99 quantiles crossed.

Results in Table 3 demonstrate that the proposed monotonic-in-$\tau$ DLNs are the best or statistically tied for the best across all four disparate simulations. The DNNs were trained with the same sampled expected pinball loss, and are sometimes close in performance, but suffer substantially from crossing quantiles. For example, on the sine-skew distribution, quantile-crossing was observed for the DNN model between at least two quantiles on 38% of test $x$ values!

Spline quantile functions (Gasthaus et al., 2019) avoid crossing by construction, but their accuracy was much worse than the DLN accuracy on three of the four datasets, and not better on the fourth. In particular, the DLNs do much better at the Griewank and Ackley simulations, which have the lowest signal-to-noise ratios (Torossian et al., 2020).

Table 3: Simulations: Quantile MSE and percent crossing violations for $\tau \in \{0.01, 0.02, \ldots, 0.99\}$.

|  | Sine-skew (1,7) | Griewank | Michalewicz | Ackley |
|---|---|---|---|---|
| Model | MSE, *Crossing* | MSE, *Crossing* | MSE, *Crossing* | MSE, *Crossing* |
| DNN | **3.53 ± 0.09**, *38%* | 1.29 ± 0.02, *5%* | 0.311 ± 0.011, *12%* | 237 ± 6, *0.3%* |
| SQF-DNN | 5.68 ± 0.11, *0%* | 1.05 ± 0.01, *0%* | **0.232 ± 0.006**, *0%* | 667 ± 81, *0%* |
| DLN | **3.51 ± 0.13**, *0%* | **0.55 ± 0.01**, *0%* | **0.219 ± 0.006**, *0%* | **206 ± 2**, *0%* |

### 5.4 Training Time Comparison

We also provide timing results in Table 4 for the Ackley simulation, which as a complex 9-dimensional problem required non-trivial training times. We compared timings of the different models with their validated hyperparameters. The validated DNN chose half the number of training batches as the DLN, and this caused it to be twice as fast. Each training batch took roughly the same amount of time though, and more granular hyperparameter tuning of the number of training batches could have led to different results. The SQF-DNN, meanwhile, is noticeably slower to train. However, while these implementations were comparable in their use of software and processors, all of these approaches could certainly be better optimized for speed. For example, compiler-optimized DLN models can execute ten times faster than an optimized C++ implementation, which was 100 times faster than a TensorFlow implementation (Zhang et al., 2021).

#### 5.4.1 Real Data

Table 5 compares these models on the four real datasets. We report the pinball loss averaged over ~~$\tau in \{0.01, 0.02, \ldots, 0.99\}$~~ $\tau \in \{0.01, 0.02, \ldots, 0.99\}$, which is a standard metric for judging conditional quantile estimates, and one which approximates the *continuous ranked probability score* metric used for measuring the

Table 4: Simulations: Training time, in seconds, for the validated model of each type on the Ackley simulation from Table 3. We also report the number of training steps used by each model (i.e. the number of batches seen by the model) and the time taken per training step.

| Model | Train Time (s) | Training Steps | Time Per Batch (ms) |
|---|---|---|---|
| DNN | 3446 | 500000 | 6.9 |
| SQF-DNN | 28042 | 100000 | 280 |
| DLN | 8444 | 1000000 | 8.4 |

Table 5: Real data experiments: Pinball loss on the test set, averaged over $\tau \in \{0.01, 0.02, \ldots, 0.99\}$.

| Model | Air Quality | Puzzles | Traffic | Wine |
|---|---|---|---|---|
| DNN | $18.041 \pm 0.088$ | $3.253 \pm 0.024$ | $0.0494 \pm 0.00037$ | $0.6603 \pm 0.0088$ |
| SQF-DNN | $\mathbf{17.356 \pm 0.106}$ | $\mathbf{3.108 \pm 0.044}$ | $0.0491 \pm 0.00022$ | $0.6552 \pm 0.0037$ |
| DLN | $\mathbf{17.280 \pm 0.157}$ | $\mathbf{3.092 \pm 0.020}$ | $\mathbf{0.0479 \pm 0.00003}$ | $\mathbf{0.6381 \pm 0.0005}$ |

quality of probabilistic forecasts (Gasthaus et al., 2019; Yan et al., 2018). The $\tau$-monotonic DLNs performed the best or statistically similar to the best on three of the four problems, and was statistically significantly better than the DNNs trained with the same expected pinball loss for all four problems. Unlike the simulation results in Table 3, the SQF-DNN (monotonic by construction) performs notably better on these real datasets than the DNN in Table 5, but is never better than the DLN.

We also report the *absolute deviation calibration error* (Chung et al., 2021) in Table 6 (some authors use a squared variant (Kuleshov et al., 2018)), the formula is given in A.8. By that metric the DLN is stat. sig. better in 3 of the 4 cases than SQF-DNN, with substantial wins over SQF-DNN on Traffic and Wine.

Table 6: Real data experiments: Calibration on the test set, averaged over $\tau \in \{0.01, 0.02, \ldots, 0.99\}$.

| Model | Air Quality | Puzzles | Traffic | Wine |
|---|---|---|---|---|
| DNN | $0.0761 \pm 0.0038$ | $0.0826 \pm 0.0031$ | $0.0562 \pm 0.0014$ | $0.0253 \pm 0.0010$ |
| SQF-DNN | $0.0798 \pm 0.0042$ | $\mathbf{0.0547 \pm 0.0031}$ | $0.0600 \pm 0.0062$ | $0.0671 \pm 0.0083$ |
| DLN | $\mathbf{0.0588 \pm 0.0057}$ | $0.0681 \pm 0.0052$ | $\mathbf{0.0210 \pm 0.0027}$ | $\mathbf{0.0125 \pm 0.0031}$ |

### 5.5 Expected Pinball Loss Experiments

We next empirically show that training a full inverse CDF by minimizing the expected pinball loss is at least as good at estimating specific quantiles as training with the pinball loss just for those specific quantiles.

### 5.5.1 Simulations

We ran simulations on the 1D sine-skew function as per Torossian et al. (2020), with three noise scenarios illustrated in Figure 5: noise parameters $(a, b) = (1, 1)$ for symmetric low-noise, $(7, 7)$ for symmetric high-noise, and $(1, 7)$ for asymmetric high-noise~~(see Fig. 5 in the Appendix).~~ These simulations are the simplest extension of the unconditioned context studied empirically and theoretically in Section 3 to the conditioned case, in that there is a single $x$ feature, and enough data for a flexible quantile regression model to approximate the corresponding sample quantile of $y$ within a small neighborhood of any value of $x$.

The results shown in Table 7 are the error in predicting the median given either $N = 100$ or $N = 1,000$ training samples. ~~For both symmetric-noise cases $(1,1)$ and $(7,7)$, the model trained with an~~ Overall, the ~~single pinball loss model struggles relative to the~~ expected pinball loss ~~produces much more accurate median estimates. This remains true~~ models, even as we increase the number of data points from 100 to 1000, at which point there are about 50 data points within each 0.1 region of $x \in [-1, 1]$.

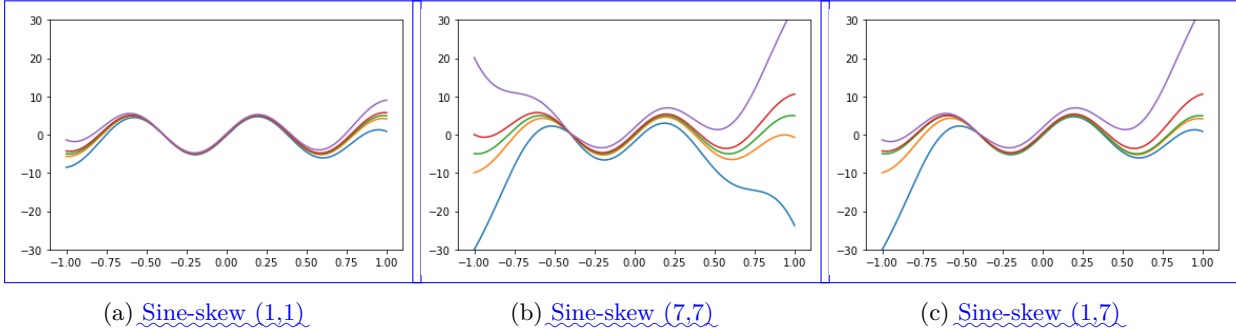

(a) Sine-skew (1,1)          (b) Sine-skew (7,7)          (c) Sine-skew (1,7)

Figure 5: The colored lines show the true quantiles for $\tau = 0.1, 0.4, 0.5, 0.6, 0.9$ for the sine-skew distribution simulations Torossian et al. (2020).

Table 7: Sine-skew experiment with different sine-skew noise choices (1,1), (7,7) and (1,7). We present results for DLNs trained with four different distributions over $\tau$: the uniform distribution, the beta distribution with a mode of 0.5 and concentration of 10, the beta distribution with a mode of 0.5 and concentration of 100, and just the 0.5 pinball loss. The metric is the MSE between the true median and estimated median, averaged over $x \sim \text{Unif}(-1, 1)$.

| $(a, b)$ | $N$ | min $E_{\mathcal{T}}$, Unif | min $E_{\mathcal{T}}$, Beta(10) | $E_{\mathcal{T}}$, Beta(100) | min single-$\tau$ loss |
|---|---|---|---|---|---|
| $(1, 1)$ | 100 | $0.421 \pm 0.025$ | $0.348 \pm 0.015$ | $\mathbf{0.301 \pm 0.016}$ | $0.588 \pm 0.036$ |
| $(1, 1)$ | 1,000 | $\mathbf{0.050 \pm 0.003}$ | $0.057 \pm 0.002$ | $0.056 \pm 0.002$ | $0.067 \pm 0.003$ |
| $(7, 7)$ | 100 | $\mathbf{7.334 \pm 0.367}$ | $6.654 \pm 0.330$ | $7.402 \pm 0.397$ | $9.860 \pm 0.560$ |
| $(7, 7)$ | 1,000 | $\mathbf{0.967 \pm 0.058}$ | $0.861 \pm 0.036$ | $1.057 \pm 0.037$ | $1.470 \pm 0.086$ |
| $(1, 7)$ | 100 | $4.458 \pm 0.259$ | $3.515 \pm 0.178$ | $\mathbf{3.031 \pm 0.179}$ | $4.416 \pm 0.386$ |
| $(1, 7)$ | 1,000 | $0.451 \pm 0.037$ | $0.336 \pm 0.022$ | $\mathbf{0.338 \pm 0.020}$ | $0.542 \pm 0.067$ |

The uniform pinball loss model which trains equally across all quantiles performs quite well in both symmetric-noise cases $(1, 1)$ and $(7, 7)$. However, for the simulation with highly skewed noise $(1, 7)$ it ~~is less useful to smooth across $\tau$, and the expected and~~ struggles a bit relative to the models trained with a Beta pinball loss, perhaps because the extreme quantiles start becoming very uninformative about the median. Still, even in that case, it performs similarly to the single pinball loss models~~perform similarly~~.

### 5.5.2 Real Data

In Table 8 we give the accuracy at predicting three specific target quantiles on the four real datasets by either training a full inverse CDF by minimizing $E_{\mathcal{T}}$ where $\mathcal{T}$ is drawn uniformly from $(0, 1)$; or training one model to minimize the discrete sum of the three pinball losses for the $\tau$s considered; or training three separate models that minimize $E_{\mathcal{T}}$ where $\mathcal{T}$ is concentrated around the target quantiles; or training three separate models that minimize each specific $\tau$'s pinball loss.

The test metric is computed on finite test sets, so the results are not as clean as the simulations where we can compare to the true quantiles.

The inverse CDF model is statistically significantly tied or better in every case for the median $\tau = 0.5$, which is the quantile we expected the smoothing of the expected pinball loss to be most helpful. We hypothesize that the win in median accuracy for Puzzles is due to the small size of the train set and high randomness of the truth, making that problem most helped by the regularization of $\tau$-smoothing. Even for the more extreme quantiles of $\tau = 0.7$ and $\tau = 0.9$, the full inverse CDF model is statistically significantly better than training for those specific quantiles (though for $\tau = 0.9$, the model trained for three discrete quantiles is even better).

Table 8: Accuracy of single quantile estimates, comaparing one model trained to minimize the expected pinball loss with respect to uniform $P_{\mathcal{T}}$; one model trained to minimize the sum of the three pinball losses for the three quantiles shown (Discrete); three separate models trained with individual beta distributions $P_{\mathcal{T}}$ with the mode set at the target quantile and the beta concentration chosen on the validation set; and three separate models trained with individual $\tau$ pinball losses (Single). All models were DLNs with the same architecture as in Table 5, and monotonic on $\tau$ and any input features noted in the text.

| Air Quality: | Model | Pinball loss ($\tau = 0.5$) | Pinball loss ($\tau = 0.9$) | Pinball loss ($\tau = 0.99$) |
|---|---|---|---|---|
| | $\tau \sim \mathrm{Unif}(0,1)$ | **23.576 ± 0.047** | 14.850 ± 0.075 | 2.947 ± 0.025 |
| | $\tau \sim$ Discrete | 24.083 ± 0.073 | 15.439 ± 0.062 | 3.042 ± 0.012 |
| | $\tau \sim$ Beta | **23.586 ± 0.183** | **13.942 ± 0.054** | **2.709 ± 0.029** |
| | Single $\tau \in \mathbf{T}$ | **23.634 ± 0.068** | 14.908 ± 0.092 | **2.700 ± 0.013** |
| **Traffic:** | Model | Pinball loss ($\tau = 0.5$) | Pinball loss ($\tau = 0.9$) | Pinball loss ($\tau = 0.99$) |
| | $\tau \sim \mathrm{Unif}(0,1)$ | **0.064386 ± 0.00014** | 0.040159 ± 0.00018 | **0.010645 ± 0.00018** |
| | $\tau \sim$ Discrete | 0.064578 ± 0.00028 | **0.039804 ± 0.00014** | 0.011290 ± 0.00014 |
| | $\tau \sim$ Beta | 0.064988 ± 0.00030 | **0.039549 ± 0.00009** | **0.01064 ± 0.00011** |
| | Single $\tau \in \mathbf{T}$ | **0.064791 ± 0.00012** | 0.039900 ± 0.00010 | **0.01070 ± 0.00018** |
| **Wine:** | Model | Pinball loss ($\tau = 0.1$) | Pinball loss ($\tau = 0.5$) | Pinball loss ($\tau = 0.9$) |
| | $\tau \sim \mathrm{Unif}(0,1)$ | 0.4099 ± 0.0006 | **0.8889 ± 0.0017** | 0.3773 ± 0.0019 |
| | $\tau \sim$ Discrete | 0.4094 ± 0.0005 | 0.8891 ± 0.0010 | 0.3706 ± 0.0003 |
| | $\tau \sim$ Beta | **0.4047 ± 0.0005** | **0.8871 ± 0.0008** | **0.3665 ± 0.0004** |
| | Single $\tau \in \mathbf{T}$ | **0.4049 ± 0.0004** | **0.8867 ± 0.0006** | **0.3663 ± 0.0003** |
| **Puzzles:** | Model | Pinball loss ($\tau = 0.5$) | Pinball loss ($\tau = 0.7$) | Pinball loss ($\tau = 0.9$) |
| | $\tau \sim \mathrm{Unif}(0,1)$ | **4.173 ± 0.013** | **4.219 ± 0.021** | 2.705 ± 0.029 |
| | $\tau \sim$ Discrete | 4.359 ± 0.014 | **4.204 ± 0.011** | **2.614 ± 0.010** |
| | $\tau \sim$ Beta | 4.318 ± 0.013 | 4.277 ± 0.018 | 2.681 ± 0.011 |
| | Single $\tau \in \mathbf{T}$ | 4.293 ± 0.013 | 4.350 ± 0.022 | 2.708 ± 0.012 |

We expected the $E_\mathcal{T}$ loss to be least useful for extreme quantiles, since the smoothing will be more one-sided. Indeed, the wins for the single-quantile models are on more extreme quantiles: the 99th percentile for Air Quality, and the 10th and 90th percentile for Wine. For Wine, we hypothesize this is because it is a fairly easy regression problem with a large number of training samples and one continuous feature (wine price) that correlates highly with the label (wine quality), so the extra regularization is not helping. But for the large Traffic dataset, even the 99th percentile is statistically tied, and the single quantile model is worse even at the extreme quantiles for Puzzles. Interestingly, between the two single-quantile modeling approaches, training with $\tau$ sampled from a Beta distribution is statistically similar or better than training with a single pinball loss in every case. Overall, the results in Table 8 show that the full inverse CDFs are at least as good as the single-quantile models.

## 6 Conclusions And Some Open Questions

We gave theoretical and empirical evidence that training quantile regression models by minimizing the expected pinball loss can perform as well or better on any individual quantile than models trained for that quantile alone. We showed that minimizing the expected pinball loss is not sufficient to provide legitimate inverse CDF estimates, one must also produce a model whose quantiles are guaranteed to not cross. We showed that DLN models can be effectively constrained to be monotonic in $\tau$ and produce state-of-the-art quantile estimates. The non-crossing issue is important not just for theoretical reasons, but because it ~~will be confusing to any non-expert end-user~~ is confusing to many non-experts if a model's quantile estimates cross, eroding trust in that model, and by association, all AI.

~~A key open~~ We provide a new asymptotic convergence rate for estimation of the full inverse CDF, assuming an oracle that minimizes the empirical loss. We leave open the question of a non-asymptotic convergence rate, and whether some of our assumptions can be relaxed, or whether better convergence rates can be had with some more stringent assumptions. Theorem 1 is for unconditioned quantile estimation, with a simple extension to separable conditional quantile estimation in Corollary 1. Important open work would be providing stronger results for conditional quantile estimation where there is smoothing across the features.

A key question is the choice of distribution of $\mathcal{T}$ when minimizing the expected pinball loss. ~~Is~~ We note one can control the regularization across quantiles both by the shape of $P_\mathcal{T}$ and by the model flexibility with respect to $\tau$. Thus we showed a uniform distribution ~~(as done here and in Tagasovska & Lopez-Paz (2019)) most efficient? If~~ $P_\mathcal{T}$ generally provides good results if one is validating the model flexibility over $\tau$, and a uniform distribution has no hyperparameters and gives you a good estimate of the entire inverse CDF for free. However, if one is only interested in a single $\tau$ ~~would a distribution more peaked around that~~, we also showed that one may do better with a beta distribution for $P_\mathcal{T}$, and that a concentration of 10 likely makes a good default choice, but for best results one would want to validate the concentration.

One open question is whether the proposed methodology improves the accuracy of prediction intervals (Pearce et al., 2018), as measured by traditional metrics such as interval width and calibration. Tagasovska & Lopez-Paz (2019) have shown that minimizing the expected pinball loss can be a useful strategy in that regard, and our results in this paper about improved quantile accuracy and calibration suggest there may be improved performance in prediction intervals as well.

Our experiments, like most published quantile regression experiments, were limited to datasets with $D = 42$ features and less. This is partly because one needs a large amount of data to make decent estimates of tail quantiles. In theory the proposed methodology can be applied to any size problem by only bringing $\tau$ ~~produce better results ?~~ in for the last layers as shown in Fig. 4, but an open question is how well this or any quantile regression works in practice with a large number of features, and whether there are novel scaling issues.

In particular, we hypothesize that for larger models there may be an increase in *quantile collapse* (Lopez-Martin et al., 2021). Quantile collapse is the edge case of non-crossing, defined as a flat $f(\tau)$ over some range of $\tau$. For example, the airline tells passengers there is an 80% chance the flight will arrive in 4 hours and 6 minutes, and that there is a 90% chance the flight will arrive in 4 hours and 6 minutes. Our

monotonicity constraint (2) allows quantile collapse because it only requires monotonicity in $\tau$, not *strict* monotonicity.

Quantile collapse can happen in regions of sparse data where there are no training samples to fit to such that the model only has to satisfy (2), which it can do with a flat $f(x, \tau)$. We belive this can partly be solved by initializing with a $f(x, \tau)$ that is strictly monotonically increasing (as we do in our experiments), as then one only gets quantile collapse if there is enough relevant training data to pull the model to the edge of the feasibility of (2).

Quantile collapse is also a risk if the model $f(x, \tau)$ has too much flexibility in $x$ such that the data is sparse with respect to that flexibility. For example, assume $x$ is a categorical feature with 1 million possible values (e.g. a language model), and that $f(x, \tau)$ is flexible enough to learn an independent estimate for each value of $x$. Suppose for one of those categorical values there is only one training sample $(x_i, y_i)$. Then since the pinball loss (expected or standard) will be minimized when $f(x, \tau) = y_i$, there is no $\tau$ dependence, and for that category you will have quantile collapse. More generally, if you have a very flexible model and sparse data in some regions, we expect you may get quantile collapse if no other regularization to avoid it is used.

**Broader Impact Statement**

We hope the broader impact of this work to be positive in increasing the deserved trustworthiness of AI by promoting quantile regression models with no crossing-quantiles. The alternative of quantile estimates that cross will undoubtedly erode public confidence in AI. For example, prior work would launch a quantile regression model that predicts there is a 90 percent chance your pizza will be delivered in 30 minutes, and that there is an 80 percent chance it will be delivered in 34 minutes. Such models are unnecessarily confusing and untrustworthy for non-experts, and using them in practice may then reduce the ability of worthy AI to have a positive impact on the world.

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

## A  Appendix

### A.1  Proof of Lemma 2

~~Proof:~~

**Lemma 2**. *Let $f(x, \tau)$ be a calibrated lattice model with piece-wise linear calibrator $c(\tau) : [0, 1] \to [0, 1]$ for $\tau$, and let there be two knots in the regular grid in the direction of $\tau$, and the lattice parameters are interpolated with standard multilinear interpolation. Then $f(x, \tau)$ represents an inverse CDF function $F^{-1}(y|x)$ where the estimated distribution for every $x$ is from the same location-scale family as the calibrator $c(\tau)$.*

*Proof.* If a random variable $Y$ conditioned on $X$ belongs to the location-scale family then for $\tau \in (0, 1)$ and $a \in \mathbb{R}$ and $b > 0$, it must hold that the conditional inverse CDF satisfies $F_{Y|X=z}^{-1}(\tau) = a + b F_{Y|X=x}^{-1}(\tau)$. Note that for $\tau \in (0, 1)$, interpolating a lattice with two lattice vertices in the $\tau$ dimension yields the estimate $\hat{F}_{Y|X=z}^{-1}(\tau) = f(z, \tau) = f(z, 0) + c(\tau)(f(z, 1) - f(z, 0))$. Thus mapping to the location-scale property, $a = f(z, 0)$, $b = f(z, 1) - f(z, 0)$, and $\hat{F}_{Y|X=x}^{-1}(\tau) = c(\tau)$. Thus every estimated conditional inverse CDF $\hat{F}_{Y|X=z}^{-1}(\tau)$ is a translation and scaling of the piecewise linear function $c(\tau)$. $\qquad\square$

### A.2  Proof of Theorem 1

**Theorem 1**. *Suppose $f(\tau; \theta)$ is well specified ~~: there exists $\theta^* \in \mathbb{R}^m$~~ with a unique $\theta^* \in \Theta \subseteq \mathbb{R}^m$ such that $f(\tau; \theta^*) = F^{-1}(\tau)$ for all $\tau \in (0, 1)$~~. Suppose~~, and let $\mathcal{T}$ be distributed such that $\theta^*$ is also a unique minimizer for the risk $R(\theta) = E_{Y, \mathcal{T}}[L_{\mathcal{T}}(Y, f(\mathcal{T}; \theta))]$. Suppose the estimator $\hat{\theta}^{(n)}$ is weakly consistent,*

$\hat{\theta}^{(n)} \xrightarrow{P} \theta^*$. *Suppose further that the function $\theta \mapsto f(\tau; \theta)$ is continuous, locally Lipschitz, and twice differentiable at $\theta = \theta^*$ for all $\tau \in (0, 1)$. Then,*

$$\sqrt{n}(f(\tau; \hat{\theta}^{(n)}) - f(\tau; \theta^*)) \xrightarrow{d} \mathcal{N}\left(0, \nabla_\theta f(\tau; \theta^*)^\top Q^{-1} V (Q^{-1})^\top \nabla_\theta f(\tau; \theta^*)\right)$$

*where,*

$$Q = E_\mathcal{T}[p(f(\mathcal{T}; \theta^*))\Gamma(\mathcal{T})], V = E_\mathcal{T}[\mathcal{T}(1 - \mathcal{T})\Gamma(\mathcal{T})],$$

*with $\Gamma(\tau) = \nabla_\theta f(\tau; \theta^*) \nabla_\theta f(\tau; \theta^*)^\top$.*

*Proof.* Consider the loss function $\bar{L}(y, \theta) = E_\mathcal{T}[L_\mathcal{T}(y, f(\mathcal{T}; \theta))]$. ~~Suppose that the risk $R(\theta) = E_Y[\bar{L}(Y, \theta)]$ has a unique minimizer $\theta^* = \arg\min_\theta R(\theta)$.~~ Let $\hat{\theta}^{(n)}$ minimize the risk with an empirical distribution over $Y$ with IID samples $\{y_i\}_{i=1}^n$, ~~$\hat{\theta}^{(n)} = \arg\min_\theta \frac{1}{n}\sum_{i=1}^n \bar{L}(y_i, \theta)$. Suppose further~~ $\hat{\theta}^{(n)} = \arg\min_\theta \frac{1}{n}\sum_{i=1}^n \bar{L}(y_i, \theta)$. We assume that $Y$ is a continuous random variable with density $p(y)$, and $\mathcal{T}$ is independent of $Y$.

We apply Theorem 5.23 from van der Vaart (1998)[1], which says that under some regularity conditions,

$$\sqrt{n}(\hat{\theta}^{(n)} - \theta^*) = -\nabla^2 R(\theta^*)^{-1} \cdot \frac{1}{\sqrt{n}} \sum_{i=1}^n \nabla_\theta \bar{L}(y_i, \theta^*) + o_P(1),$$

where $o_P(1)$ refers to a sequence of random variables that converges in probability to 0: $X_n = o_P(1)$ if and only if $X_n \xrightarrow{P} 0$. Specifically, ~~this theorem~~ Theorem 5.23 from van der Vaart (1998) requires the following conditions:

(i) ~~$\hat{\theta}^{(n)} \xrightarrow{P} \theta^*$: true under the central limit theorem.~~ $\hat{\theta}^{(n)}$ is weakly consistent for $\theta^*$: $\hat{\theta}^{(n)} \xrightarrow{P} \theta^*$. This is assumed, but in general this consistency depends on properties of the function $f$ and the feasible parameter space $\Theta$. See Lemma 1 for an example.

(ii) $\frac{1}{n}\sum_{i=1}^n \bar{L}(y_i, \hat{\theta}^{(n)}) \leq \inf_\theta \frac{1}{n}\sum_{i=1}^n \bar{L}(y_i, \theta) + o_P(1/n)$: true by optimality of $\hat{\theta}^{(n)}$.

(iii) The loss $\bar{L}(y, \theta)$ is locally Lipschitz near $\theta = \theta^*$. This holds as long as $f(\tau; \theta)$ is locally Lipschitz near $\theta = \theta^*$.

(iv) For $P$-almost all $y$, the function $\theta \mapsto \bar{L}(y, \theta)$ is differentiable at $\theta = \theta^*$.

(v) $R(\theta)$ is twice differentiable at $\theta^*$ with positive definite Hessian $\nabla^2 R(\theta^*) \succ 0$.

To verify condition (iv), note that

$$\nabla_\theta \bar{L}(y, \theta^*) = \nabla_\theta E_\mathcal{T}[L_\mathcal{T}(y, f(\mathcal{T}; \theta^*))] = E_\mathcal{T}[\nabla_\theta L_\mathcal{T}(y, f(\mathcal{T}; \theta^*))].$$

Since $Y$ is a continuous random variable with a density near $f(\tau; \theta^*)$, $\nabla_\theta L_\tau(y, f(\tau, \theta^*))$ exists for $P$-almost every $y$ as long as $f(\tau, \theta)$ is also differentiable at $\theta = \theta^*$. This is because for continuous $Y$, the $\tau$-quantile is unique, and the derivative $\nabla_\gamma L_\tau(y, \gamma)$ exists with probability 1. The full gradient is:

$$\nabla_\theta L_\tau(y, f(\tau; \theta^*)) = ((1 - \tau)\mathbb{1}(f(\tau; \theta^*) \geq y) - \tau\mathbb{1}(f(\tau; \theta^*) \leq y))\nabla_\theta f(\tau; \theta^*).$$

Therefore, $\nabla_\theta \bar{L}(y, \theta^*) = E_\mathcal{T}[\nabla_\theta L_\mathcal{T}(y, f(\mathcal{T}; \theta^*))]$ is also well defined and exists for $P$-almost every $y$.

To verify condition (v), we explicitly compute the Hessian:

$$\begin{aligned}
\nabla R(\theta^*) &= E_Y[\nabla_\theta \bar{L}(Y, \theta^*)] \\
&= E_Y[E_\mathcal{T}[\nabla_\theta L_\mathcal{T}(Y, f(\mathcal{T}; \theta^*))]] \\
&= E_\mathcal{T}[E_Y[\nabla_\theta L_\mathcal{T}(Y, f(\mathcal{T}; \theta^*))]] \quad \text{since } \mathcal{T} \perp\!\!\!\perp Y \\
&= E_\mathcal{T}[((1 - \mathcal{T})P(f(\mathcal{T}; \theta^*) \geq Y) - \mathcal{T}P(f(\mathcal{T}; \theta^*) \leq Y))\nabla_\theta f(\mathcal{T}; \theta^*)] \\
&= E_\mathcal{T}[(P(Y \leq f(\mathcal{T}; \theta^*)) - \mathcal{T})\nabla_\theta f(\mathcal{T}; \theta^*)] \\
&= E_\mathcal{T}[(F(f(\mathcal{T}; \theta^*)) - \mathcal{T})\nabla_\theta f(\mathcal{T}; \theta^*)].
\end{aligned}$$

---

[1]A. W. van der Vaart. *Asymptotic Statistics*. Cambridge Series in Statistical and Probabilistic Mathematics. Cambridge University Press, 1998.

$$\nabla^2 R(\theta^*) = E_{\mathcal{T}}[\nabla_\theta \{(F(f(\mathcal{T}; \theta^*)) - \mathcal{T})\nabla_\theta f(\mathcal{T}; \theta^*)\}]$$
$$= E_{\mathcal{T}}[\nabla_\theta f(\mathcal{T}; \theta^*) F'(f(\mathcal{T}; \theta^*))\nabla_\theta f(\mathcal{T}; \theta^*)^\top + (F(f(\mathcal{T}; \theta^*)) - \mathcal{T})\nabla_\theta^2 f(\mathcal{T}; \theta^*)]$$
$$= E_{\mathcal{T}}[p(f(\mathcal{T}; \theta^*))\nabla_\theta f(\mathcal{T}; \theta^*)\nabla_\theta f(\mathcal{T}; \theta^*)^\top]$$
$$= Q \succ 0.$$

The third equality in the computation of the Hessian above follows by assumption that $f$ is well specified, since $f(\tau; \theta^*) = F^{-1}(\tau)$, so $F(f(\tau; \theta^*)) - \tau = 0$.

With all conditions verified, applying Theorem 5.23 from van der Vaart (1998) we have,

$$\sqrt{n}(\hat{\theta}^{(n)} - \theta^*) = -Q^{-1} \cdot \frac{1}{\sqrt{n}} \sum_{i=1}^n E_{\mathcal{T}}[\nabla_\theta L_{\mathcal{T}}(y_i, f(\mathcal{T}; \theta^*))] + o_P(1). \tag{5}$$

Focusing on the sum term,

$$\frac{1}{\sqrt{n}} \sum_{i=1}^n E_{\mathcal{T}}[\nabla_\theta L_{\mathcal{T}}(y_i, f(\mathcal{T}; \theta^*))]$$
$$= \frac{1}{\sqrt{n}} \sum_{i=1}^n E_{\mathcal{T}}\left[((1 - \mathcal{T})\mathbb{1}(f(\mathcal{T}; \theta^*) \geq y_i) - \mathcal{T}\mathbb{1}(f(\mathcal{T}; \theta^*) \leq y_i))\nabla_\theta f(\mathcal{T}; \theta^*)\right]$$
$$= \frac{1}{\sqrt{n}} \sum_{i=1}^n E_{\mathcal{T}}\left[(\mathbb{1}(y_i \leq f(\mathcal{T}; \theta^*)) - \mathcal{T})\nabla_\theta f(\mathcal{T}; \theta^*)\right]$$

Since $\mathbb{1}(y_i \leq f(\mathcal{T}; \theta^*))$ are IID samples from Bernoulli$(\mathcal{T})$ and $\mathcal{T}$ is independent of $Y$, the random variable $E_{\mathcal{T}}\left[(\mathbb{1}(Y \leq f(\mathcal{T}; \theta^*)) - \mathcal{T})\nabla_\theta f(\mathcal{T}; \theta^*)\right]$ has mean $\mathbf{0}$ and variance

$$V = E_{\mathcal{T}}[\mathcal{T}(1 - \mathcal{T})\nabla_\theta f(\mathcal{T}; \theta^*)\nabla_\theta f(\mathcal{T}; \theta^*)^\top].$$

By the central limit theorem, the sum converges in distribution to $\mathcal{N}(\mathbf{0}, V)$. Combining this with Equation (5) yields,

$$\sqrt{n}(\hat{\theta}^{(n)} - \theta^*) \xrightarrow{d} \mathcal{N}(\mathbf{0}, Q^{-1}V(Q^{-1})^\top).$$

Applying the delta method with $f(\tau; \theta)$ as a function of $\theta$ for a fixed desired quantile $\tau$, we have the final result,

$$\sqrt{n}(f(\tau; \hat{\theta}^{(n)}) - f(\tau; \theta^*)) \xrightarrow{d} \mathcal{N}\left(\mathbf{0}, \nabla_\theta f(\tau; \theta^*)^\top Q^{-1}V(Q^{-1})^\top \nabla_\theta f(\tau; \theta^*)\right).$$

$\square$

### A.3 Sufficient conditions for consistency

**Lemma 1.** *If $f(\tau, \theta)$ is convex in $\theta$ for all $\tau \in (0, 1])$, $\Theta$ is convex, and $\theta^*$ is in the interior of $\Theta$, then $\hat{\theta}^{(n)}$ is weakly consistent, $\hat{\theta}^{(n)} \xrightarrow{P} \theta^*$.*

*Proof.* The result follows from Proposition 7.4 from Hayashi (2011), provided we show that $\bar{L}(y, \theta)$ is also convex.

$$\bar{L}(y, \theta) = E_{\mathcal{T}}[L_{\mathcal{T}}(y, f(\mathcal{T}; \theta))]$$
$$= E_{\mathcal{T}}[\max(\mathcal{T}(y - f(\mathcal{T}; \theta)), (\mathcal{T} - 1)(y - f(\mathcal{T}; \theta)))]$$

For any fixed scalar value $\tau$, the term $\tau(y - f(\tau; \theta))$ is convex in $\theta$ since it is a linear mapping of $f(\tau; \theta)$. The same applies to the term $(\tau - 1)(y - f(\tau; \theta))$. Therefore, $\max(\tau(y - f(\tau; \theta)), (\tau - 1)(y - f(\tau; \theta)))$, as the pointwise maximum of two convex functions, is also convex in $\theta$. Finally, the linearity of the $E_{\mathcal{T}}[\cdot]$ operator gives that $\bar{L}(y, \theta)$ is convex in $\theta$. $\square$

## A.4    A note on uniqueness of $\theta^*$

Theorem 1 and Lemma 1 assume that $f(\tau; \theta)$ is well specified with a unique $\theta^* \in \Theta \subseteq \mathbb{R}^m$ such that $f(\tau; \theta^*) = F^{-1}(\tau)$ for all $\tau \in (0, 1)$, and that $\theta^*$ is also the unique minimizer for the risk $R(\theta) = E_{Y, \mathcal{T}}[L_\mathcal{T}(Y, f(\mathcal{T}; \theta))]$. The property that $f(\tau; \theta^*) = F^{-1}(\tau)$ directly implies that $\theta^*$ is a minimizer of $R(\theta)$; however, whether $\theta^*$ is the *unique* minimizer of $R(\theta)$ depends on the distribution of $\mathcal{T}$ and function class defined by $f$ and $\Theta$.

**Lemma 3.** Suppose $f(\tau; \theta)$ is well specified with a unique $\theta^* \in \Theta \subseteq \mathbb{R}^m$ such that $f(\tau; \theta^*) = F^{-1}(\tau)$ for all $\tau \in (0, 1)$. For $\theta \in \Theta$, define the set $T_\theta := \{\tau : f(\tau; \theta) \neq f(\tau; \theta^*)\}$. If for all $\theta \in \Theta$ where $\theta \neq \theta^*$, the set $T_\theta$ has measure greater than 0 under random variable $\mathcal{T}$, then $\theta^*$ is also the unique minimizer for the risk $R(\theta) = E_{Y, \mathcal{T}}[L_\mathcal{T}(Y, f(\mathcal{T}; \theta))]$.

*Proof.* Suppose there exists $\theta'$ where $\theta' \neq \theta^*$ but $\theta'$ also minimizes of the risk $R(\theta)$. Since $\theta' \neq \theta^*$, the set $T_{\theta_b}$ is nonempty, and for all $\tau \in T_{\theta'}$,

$$f(\tau; \theta') \neq F^{-1}(\tau) \implies \theta' \neq \arg\min_\theta E_Y[L_\tau(Y, f(\tau; \theta^*))]$$

$$\implies E_Y[L_\tau(Y, f(\tau; \theta'))] > E_Y[L_\tau(Y, f(\tau; \theta^*))].$$

Then since $T_{\theta'}$ has measure greater than 0 under random variable $\mathcal{T}$, this implies

$$E_\mathcal{T}[E_Y[L_\mathcal{T}(Y, f(\mathcal{T}; \theta'))]] > E_\mathcal{T}[E_Y[L_\mathcal{T}(Y, f(\mathcal{T}; \theta^*))]],$$

which contradicts that $\theta'$ minimizes the risk $R(\theta)$. $\square$

The conditions of uniqueness can be satisfied by a combination of wide enough support of $\mathcal{T}$, and a limited enough function class given by $f, \Theta$. For example, if $\mathcal{T}$ is supported on the full interval $(0, 1)$, and $f(\tau; \theta) = \sum_{j=1}^m \theta_j \tau^j$ is a polynomial function over $\Theta \subseteq \mathbb{R}^m$, then any pair $\theta \neq \theta'$ would yield a set $\{\tau : f(\tau; \theta) \neq f(\tau; \theta')\}$ with measure greater than 0.

Uniqueness breaks down of the function class is too expressive. For example, if $f(\tau; \theta)$ is infinitely expressive over $\Theta$ (e.g. $\Theta = \mathbb{R} \times (0, 1)$, and $f(\tau; \theta) = \theta_0 \mathbb{1}(\theta_1 = \tau)$), then it's possible to find a pair $\theta \neq \theta'$ that yield a set $\{\tau : f(\tau; \theta) \neq f(\tau; \theta')\}$ that contains only a single value of $\tau$ with measure 0.

Uniqueness can also break down if the support of $\mathcal{T}$ is too limited. For example, if $\mathcal{T}$ is only supported on a single value $\tau_0$, and $f(\tau; \theta) = \sum_{j=1}^m \theta_j \tau^j$ is a polynomial function, then it's possible to find a pair $\theta \neq \theta'$ where $f(\tau_0; \theta) = f(\tau_0; \theta')$, and thus $\{\tau : f(\tau; \theta) \neq f(\tau; \theta')\}$ has measure 0 for $\mathcal{T}$.

## A.5    Basic extension to categorical features

**Corollary 1.** *Let $X$ be distributed uniformly over a finite set of categorical values $\mathcal{X}$. For notational convenience, let $\mathcal{X} = \{1, ..., m\} \subset \mathbb{N}$. Suppose that a dataset $\{(x_i, y_i)\}_{i=1}^n$ is created by sampling $\{y_j\}_{j=1}^{\frac{n}{m}}$ values IID from the conditional distribution $Y|X = x$ for each value of $x$ (assuming $m$ divides $n$), and taking the union of these sets.*

*Let $f : \mathcal{X}, (0, 1), \Theta \to \mathbb{R}$ be fully separable over $x$; that is, let $\Theta = \Theta_1 \times ... \times \Theta_m$, where $\Theta_x$ represents a copy of a given parameter space $\Theta_0$ for each value of $x \in \mathcal{X}$. Let $f$ take the form*

$$f(x, \tau; \theta) = \sum_{x_0 \in \mathcal{X}} \mathbb{1}(x = x_0) g(\tau; \theta_{x_0})$$

*for some function $g : (0, 1), \Theta_0 \to \mathbb{R}$.*

*Suppose $f(x, \tau; \theta)$ is well specified with a unique $\theta^* \in \Theta \subseteq \mathbb{R}^m$ such that $f(x, \tau; \theta^*) = q_\tau(x)$ for all $\tau \in (0, 1)$ and all $x \in \mathcal{X}$, and that $\theta^*$ is also a unique minimizer for the risk*

$R(\theta) = \frac{1}{m} \sum_{x \in \mathcal{X}} E_{Y,\mathcal{T}}[L_{\mathcal{T}}(Y, f(x, \mathcal{T}; \theta))|X = x]$. Suppose the estimator $\hat{\theta}^{(n)}$ is weakly consistent, $\hat{\theta}^{(n)} \xrightarrow{P} \theta^*$. Suppose further that the function $\theta \mapsto f(x, \tau; \theta)$ is continuous, locally Lipschitz, and twice differentiable at $\theta = \theta^*$ for all $\tau \in (0, 1)$ and all $x \in \mathcal{X}$.

Then for each $x \in \mathcal{X}$,

$$\sqrt{\frac{n}{m}}(f(x, \tau; \hat{\theta}^{(n)}) - f(x, \tau; \theta^*)) \xrightarrow{d} \mathcal{N}\left(0, \nabla_\theta f(x, \tau; \theta^*)^\top Q^{-1} V (Q^{-1})^\top \nabla_\theta f(x, \tau; \theta^*)\right)$$

where,

$$Q = E_\mathcal{T}[p_x(f(x, \mathcal{T}; \theta^*))\Gamma(\mathcal{T})], V = E_\mathcal{T}[\mathcal{T}(1 - \mathcal{T})\Gamma(\mathcal{T})],$$

with $p_x(y)$ being the density of $Y|X = x$ and $\Gamma(\tau) = \nabla_\theta f(x, \tau; \theta^*) \nabla_\theta f(x, \tau; \theta^*)^\top$.

*Proof.* For notational convenience, let $\mathcal{X} = \{1, ..., m\} \subset \mathbb{N}$. Define loss function $\bar{L}(x, y, \theta) = E_\mathcal{T}[L_\mathcal{T}(y, f(x, \mathcal{T}; \theta))]$. Let $\hat{\theta}^{(n)}$ minimize the empirical risk over samples $\{x_i, y_i\}_{i=1}^n$, $\hat{\theta}^{(n)} = \arg\min_\theta \frac{1}{n} \sum_{i=1}^n \bar{L}(y_i, x_i, \theta)$. By assumption of the separability of $f$ and $\Theta$ by $x$,

$$\hat{\theta}^{(n)} \in \arg\min_{\theta \in \Theta} \frac{1}{n} \sum_{i=1}^n \bar{L}(y_i, x_i, \theta)$$

$$= \arg\min_{\theta \in \Theta} \sum_{x \in \mathcal{X}} \frac{1}{m} \left( \frac{1}{n/m} \sum_{i=1}^n \bar{L}(y_i, x, \theta) \mathbb{1}(x_i = x) \right)$$

$$= \arg\min_{\theta \in \Theta} \sum_{x \in \mathcal{X}} \frac{1}{m} \left( \frac{1}{n/m} \sum_{i=1}^n E_\mathcal{T}[L_\mathcal{T}(y_i, f(x, \mathcal{T}; \theta))] \mathbb{1}(x_i = x) \right)$$

$$= \arg\min_{\theta \in \Theta} \sum_{x \in \mathcal{X}} \frac{1}{m} \left( \frac{1}{n/m} \sum_{i=1}^n E_\mathcal{T}[L_\mathcal{T}(y_i, g(\mathcal{T}; \theta_x))] \mathbb{1}(x_i = x) \right)$$

$$= \sum_{x \in \mathcal{X}} \frac{1}{m} \left( \arg\min_{\theta_x \in \Theta_x} \frac{1}{n/m} \sum_{i=1}^n E_\mathcal{T}[L_\mathcal{T}(y_i, g(\mathcal{T}; \theta_x))] \mathbb{1}(x_i = x) \right)$$

Therefore, each component $\hat{\theta}_x^{(n)}$ minimizes an empirical loss over points with $x_i = x$:

$$\hat{\theta}_x^{(n)} \in \arg\min_{\theta_x \in \Theta_x} \frac{1}{n/m} \sum_{i=1}^n E_\mathcal{T}[L_\mathcal{T}(y_i, g(\mathcal{T}; \theta_x))] \mathbb{1}(x_i = x).$$

By assumption, components of the optimum $\theta_x^*$ also uniquely minimize each conditional risk

$$\theta_x^* = \arg\min_{\theta_x \in \Theta_x} E_{Y,\mathcal{T}}[L_\mathcal{T}(Y, f(x, \mathcal{T}; \theta))|X = x].$$

Therefore, we may apply Theorem 1 to describe the convergence of each component $\hat{\theta}_x^{(n)}$ to its respective optimum $\theta_x^*$ for each $x \in \mathcal{X}$, yielding the result. $\square$

## A.6 Additional Examples Illustrating Theorem 1

**Single parameter uniform anchored at 0.** Suppose the true distribution of interest is $Y \sim \text{Unif}(0, \alpha)$. Let $f(\tau; \theta) = \theta\tau$, which correctly parameterizes the true linear inverse CDF. By Theorem 1, the asymptotic

variance of $\sqrt{n}(\hat{\theta}^{(n)} - \theta^*)$ is

$$Q^{-1}V(Q^{-1})^\top = \alpha^2(E_\mathcal{T}[\mathcal{T}^3] - E_\mathcal{T}[\mathcal{T}^4])/E_\mathcal{T}[\mathcal{T}^2]^2.$$

Thus, the asymptotic variance depends on the distribution of $\mathcal{T}$ through its third and fourth moments, and depends on the distribution of $Y$ through $\alpha$. For $\mathcal{T} \sim \text{Unif}(0, 1)$, the exact asymptotic variance is $0.45\alpha^2$. To estimate a specific $\tau_0$-quantile, we would evaluate the function $f(\tau_0; \hat{\theta}^{(n)}) = \hat{\theta}^{(n)}\tau_0$, which would have an asymptotic variance of $0.45\alpha^2\tau_0^2$.

### A.7 A Connection To Regression On Sample Quantiles

We provide some more intuition here about how the training procedure we describe in this paper - learning a function $f(\tau)$ with an expected pinball loss - smooths sample quantiles in the case that there are no conditioning $x$-features. In particular, we can show that our proposed method (except with a uniform discrete rather than uniform continuous distribution for $\tau$) is equivalent to a regression on the observed sample quantiles with a particular cost function.

**Proposition 1.** *Given a dataset $\{y_i\}_{i=1}^n$ of $n$ elements, construct the dataset $\{\tau(i), y^{(i)}\}_{i=1}^n$ where $y^{(i)}$ represents the ith order statistic of the data and $\tau(i) = \frac{i-1}{n-1}$ is the sample quantile that order statistic represents. Then training a model with an expected pinball loss (1) where $\tau$ is drawn uniformly from the discrete distribution $\{\tau(i)\}_{i=1}^n$ is equivalent to learning a single-variable regression over the data points $\{\tau(i), y^{(i)}\}_{i=1}^n$. The cost function c for the regression at the specific point $(\tau(i), y^{(i)})$ is 0 when $f(\tau(i)) = y^{(i)}$ and otherwise piecewise-linear with slope*

$$c'(f(\tau(i))) = \begin{cases} -\tau(i) - \sum_{j=1}^{i-1} \mathbb{1}(f(\tau(i)) < y^{(j)}) & \text{for } f(\tau_i) < y^{(i)} \\ (1 - \tau(i)) + \sum_{j=i+1}^n \mathbb{1}(f(\tau(i)) > y^{(j)}) & \text{for } f(\tau_i) > y^{(i)} \end{cases}$$

*Proof.* The $\tau$-pinball loss can be expressed as

$$L_\tau(y, f(\tau)) = \begin{cases} \tau(y - f(\tau)) & \text{if } f(\tau) < y \\ (1 - \tau)(f(\tau) - y) & \text{if } f(\tau) \geq y \end{cases}$$

When training with an expected pinball loss over the discrete values of $\tau$ corresponding to the sample quantiles of the data, this means that the expected pinball loss is a function only of $f(\tau(i))$, or the quantile predictions of $f$ at those sample quantiles. This is precisely the context of ordinary regression; the loss we face is a function only of the values our function takes at the $x$-coordinates of our data points, which here are set up to be exactly the $\{\tau(i)\}_{i=1}^n$.

Let us consider the pinball loss applied to $\tau(i)$. To optimize (1), we average the losses across our data points $\{y_i\}_{i=1}^n$; for this proof, we (equivalently) work with the sums instead. Let our regression-equivalent cost $c(f(\tau(i)))$ be equal to the relative increase in the $\tau(i)$ pinball loss we face, summed over the data points, compared to if we had predicted $f(\tau(i)) = y^{(i)}$.

By the definition of the $\tau(i)$ pinball loss, decreasing $f(\tau(i))$ by $\epsilon$ will mean that we face an additional cost of $\tau(i)\epsilon$ for every data point we are below and moving away from and recover $(1-\tau(i))\epsilon$ for every data point we are above and moving towards. Say we start at $f(\tau(i)) = y^{(i)}$ and choose an $\epsilon$ such that $f(\tau(i)) - \epsilon > y^{(i-1)}$. The point $y^{(i)}$ has $\tau(i)(n-1)$ data points below it and $(1-\tau(i))(n-1)$ data points above it by definition. So our marginal loss from decreasing $f(\tau(i))$ by $\epsilon$ is $\tau(i)(1-\tau(i))(n-1)\epsilon$ due to the points above $y^{(i)}$ while we recover $(1-\tau(i))\tau(i)(n-1)\epsilon$ due to the points below it. These are equal! But we also lose $\tau(i)\epsilon$ because we are now moving away from the point $y^{(i)}$ itself, making the slope of $c(f(\tau(i)))$ equal to $-\tau(i)$ in the region between $y^{(i-1)}$ and $y^{(i)}$. Each time we pass a neighboring data point, we start losing $\tau(i)\epsilon$ from that point instead of gaining $(1 - \tau(i))\epsilon$, increasing the steepness of the slope by 1. This is exactly what we want to show: $c'(f(\tau(i)))$ is equal to $-\tau(i)$ minus the number of data points smaller than $y^{(i)}$ that $f(\tau(i))$ is smaller than.

The opposite case - analyzing what happens when we increase $f(\tau(i))$ by $\epsilon$ is essentially analagous. Increasing $f(\tau(i))$ by $\epsilon$ leads to an additional cost of $(1 - \tau(i))\epsilon$ for every data point we are above and moving away from and a decreased cost of $\tau(i)\epsilon$ for every data point we are below and moving towards. This implies the

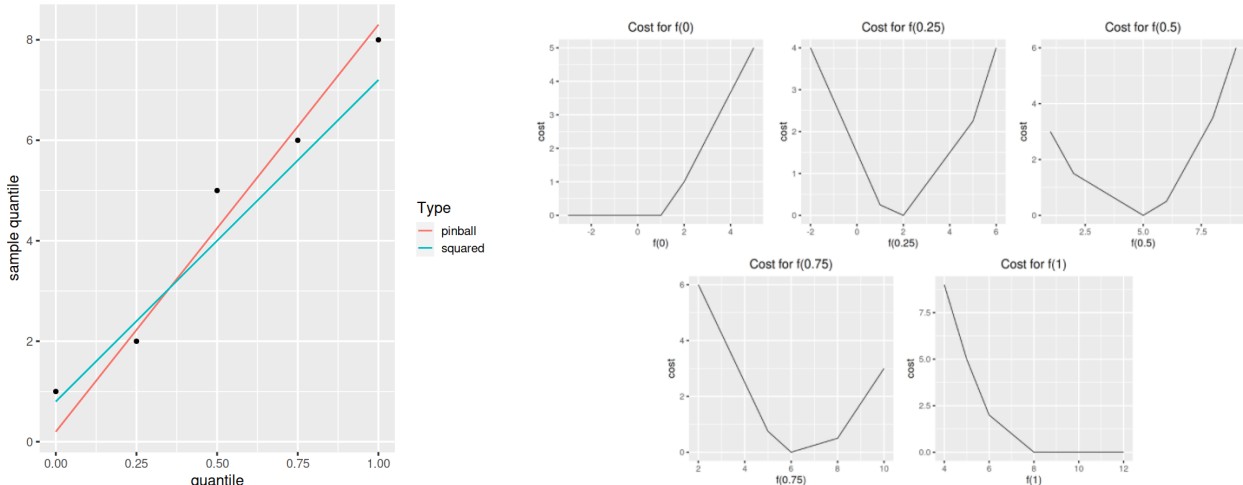

Figure 6: Given the dataset $\{1, 2, 5, 6, 8\}$, we show visually how learning $f(\tau)$ via the expected pinball loss with $\tau$ as a feature can be seen as a type of regression on the sample quantiles. For simplicity, we take $f(\tau)$ to be linear, which would be the correct functional form if the data are drawn from a uniform distribution. On the left, we plot the sample quantiles and compare the lines learned by a direct squared-loss regression on those data points (i.e. $\{\tau(i), y^{(i)}\}_{i=1}^{5}$) with the line learned by our method (i.e. learning $f(\tau)$ with the expected pinball loss). On the right, we show the equivalent costs imposed on predictions for each sample quantile if we reframed training $f(\tau)$ with an expected pinball loss over $\tau = \{0, 0.25, 0.5, 0.75, 1\}$ as a regression on sample quantiles. The cost is 0 when the prediction is exactly the corresponding sample quantile and weakly rising as it varies.

slope of the cost function between $y^{(i)}$ and $y^{(i+1)}$ is $(1 - \tau(i))$, as the cost accrued and recovered by the points above and below $y^{(i)}$ cancel out and we only have the net cost due to moving away from $y^{(i)}$ itself. And for the same reason described above, passing data points adds to the slope by 1. □

See Figure 6 for some example cost functions.

This cost function is nonstandard in a couple ways. First, it varies per point. And second, its steepness is a function of how many neighboring sample data points the prediction $f(\tau)$ is off by rather than just how far away it is from the sample quantile in an absolute sense.

Still, it otherwise follows the normal properties of a cost function, such as taking the value 0 when the prediction is exactly correct and (weakly) rising in a convex fashion as the prediction is too high or too low. This makes it clear why a sufficiently flexible function $f(\tau)$ will exactly pass through all of the sample quantiles and also why a more limited $f(\tau)$ will serve as a smoother, passing below some sample quantiles and above others.

**Worked Example:** Consider the dataset $\{1, 2, 5, 6, 8\}$. These correspond to the sample quantiles $\{0, 0.25, 0.5, 0.75, 1\}$. Figure 6 plots the sample quantiles on a graph, the least squares regression fit, and the expected pinball loss fit with linear $f(\tau)$. We can see that both are quite similar. For example, both estimate the median as being lower than the observed sample median. The figure also shows the regression-equivalent cost function for each of the quantile predictions, if we were to think of our method as a regression on sample quantiles.

## A.8 Sine-skew Simulated Conditional Distribution

Fig. 5 illustrates the true quantiles of the sine-skew distribution with the different parameters used in our simulations. Sine-skew (1,1) Sine-skew (7,7) Sine-skew (1,7) True quantiles of the sine-skew

~~distribution with different noise parameters Torossian et al. (2020). The colored lines show the quantiles for $\tau = 0.1, 0.4, 0.5, 0.6, 0.9.$~~

### A.8  Calibration Error Metric Definition

A predicted conditional inverse CDF $f(x, \tau; \theta)$ is *calibrated* if

$$P(Y \leq f(X, \tau; \theta)) = \tau.$$

A calibration error metric measuring the difference between $P(Y \leq f(X, \tau; \theta))$ and $\tau$ can be approximated over a finite dataset with data points $\{x_i, y_i\}_{i=1}^n$ and quantiles $\tau_1, ..., \tau_m$. We consider the absolute deviation:

$$\frac{1}{m} \sum_{j=1}^{m} \left| \left( \frac{1}{n} \sum_{i=1^n} \mathbb{1}(y_i \leq f(x_i, \tau_j; \theta)) \right) - \tau_j \right| \tag{6}$$

Others such as Kuleshov et al. (2018) have considered the squared deviation.

### A.9  Hyperparameter Tuning Details

For each real data experiment, we tune hyperparameters over a validation dataset, where the validation dataset is selected according to Section 5.2. The tuning criterion was the validation pinball loss averaged over $\tau \in \{0.01, ..., 0.99\}$. For DLNs, the search spaces for real data experiments were the following:

- The number of calibration keypoints for the piecewise-linear calibration function over $\tau$ were tuned between $\{10, 20, 50, 100\}$. Note that as this number goes up, you get more model flexibility with respect to $\tau$ (only), which makes it effectively like just training separate models for each $\tau$.

- The number of lattice keypoints for $\tau$ was tuned between $\{2, 3, 5, 7, 10\}$. That controls the flexibility of the interactions between $\tau$ and the other features $x$.

- Other feature calibration keypoints were tuned between $\{5, 10, 15, 20\}$, which are common values for DLNs.

- Step sizes were tuned between $\{0.001, 0.005, 0.01, 0.05, 0.1\}$

- minibatch sizes were tuned between $\{1000, 10000\}$.

- Number of steps was tuned between $\{100, 1000, 10000\}$.

To tune the DLN architecture for experiments with real data, we searched over the following hyperparameters:

- Air Quality: number of lattices $\in \{4, 8\}$,number of features per lattice $= 6$,

- Puzzles: number of lattices $\in \{8, 16, 32\}$,number of features per lattice $= 5$

- Traffic: single lattice which includes all features (not ensemble)

- Wine: number of lattices $\in \{100, 200, 400, 800\}$, number of features per lattice $\in \{2, 4, 8\}$.

