# OpenReview forum: "Expected Pinball Loss For Quantile Regression And Inverse CDF Estimation"
_TMLR — Rejected by TMLR_

### Review · Reviewer_Rz26 · 2022-12-03

**Summary Of Contributions:**

This paper considers the problem of estimating quantiles of a random
variable. Instead of focusing on a single quantile, it estimate the whole
inverse function of the CDF. This requires specifying a parametric model for the
inverse CDF, and the authors recommend using deep lattice networks (DLNs), with a
monotonicity constraint to avoid quantile crossing. For model training, an
expected pinball loss is used, where the expectation is taken with respect the a
user-specified distribution of the quantiles. Both theoretical results and
numerical results are provided to demonstrate the advantage of the proposed
method.


**Audience:**

Yes

**Broader Impact Concerns:**

No concern.

**Claims And Evidence:**

Yes

**Requested Changes:**

It seems that a critical reason that the expected pinball loss can be better
than the single pinball loss is the correctness of the inverse CDF
$f(\tau,\theta)$. With this correctly specified parametric model, the pinball
loss is informative at every quantile value, even if it is not of interest. This
means the expected pinball loss uses additional assumptions to gain the
estimation efficiency. It would be helpful if the authors can discuss this
point. Furthermore, some investigation on the misspecification of the
$f(\tau,\theta)$ would be useful. If theoretical results are difficult, some
numerical experiments would be fine.

Please explain why the proposed method is better for more central quantile in
Sections 3.3.3 and 3.3.4, but it is better for the extreme quantile in
Figure 1.

I am confused by the fact that the asymptotic variance for estimating a specific
$\tau_{0}$ is exactly zero if $\tau=0.5$. This means that there is no estimation
error at all. Please give an explanation on this.

In the list of required conditions in page 15, some details are required. For
example, (i) cannot be directly obtained from a central limit theorem. In
general, a central limit theorem is easier to get compared with the
consistency.


**Strengths And Weaknesses:**

Strengths:
The paper is well written overall. A theoretical result is provided to show that
estimating the whole inverse CDF can be better than focusing on a single
quantile. The comparison to Harrell-Davis estimator is also very helpful to
demonstrate the advantage of the proposed method.

Numerical experiments were well designed, and multiple benchmark datasets are
used.

Weaknesses:
It is better to give an intuitive explanation on why estimating the whole
inverse CDF can be better than estimating a single quantile.

There seems to be a contradiction on when the proposed method is better than the
single pinball loss. Section 3.3 shows that the proposed method is better when
the quantile is close to 0.5, while Section 3.2 in Figure 1 seems to indicate
that the proposed method has a more significant advantage for the extreme
quantile.

There seems to be some gaps in the proof.

---

> ### Author Response · Authors · 2023-01-07
> **Response to Reviewer Rz26**
>
> Thank you for the review! We include specific responses to your comments and suggestions here.
>
> > It is better to give an intuitive explanation on why estimating the whole inverse CDF can be better than estimating a single quantile.
>
> The intuition is regularization through quantile smoothing; in estimating the whole inverse CDF, information about neighboring quantiles can augment the estimate of any particular quantile. We have added some language to  try to make that clearer. See especially Sec 3.1 and Sec 3.2, and also Appendix A.7.
>
> > There seems to be a contradiction on when the proposed method is better than the single pinball loss. Section 3.3 shows that the proposed method is better when the quantile is close to 0.5, while Section 3.2 in Figure 1 seems to indicate that the proposed method has a more significant advantage for the extreme quantile.
>
> We showed in Section 3.3 that when the true distribution is known to be uniform, our method would perform relatively better at central quantiles compared to single-quantile estimators. We have added Table 2 which validates this empirically. Figure 1 in Section 3.2 analyzes a different case where the true distribution is unknown, and so our DLN approximates it with a piecewise-linear function; in this case, we do not provide theoretical guarantees and our experiments show that we can sometimes improve at extreme quantiles in this general setting.
>
> > There seems to be some gaps in the proof.
>
> See below response.
>
> > It seems that a critical reason that the expected pinball loss can be better than the single pinball loss is the correctness of the inverse CDF . With this correctly specified parametric model, the pinball loss is informative at every quantile value, even if it is not of interest. This means the expected pinball loss uses additional assumptions to gain the estimation efficiency. It would be helpful if the authors can discuss this point. Furthermore, some investigation on the misspecification of the  would be useful. If theoretical results are difficult, some numerical experiments would be fine.
>
> All of our experiments, including our simulations in Section 3.2, concern the case where the inverse CDF is not correctly specified by the DLN. The fact that we sometimes see improvements suggests that training with an expected pinball loss is competitive even without being able to take advantage of any distributional knowledge.
>
> We can think about varying the number of keypoints, as in Figure 1, as a way of exploring misspecification as well. Too few keypoints lead the model to poorly approximate the true inverse CDF, while too many can match the quantiles of the training data more precisely but perhaps generalize less well.
>
> > Please explain why the proposed method is better for more central quantile in Sections 3.3.3 and 3.3.4, but it is better for the extreme quantile in Figure 1.
>
> See above response.
>
> > I am confused by the fact that the asymptotic variance for estimating a specific $\tau_0$ is exactly zero if $\tau = 0.5$. This means that there is no estimation error at all. Please give an explanation on this.
>
> In the Uniform($-\frac{\alpha}{2}, \frac{\alpha}{2}$) example, the inverse CDF we learn is parameterized as $\theta*(\tau-0.5)$, which ensures it predicts the correct value of 0 for the 0.5 quantile. Our Uniform($\alpha, \beta$) example is perhaps a more realistic scenario when all quantiles are being estimated and none are known “for free” based on the functional form.
>
> > In the list of required conditions in page 15, some details are required. For example, (i) cannot be directly obtained from a central limit theorem. In general, a central limit theorem is easier to get compared with the consistency.
>
> Thank you for pointing this out – we agree that it would be valuable to elaborate on the consistency requirement, and have made changes to the statement and proof of Theorem 1 and added an additional Lemma 1 to address this. The reviewer is correct that consistency is not a given here, and depends on properties of the function $f$ and the parameter space $\Theta$. We have moved the assumption of weak consistency directly into the statement of the theorem, and have added Lemma 1 to provide an example set of sufficient conditions for weak consistency. We have also provided more details on the uniqueness of the optimum $\theta^*$ in the Appendix. We hope that this addresses any remaining concerns surrounding the details of Theorem 1.

---

> > ### Comment · Reviewer_Rz26 · 2023-02-22
> > **Thank you for your response**
> >
> > I thank the authors responding my comments, most of which have been adequately addressed. Here are some followup comments.
> >
> > 1. I still think a key reason that the expected pinball loss can be better than
> >    the single pinball loss is the correctness of the inverse CDF, at least
> >    theoretically. I am not very convinced by the numerical results. It is
> >    possible that the DLN approximate the inverse well enough so that the bias is
> >    negligible. But results may change for a different setting. It is fine to not
> >    providing theoretical results, but I think the authors should at least
> >    discuss the point of incorrect inverse CDF.
> >
> > 2. It would be better to add one or two sentences explaining the special
> >    situation of $\tau=0$.
> >
> > 3. I agree with another reviewer that the theoretical results in Theorem 1 does
> >    not take into account the constraint in (2). The theoretical result is weaker
> >    than it seems, which I think is fine, but it is better to be more explicit that
> >    Theorem 1 is about the estimator in the second displayed equation in page 7
> >    instead of the estimator defined in (1).

---

> > > ### Author Response · Authors · 2023-02-28
> > > **Follow-up Response to Reviewer Rz26**
> > >
> > > Thank you for your comments. We are not sure if we are supposed to upload a new version of the paper at this point, so let us know if that is what you are looking for. Until then, here are some responses.
> > >
> > > > I still think a key reason that the expected pinball loss can be better than the single pinball loss is the correctness of the inverse CDF, at least theoretically. I am not very convinced by the numerical results. It is possible that the DLN approximate the inverse well enough so that the bias is negligible. But results may change for a different setting. It is fine to not providing theoretical results, but I think the authors should at least discuss the point of incorrect inverse CDF.
> > >
> > > We agree that correctness of the inverse CDF is important to our theoretical results about when an expected pinball loss can outperform a single pinball loss, and can make that point clearer in the main body of our paper. We can also add more thoughts around the relative performance of an expected pinball loss model in practical settings where the true inverse CDF is not known.
> > >
> > > On that point, we think Section 3.2 provides helpful explanations and intuition about how the flexibility of the function class matters. Namely, with a function class sufficiently flexible in $\tau$, we tend to converge in performance to the single-quantile pinball loss; while with less flexibility, we may be able to perform better but also risk performing worse. As noted in Section 4.1, the DLN function class does have a universal approximation property in the general setting, with tunable hyperparameters reflecting that flexibility; this may provide some explanation for why we see fair good results for the expected pinball loss in our simulations and real-data experiments where the true inverse CDFs are not known.
> > >
> > > > It would be better to add one or two sentences explaining the special situation of $\tau=0$
> > >
> > > We can add an explanation about the $\tau=0$ (and $\tau=1$) cases to the paper. Practically speaking, the values the learned function takes at those extreme values will likely be such that the range covers all of the sample data; this may be useful when the output distribution is known to be bounded and less so in the general unbounded case.
> > >
> > > > I agree with another reviewer that the theoretical results in Theorem 1 does not take into account the constraint in (2). The theoretical result is weaker than it seems, which I think is fine, but it is better to be more explicit that Theorem 1 is about the estimator in the second displayed equation in page 7 instead of the estimator defined in (1).
> > >
> > > The constraints would affect Theorem 1 in the following way: if the constraints in (2) produce a feasible set $\Theta$ which is not convex, then Theorem 1 may not hold since Lemma 1 would no longer hold (though we can add example results showing the feasible set is in fact convex for some monotonic model architectures). Of course, we acknowledge that this is a very general treatment of constraints, and it’s possible that if we included the specific non-crossing constraints in (2) explicitly in the loss when analyzing asymptotic convergence, then the resulting rate could be even better than what’s given in Theorem 1. We will clarify that Theorem 1 is really about the simpler estimator on Page 7.

---

> > > > ### Comment · Reviewer_Rz26 · 2023-02-28
> > > > **Would sufficiently address my concern with the proposed changes**
> > > >
> > > > Thank you for your proposal.
> > > >
> > > > Your proposed changes in responses to item 1 are exact I am looking for.
> > > >
> > > > Item 2 is more minor and I believe I understand now.
> > > >
> > > > For item 3, I understood the difficult of obtaining theoretical results and was just looking for a clarification so that the theoretical contribution was not over-stated.

---

### Review · Reviewer_SsMT · 2022-12-23

**Summary Of Contributions:**

This paper consists of a theoretical and empirical analysis of a recent simultaneous quantile regression (SQR) estimator. The authors examine and extend the SQR framework over two aspects: convergence rate/asymptotic properties of the expected loss estimator and satisfiability of monotonicity constraints i.e. the crossing quantile problem. Overall, the presented analysis does improve the understanding and implications of the expected pinball-loss/quantile estimator. The paper could be strengthened by relating to applications more familiar to the community such as uncertainty quantification.

**Audience:**

Yes

**Broader Impact Concerns:**

As mentioned above, relating to the aspect of uncertainty quantification could emphasize the broader impact of the paper.

**Claims And Evidence:**

Yes

**Requested Changes:**

Please address the points raised in Weaknesses.
Relating to W2, since the goal of the paper is empirical evaluation of the expected pinball loss estimation, further analysis of extreme quantiles will contribute to a complete analysis. In particular, the point raised by the authors themselves, in the conclusion
"Is a uniform distribution (as done here and in Tagasovska & Lopez-Paz (2019)) most efficient? If one is only
interested in a single   would a distribution more peaked around that  produce better results?"
would extend the analysis nicely, even if the results are negative.

Quantiles have been widely used in the deep learning community for uncertainty quantification, in particular aleatoric uncertainty. It would help the paper and the community if the authors promote this work in this direction as well. This will require adding prediction intervals and evaluation of their quality such as in [1].

Related to the uncertainty, an important aspect is what happens with the quantile/inverse CDF outside of the training support, in OOD regions. Do all quantiles collapse? Are some regularisation techniques/architectural choices better than others?

Lastly, an easy solution for the DNN method to account for the quantile crossing problem by adding a regularisation term $max(\frac{\partial f(x, \tau)}{\partial \tau}, 0)$ for each training point $x_i$ or penalizing for margin of quantiles above or below the specified level $f(x, \tau + \epsilon)$. Were any of those considered in the experiments?

Minor:
- an unfinished sentence pg 7 par 1
- changing the title from Inverse CDF to  quantile might help the visibility of the paper
- DLNs could be introduced before 3.2
- In 5.3.2. $\tau \in$ is missing
- I appreciate the pizza example but would be more in favor to have a more impactful example in critical decision making.


[1] Pearce, Tim, et al. "High-quality prediction intervals for deep learning: A distribution-free, ensembled approach." International conference on machine learning. PMLR, 2018.

**Strengths And Weaknesses:**

Strengths:
S1: The paper reads well, it is self-contained and I appreciate the proper relevant literature review and overall presentation of related work.
S2: The theoretical analysis is supported by analytical examples and empirical results.
S3: The authors used proper scoring rules for evaluation.

Weaknesses:
W1: I am not entirely convinced that the analysis of the unconditional case extends to the conditional use of pinball loss. Further elaboration would help in supporting this statement.
W2: As seen and discussed in the experimental section, all methods suffer from the difficulty of accurate quantile estimation in the tails or higher quantiles. How would the authors try to address this issue? Why were lower tails never examined?
W3: Relying on the DLN architecture might be limiting for certain applications/data modalities, could the authors elaborate more on this point? What would be the alternative in that case?

---

> ### Author Response · Authors · 2023-01-07
> **Response to Reviewer SsMT**
>
> Thank you for the review! We include specific responses to your comments and suggestions here.
>
> > I am not entirely convinced that the analysis of the unconditional case extends to the conditional use of pinball loss. Further elaboration would help in supporting this statement.
>
> Conditioning on features adds a significant amount of complexity to the theoretical characterization of the full inverse CDF estimator, and as far as we know this is still an open problem. However, our unconditional theory can be extended in a very basic way to a setting with finite categorical features $X$ and a function class which is fully flexible and separable over the values of $X$. We’ve added Corollary 1 to formalize this extension. We’ve also added more discussion to Section 3 of the intuition that our theory can provide for the extension to conditional quantile regression. We’ve noted that stronger results for conditional estimation is an open question in the Conclusions.
>
> > As seen and discussed in the experimental section, all methods suffer from the difficulty of accurate quantile estimation in the tails or higher quantiles. How would the authors try to address this issue? Why were lower tails never examined?
>
> We chose quantiles that were of practical concern for the selected problems, we do not see any reasons that the lower tails would behave better or worse than the higher tail quantiles used in the experiments. Of course, extreme quantiles are hard in general, and one needs a lot of data to make decent conditional 99% quantile estimates! Working with the Beta distribution (also discussed below) may help.
>
> > Relying on the DLN architecture might be limiting for certain applications/data modalities, could the authors elaborate more on this point? What would be the alternative in that case?
>
> We have added a figure showing how the proposed monotonic-on-tau DLN layers can be integrated into generic architectures. An alternative is to impose monotonicity on tau using another function class, but we have found DLN models to be easy to train and work well (as seen in these experiments).
>
> > Relating to W2, since the goal of the paper is empirical evaluation of the expected pinball loss estimation, further analysis of extreme quantiles will contribute to a complete analysis. In particular, the point raised by the authors themselves, in the conclusion "Is a uniform distribution (as done here and in Tagasovska & Lopez-Paz (2019)) most efficient? If one is only interested in a single $\tau$ would a distribution more peaked around that produce better results?" would extend the analysis nicely, even if the results are negative.
>
> Great intuition - yes, a beta distribution over $\tau$ concentrated on the quantile of interest can be better than uniform if you just care about one quantile.  We didn’t include those results in the original submission because our experiments showed you can do surprisingly well just with the uniform distribution over $\tau$, and that requires no tuning. But if you don’t mind a hyperparameter (how concentrated the beta distribution is), it’s a good strategy.  We have added experimental results showing the sensitivity to the beta concentration in our simulations, and the real-data results with the beta concentration cross-validated. We will add cross-validated results for the Harrell-Davis comparison as we have done for the uniform distribution comparison.
>
> > Quantiles have been widely used in the deep learning community for uncertainty quantification, in particular aleatoric uncertainty. It would help the paper and the community if the authors promote this work in this direction as well. This will require adding prediction intervals and evaluation of their quality such as in [1].
>
> We have added some related work comments addressing this to both the Introduction and Conclusions/Open Questions sections. The improvement in calibration of our quantiles, shown in Table 6, suggests promise in the prediction intervals application.
>
> > Related to the uncertainty, an important aspect is what happens with the quantile/inverse CDF outside of the training support, in OOD regions. Do all quantiles collapse? Are some regularisation techniques/architectural choices better than others?
>
> Quantile collapse is a challenging problem indeed!  We have added a discussion of quantile collapse to the Conclusions, emphasizing its interplay with initialization and model flexibility.

---

> > ### Author Response · Authors · 2023-01-07
> > **Response to Reviewer SsMT, Part 2**
> >
> > > Lastly, an easy solution for the DNN method to account for the quantile crossing problem by adding a regularisation term...or penalizing for margin of quantiles...Were any of those considered in the experiments?
> >
> > No, but we have added some related work notes about that to Section 2.  We’ve been working with monotonic models for a long time, and our experience in practice is that *guarantees* are much more useful, so we focused our paper on methods that guarantee non-crossing, including comparing to the work of Gasthaus et al (2019).   To provide publishable evidence (rather than just our anecdotal evidence from our own work experience) that guarantees of non-crossing are useful, we have surveyed real-world customers and provided their negative responses to non-crosssing quantiles in the Introduction as motivation for our focus on guarantees of non-crossing.
> >
> > > an unfinished sentence pg 7 par 1
> >
> > Thank you, we rewrote part of that section for clarity and fixed this.
> >
> > > changing the title from Inverse CDF to quantile might help the visibility of the paper
> >
> > Great idea! We’ve changed it to: “Expected Pinball Loss For Quantile Regression And Inverse CDF Estimation.”
> >
> > > DLNs could be introduced before 3.2
> >
> > We found that hard to do without disrupting the narrative flow, but we’ve added a citation to make it easier to go look up DLNs when first introduced.
> >
> > > In 5.3.2. $\tau \in$ is missing
> >
> > Thank you, we rewrote part of that section for clarity and fixed this.
> >
> > > I appreciate the pizza example but would be more in favor to have a more impactful example in critical decision making.
> >
> > Changed to Airline scheduling.

---

### Review · Reviewer_VW4n · 2022-12-24

**Summary Of Contributions:**

This paper studies the topic of training a quantile regression model by minimizing an expected pinball loss over all quantiles. Moreover, the authors employ deep lattice networks to overcome the crossing quantile issues. Overall, theoretically, this paper only presents a shaky theorem for the proposed method. Many key theoretical points of the proposed method should be addressed. It turns out that many simulations are done without theory support. As a result, I think this paper is not ready for submission.

**Audience:**

Yes

**Claims And Evidence:**

Yes

**Requested Changes:**

Critical for acceptance: Please try to give the theoretical guarantees highlighted in weaknesses 1-4 and add more numerical evidence to answer weakness 5.

Optional to strengthen submission: modify some typo errors such as weakness 6.


**Strengths And Weaknesses:**

Strength:
1. The targeted problem is interesting. The writing of the paper is clear and easy to follow.
2. Several numerical experiments are conducted.

Weakness:
1. The only theorem this paper presents is Theorem 1. The theorem only proves the asymptotic normality with respect to equation (1). It does not take constraint (2) into consideration.
2. The asymptotic normality of quantile regression with features is not presented.
3. The convergence rate of the algorithm is not given.
4. In Figure 1, the authors show the MSE comparison of the HD, Sample, and DLN methods. On the graph, the DLN method works better than the other two. Meanwhile, on the right graph, the DLN method works better than the other two when the number of key points is larger than 50. No evidence is given or checked on how to choose such a number of key points.
5. On page 7, the authors mentioned that ``we found our DLN models trained in a similar amount of time as the DNN and SQF-DNN models’’. However, in the simulation study, no such evidence is given. Theoretically, the computation complexity is not given, either.
6. On page 4, the very last paragraph. The $\hat{\theta}_{\tau}^{(n)}$ should be

$\hat{\theta}_{\tau}^{(n)}$=

$\arg\min_{\theta} \frac{1}{n} \sum_{i=1}^{n} [L_{\tau}(y_{i}, \theta)]$.

---

> ### Author Response · Authors · 2023-01-07
> **Response to Reviewer VW4n**
>
> Thank you for the review! We include specific responses to your comments and suggestions here.
>
> > The only theorem this paper presents is Theorem 1. The theorem only proves the asymptotic normality with respect to equation (1). It does not take constraint (2) into consideration.
>
> We have added more detail on how the constraint (2) is considered in Theorem 1. Specifically, Theorem 1 defines a function $f(\tau; \theta)$ over a feasible parameter space $\Theta$, and the constraints (2) can be included in the definition of $\Theta$. We also added an additional Lemma 1 to show that the convexity of the feasible set $\Theta$ is important as a sufficient condition for consistency of the estimator in Theorem 1.
>
> > The asymptotic normality of quantile regression with features is not presented.
>
> The inclusion of features adds a significant amount of complexity to the theoretical characterization of the full inverse CDF estimator, and as far as we know this is still an open problem. However, our unconditional theory can be extended in a very basic way to a setting with finite categorical features $X$ and a function class which is fully flexible and separable over the values of $X$. We’ve added Corollary 1 to formalize this extension. We’ve also added more discussion to Section 3 of the intuition that our theory can provide for the extension to conditional quantile regression. We’ve added stronger theory for conditional estimators to the Conclusions and Open Questions section.
>
> > The convergence rate of the algorithm is not given.
>
> We provide an asymptotic convergence rate for estimation of the full inverse CDF, assuming an oracle that minimizes the empirical loss. Perhaps the reviewer is referring to a nonasymptotic convergence rate, which would be interesting future work.  We have added a note about that to the Conclusions and Open Questions section.
>
> > In Figure 1, the authors show the MSE comparison of the HD, Sample, and DLN methods. On the graph, the DLN method works better than the other two. Meanwhile, on the right graph, the DLN method works better than the other two when the number of key points is larger than 50. No evidence is given or checked on how to choose such a number of key points.
>
> We have added a table showing what happens when you cross-validate the number of keypoints (which controls the DLN flexibility).  The plot shows that there is a wide range of DLN flexibility where DLNs match or exceed the performance of alternative single-quantile estimators, in cases where we do not know the true function class of the distribution.
>
> > On page 7, the authors mentioned that ``we found our DLN models trained in a similar amount of time as the DNN and SQF-DNN models’’. However, in the simulation study, no such evidence is given. Theoretically, the computation complexity is not given, either.
>
> We have added some illustrative timing numbers, though of course specific timings depend greatly on the particular hyperparameters selected in a given experiment, the computing environment, and so on.
>
> > On page 4, the very last paragraph...
>
> Thank you for pointing this out! We’ve fixed this typo.

---

### Author Response · Authors · 2023-01-07
**Common Response**

We thank the reviewers for the careful reviews. We have made a number of revisions to the paper to address these comments, which we summarize below. We have also attached a diff between our revised and original submission to the end of the revised PDF to make it easier to review our changes.

**Theory:**
- To elaborate on the theoretical implications for conditional quantile regression as requested by Reviewer VW4n and SsMT, we’ve added an additional Corollary 1 to Section 3 that formalizes a basic extension of Theorem 1 to conditional quantile regression with finite categorical features $X$ and a function class which is fully flexible and separable over the values of $X$. General asymptotic theory for conditional inverse CDF estimation is still an open problem.
- We’ve elaborated on several assumptions in Theorem 1 in Section 3 and in the Appendix and made the theorem statement more precise, as requested by Reviewer Rz26.

**Experiments:**
- We have added Table 1, which cross-validates over the training data to perform keypoint selection in our unconditional quantile estimation simulations, as requested by Reviewer VW4n.
- To build on Reviewer SsMT’s suggestion about exploring alternative $tau$-sampling strategies that are concentrated around the desired quantile, we have augmented Figure 1, Table 7, and Table 8 to add results that train with a beta distribution over the quantiles.
- We measured and reported results on training time for the different function classes in Table 4, as requested by Reviewer VW4n.

**Writing:**
- We expanded the introduction with related work on other uncertainty estimation problems and more motivation for non-crossing.
- We greatly expanded Open Questions discussion in the Conclusions.
- We added some discussion of the beta distribution, in line with the experiments described above and Reviewer SsMT’s comments.
- We augmented our explanation of DLNs in Section 4.1 with new figures showing example 2D lattice functions and how their monotonicity-preserving layers can be incorporated into more general architectures.

---

### Decision · Action_Editors · 2023-03-09

**Recommendation:** Reject

**Comment:**

The reviewers identified some aspects of the paper that were presented in a confusing way, potentially leading to the results being misinterpreted. At a high level, the paper should be revised to more clearly describe the relationship between the motivation, the theoretical results, and simulations. When theoretical results are related but not directly applicable to motivating examples or simulation studies, this should be clearly stated, with the gap described precisely. In particular, the issue of "incorrect inverse CDF" (misspecification of the inverse CDF model, if I'm not mistaken) should be front and center in the discussion regarding limitations of the analysis and potentially limitations / unpredictable-guarantees of the methods. I agree with the observation that improvements may be due to sharing statistical strength, which may be favorable when the type of misspecification is in some sense benign.

Some reviewers demanded additional "theoretical convergence analyses". This is not a prerequisite for publication, but I would ask that the authors highlight what meaningful analyses are not carried out, potentially by pointing out gaps that exist in existing theoretical results. While it seems that many of the suggestions made by SsMT were implemented in a revision, the suggestions made in response to Rz26 seem to have not been implemented. These should be implemented. (Indeed, they relate precisely to the abstract issues I've outlined above.)

Finally, I encourage the authors to find experimental settings that demonstrate failure modes arising from gross violations of the assumptions of the present convergence result (perhaps arising due to nonconvexity, misspecification, lack of generalization).

**Audience:**

There is increasing interest in uncertainty quantification and related problems like quantile regression, which is the topic of this paper. As such, I believe a good fraction of TMLR's audience will be interested in these results.

**Claims And Evidence:**

The authors propose to minimize the expected pinball loss in order to train a quantile regression model, and claim that this can be more efficient than aiming to estimate a single quantile. A theoretical asymptotic guarantee is formalized, but the hypotheses of this result may be violated by 1) constraints in the proposed estimator and 2) misspecification of the inverse CDF, if I'm not mistaken, limiting the purview of the theory, and perhaps confusing readers. Experiments with deep lattice networks demonstrate promising performance, but don't shed too much light on the failure modes.